# GRAPH REPRESENTATIONAL LEARNING: WHEN DOES MORE EXPRESSIVITY HURT GENERALIZATION?

**Sohir Maskey**[1]*  **Raffaele Paolino**[2,3]    **Fabian Jogl**[4]
**Gitta Kutyniok**[2,3,5,6]    **Johannes F. Lutzeyer**[7]
[1]Aleph Alpha Research
[2]Department of Mathematics, LMU Munich
[3]Munich Center for Machine Learning (MCML)
[4]Machine Learning Research Unit, CAIML, TU Wien
[5]Institute for Robotics and Mechatronics, DLR-German Aerospace Center
[6]Department of Physics and Technology, University of Tromsø
[7]LIX, CNRS, École Polytechnique, Institut Polytechnique de Paris, France

## ABSTRACT

Graph Neural Networks (GNNs) are powerful tools for learning on structured data, yet the relationship between their expressivity and predictive performance remains unclear. We introduce a family of pseudometrics that capture different degrees of structural similarity between graphs and relate these similarities to generalization, and consequently, the performance of expressive GNNs. By considering a setting where graph labels are correlated with structural features, we derive generalization bounds that depend on the distance between training and test graphs, model complexity, and training set size. These bounds reveal that more expressive GNNs may generalize worse unless their increased complexity is balanced by a sufficiently large training set or reduced distance between training and test graphs. Our findings relate expressivity and generalization, offering theoretical insights supported by empirical results. Our code is available on GitHub.

## 1 INTRODUCTION

Graph Neural Networks (GNNs) (Scarselli et al., 2009; Bronstein et al., 2017) have become a central tool for learning representations of structured data. A major line of research has focused on improving their *expressivity*, that is, their capacity to distinguish non-isomorphic graphs, often evaluated with respect to the Weisfeiler-Lehman (WL) hierarchy of graph isomorphism tests (Weisfeiler & Lehman, 1968; Xu et al., 2019; Morris et al., 2020b; 2023b).

The relationship between expressivity and performance of GNNs remains poorly understood. While more expressive models are theoretically capable of distinguishing a broader range of non-isomorphic graphs, their practical effectiveness in real-world tasks is not always apparent. On the one hand, more expressive models have been shown to outperform standard architectures (Maron et al., 2019a; Bodnar et al., 2021; Bouritsas et al., 2023), even when their additional expressive power does not result in separating more non-isomorphic graphs in the benchmarks at hand (Zopf, 2022). For instance, the 1-WL test, and by extension simple message-passing neural networks (MPNNs) such as GIN (Xu et al., 2019), can separate all graphs in widely used datasets (Zopf, 2022; Kriege et al., 2020) and almost all random graphs (Babai et al., 1980), but performance is still far from saturated. On the other hand, less expressive models can outperform their more expressive counterparts. For example, Bechler-Speicher et al. (2024) show that models which completely discard the graph structure (hence, less-expressive than 1-WL) can outperform sophisticated GNNs in certain tasks.

These contrasting observations suggest a complex and nuanced relationship between expressivity and performance, raising two critical questions:

1. When does increased expressivity in GNNs help, and when does it hurt performance?

---

*Email: `sohir.maskey@aleph-alpha-research.com`. Work done while at LMU Munich.

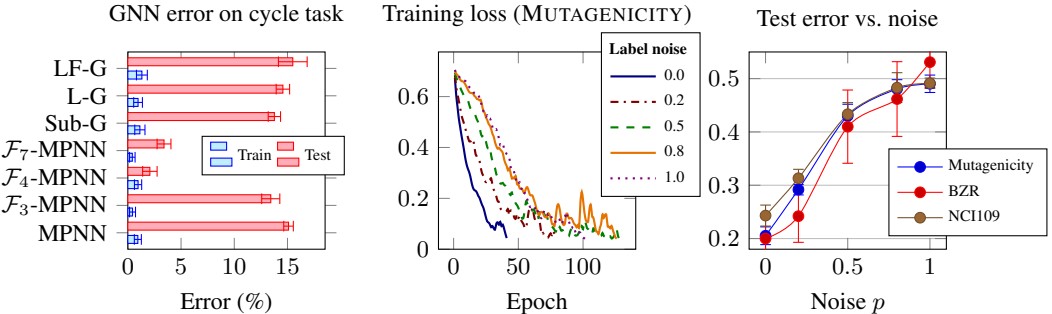

Figure 1: **Left:** Train–test errors for several GNN variants on a synthetic cycle-counting task; moderately expressive models such as $\mathcal{F}_4$-MPNN (Barceló et al., 2021), i.e., MPNNs augmented with cycle counts, generalize best, while more expressive ones tend to overfit. **Center:** Training-loss curves of a MPNN on MUTAGENICITY under increasing label noise $p$. **Right:** Corresponding test errors on BZR, MUTAGENICITY, and NCI109 rises sharply as label-structure correlation is essential for generalization (mean $\pm$ standard deviation across five seeds). See Appendix I.1 for more details.

    2. If a more expressive GNN performs better on a given task, is it due to its improved expressivity in terms of graph separability?

To explore these questions, we begin with a synthetic graph-classification task in which the labels are *designed* to depend on a known structural feature, namely the number of cycles. We train a sequence of models whose expressive power increases from a plain MPNN to an LF-GNN (Zhang et al., 2024). As Figure 1 (left) shows, *moderately* expressive models achieve the lowest test error, while the most expressive ones overfit, and performance deteriorates on unseen graphs. The result echoes a well-documented phenomenon in Euclidean deep learning: large neural networks can memorize arbitrary labels, yet they generalize only when there is genuine correlation between data and labels. Classical complexity measures such as the VC dimension or Rademacher complexity cannot account for this behavior in over-parametrized regimes (Zhang et al., 2017). A more plausible explanation is that generalization requires a balance between the model's inductive bias (in our case, its expressivity) and the structure–label correlation present in the data.

We further test this hypothesis on three real-world graph datasets: BZR, MUTAGENICITY, and NCI109 (Morris et al., 2020a). We follow the setup from Zhang et al. (2017) by progressively resampling the labels uniformly at random, i.e., introducing label noise, while leaving the graphs untouched. Figure 1 (center and right) shows that the training loss still converges to zero, confirming the network's ability to memorize, yet the test error rises sharply as soon as the correlation between the graphs and labels is destroyed. Together, these experiments indicate that *expressivity by itself is neither strictly harmful nor helpful*; model performance ultimately is determined by the model's ability to measure similarity in a way that reflects the task-relevant relation between graphs and labels. In this work, we set out to better understand this phenomenon.

## 1.1 OUR CONTRIBUTION

We formalize the empirical insight that generalization heavily depends on structure–label correlation as follows. We introduce a family of pseudometrics, called $\zeta$-Tree Mover Distances ($\zeta$-TMDs), each parameterized by a graph invariant (e.g., degree distributions or $k$-WL colors), which measure a specific level of expressivity. We then consider a graph classification setting where labels correlate with a fixed $\zeta$-TMD: two graphs are likely to share the same label if they are similar under the chosen pseudometric. This captures structure–label alignment, a property we empirically validate across several real-world graph learning tasks, where generalization depends critically on such alignment.

Within this framework, we derive data-dependent generalization bounds for models with fixed encoders (e.g., random GNN features) and expressive, end-to-end trainable GNNs (e.g., GIN, GAT, $k$-GNNs). Our bound (Theorem 4.1, Equation (4)) decomposes the generalization gap into two terms: a capacity term, depending on model width, weight norms, and maximum node degree, and a structural similarity term, which measures the distance between training and test graphs under

the chosen $\zeta$-TMD. This decomposition highlights that generalization improves when the model maps structurally similar graphs (with respect to the $\zeta$-TMD that correlates with the labels) to similar representations while keeping model capacity in check.

This perspective not only explains existing empirical observations but also guides model design. Deeper or more expressive GNNs can improve performance—but only if they enhance the structural similarity between train and test graphs. Otherwise, increased complexity may fail to improve label alignment, thereby degrading both generalization and computational efficiency. This trade-off is especially relevant since higher-order methods often introduce significant computational overhead. In Section 5, we formalize this phenomenon and show that, in a concrete setting, the optimal GNN is precisely as expressive as required to capture the relevant structure-label correlation—no more, no less.

## 1.2 RELATED WORK

**Expressivity in GNNs.** A common benchmark for GNN expressivity is the Weisfeiler-Leman ($k$-WL) hierarchy, which measures the ability to distinguish non-isomorphic graphs. Standard MPNNs are provably limited by 1-WL expressivity (Xu et al., 2019; Morris et al., 2019), while higher-order $k$-GNNs (Morris et al., 2020b) and subgraph-based models (Frasca et al., 2022) extend this by incorporating higher-order structural interactions. Structural and positional encodings (Vignac et al., 2020; Barceló et al., 2021) enhance expressivity by injecting global features into node representations. Jogl et al. (2024) offer a unifying view by showing that many expressive architectures can be simulated via input transformations followed by standard message passing.

However, expressivity often comes at a computational cost: while MPNNs scale linearly with edges ($\mathcal{O}(|E|)$), subgraph-based and higher-order models scale super-linearly (e.g., $\mathcal{O}(|V||E|)$ or $\mathcal{O}(|V|^k)$), making them less viable for large-scale graphs.

**Generalization in GNNs.** Generalization theory for GNNs traditionally relied on classical complexity measures such as VC-dimension (Scarselli et al., 2018), Rademacher complexity (Garg et al., 2020; Abbahaddou et al., 2025), and PAC-Bayes bounds (Liao et al., 2021). Graphon-based frameworks (Levie, 2024) establish bounds using covering numbers in continuous function spaces. Similarly, Maskey et al. (2022a; 2025) establish tighter bounds using graphon models, but under stronger assumptions on the data distribution and their analysis does not directly handle expressive GNNs.

Recent works increasingly focus on the interplay between expressivity and generalization. Morris et al. (2023a) relate 1-WL to VC-dimension in MPNNs. Li et al. (2025) analyze a trade-off between intra-class concentration and inter-class separation, but only for fixed graph encoders. Wang et al. (2024) study graphs sampled from manifolds, and Ma et al. (2021) model label–feature correlations in node classification, but neither framework extends to graph-level tasks or expressive GNNs.

Most recently, Vasileiou et al. (2025) derive tighter generalization bounds by combining covering number arguments with the robustness framework of Xu & Mannor (2012), using the fact that MPNNs are Lipschitz with respect to Tree and Forest distances. Their analysis does not model structure–label correlation and is limited to standard MPNNs. We refer to Appendix A for further discussion.

**Comparison with Prior Work.** While existing approaches have deepened our understanding of GNN generalization, many rely on idealized assumptions—such as known graphon distributions or fixed feature maps—that limit applicability to real-world graph learning. The work most closely related to ours is Ma et al. (2021), which models label-feature alignment but does not support graph-level tasks or trainable GNNs. In contrast, our framework explicitly models structure-label correlation using task-aligned pseudometrics and supports expressive, end-to-end trainable GNNs. This enables a fine-grained analysis of how architectural expressivity and task alignment interact. Our generalization bounds identify when increased expressivity may improve performance—and when it leads to overfitting—addressing a key open question posed by Morris et al. (2024).

## 2 PRELIMINARIES

Let $\mathcal{G}$ denote the set of all simple, undirected graphs, and let $G = (V, E) \in \mathcal{G}$, where $V$, or $V(G)$, is the set of nodes and $E$ is the set of edges. For any node $v \in V$, the *neighborhood* of $v$ is defined as: $\mathcal{N}(v) := \{u \in V \mid \{v, u\} \in E\}$.

**Definition 2.1** (Graph Invariant and CRA). A *graph invariant* is a function $\zeta : \mathcal{G} \to \mathcal{C}$ that assigns a value to each graph such that for any isomorphic graphs $G$ and $H$, it holds that $\zeta(G) = \zeta(H)$.

A *color refinement algorithm (CRA)* $\zeta_{(\cdot)}$ is a mapping that assigns to each graph $G$ a function $\zeta_G : V(G) \to P$ such that for any graph $H$ isomorphic to $G$, and any isomorphism $h : V(H) \to V(G)$, the CRA satisfies $\zeta_G(v) = \zeta_H(h(v))$ for all $v \in V(G)$.

We note that every CRA $\zeta_{(\cdot)}$ induces a graph invariant $\zeta$ by aggregating the vertex labels into a multiset. Specifically, the induced graph invariant is defined as $\zeta(G) \coloneqq \{\!\{\zeta_G(v)\}\!\}_{v \in V(G)}$, where $\{\!\{\cdot\}\!\}$ denotes a multiset. Throughout this work, we use the terms "graph invariant" and "CRA" interchangeably when the context allows.

Graph invariants vary in their ability to distinguish between graphs. We say that a graph invariant $\zeta$ is *more expressive* than another graph invariant $\theta$ if $\zeta(G) = \zeta(H)$ implies $\theta(G) = \theta(H)$ for all $G, H \in \mathcal{G}$.

**Definition 2.2** (Message Passing Neural Networks (MPNNs)). For a graph $G = (V, E)$ with node features $x \in \mathbb{R}^{|V| \times F}$, an *MPNN* updates node $v \in V$ at layer $t$ as:

$$x_v^{(t+1)} = f^{(t+1)} \left( x_v^{(t)}, \square_{u \in \mathcal{N}(v)} \left\{\!\left\{ g^{(t+1)} \left( x_u^{(t)} \right) \right\}\!\right\} \right),$$

where $f^{(t+1)}$ and $g^{(t+1)}$ are MLPs, and $\square$ denotes a permutation-invariant aggregation function.

We focus on sum aggregation, but our framework naturally extends to mean or weighted sum aggregation. After the final message passing layer, node features are typically aggregated into a graph-level representation, followed by a final MLP.

Similar to MPNNs, the *1-Weisfeiler-Lehman (1-WL) Test* updates the node features of input graphs through local neighborhood aggregation. However, a key distinction is that in 1-WL, both the message and update functions are necessarily injective. 1-WL provides a tight upper bound on the expressivity of MPNNs (Xu et al., 2019; Morris et al., 2020b).

To quantify similarity between graphs, we use the notion of pseudometrics. In particular, every graph invariant naturally induces a pseudometric $d_\zeta(G, H)$ with $d_\zeta(G, H) = 0$ if $\zeta(G) = \zeta(H)$ and $d_\zeta(G, H) = 1$ otherwise. However, such pseudometrics only indicate whether two graphs are distinguishable by $\zeta$ (distance 0 or 1). To capture finer structural similarities, general pseudometrics are often employed. Conversely, any pseudometric $d$ can induce a graph invariant $\zeta_d$ by anchoring comparisons to a fixed graph $A \in \mathcal{G}$ with $\zeta_{d,A}(G) := d(G, A)$. Thus, pseudometrics can be seen as generalizations of graph invariants, offering richer measures of graph similarity.

The *Tree Mover's Distance (TMD)* (Chuang & Jegelka, 2022) is a pseudometric that quantifies the dissimilarity between two graphs. Formally, for graphs $G$ and $H$, and a depth $t$, the TMD is defined as the Wasserstein distance between their distributions of rooted trees up to depth $t$:

$$\text{TMD}^t(G, H) = \text{Wasserstein} \left( \mathcal{T}^t(G), \mathcal{T}^t(H) \right),$$

where $\mathcal{T}^t(G)$ and $\mathcal{T}^t(H)$ are the multisets of rooted trees of depth $t$ generated by the 1-WL test for $G$ and $H$, respectively. For further details, we refer to Appendix C.

## 3 GENERALIZED TREE MOVER'S DISTANCE FOR STRONGLY SIMULATABLE COLORINGS

We now extend the concept of the TMD to a broader class of CRAs that can be *strongly simulated* by the 1-WL test. For detailed proofs of the results presented in this section, refer to Appendix C.2.

A CRA $\zeta$ is said to be *strongly simulatable* if, for any graph $G$, running $t$ iterations of the CRA on $G$ can be simulated by running $t$ iterations of the 1-WL test on a suitably transformed graph $R^\zeta(G)$. This transformation, called the *strong simulation under* $\zeta$, ensures that the colorings at each 1-WL iteration on $R^\zeta(G)$ are at least as expressive as those of $\zeta$ (Jogl et al., 2024) on $G$. See Appendix B for more details and examples of CRAs and their corresponding transformed graphs.

For any strongly simulatable CRA $\zeta$, we define a generalized pseudometric as the TMD between the strong simulations of the graphs under $\zeta$.

**Definition 3.1.** Let $\zeta$ be a strongly simulatable CRA. For any depth $t > 0$, the $\zeta$-TMD is defined as:

$$\zeta\text{-TMD}^t(G, H) := \text{TMD}^t(R^\zeta(G), R^\zeta(H)), \tag{1}$$

where $R^\zeta(G)$ and $R^\zeta(H)$ are the strong simulations of $G$ and $H$ under $\zeta$, respectively.

**Proposition 3.2.** *Let $\zeta$ be a strongly simulatable CRA. For every $t > 0$, $\zeta$-TMD$^t$ is a pseudometric.*

Similar to the standard TMD, $\zeta$-TMD$^t(G, H)$ can be zero even if $G \neq H$, as it is a pseudometric. Nonetheless, it can distinguish graphs that are differentiable by the color refinement algorithm $\zeta$ within $T$ iterations.

**Proposition 3.3.** *Let $\zeta$ be a strongly simulatable CRA. If two graphs $G$ and $H$ are distinguished by $\zeta$ after $T$ iterations, then $\zeta$-TMD$^{T+1}(G, H) > 0$.*

Another immediate consequence is that MPNNs corresponding to the CRA $\zeta$, referred to as $\zeta$-*MPNNs*, are Lipschitz continuous with respect to the $\zeta$-TMD. Specifically, we have the following result:

**Theorem 3.4.** *Let $\zeta$ be a strongly simulatable CRA, and let $h : \mathcal{G} \to \mathbb{R}^K$ be a $\zeta$-MPNN with $T$ layers, where the message and update functions are Lipschitz continuous with Lipschitz constants bounded by $L_{g^{(t)}}$ and $L_{f^{(t)}}$, respectively. Suppose $h$ includes a global sum pooling layer followed by a Lipschitz continuous classifier $c$ with Lipschitz constant $L_c$. Then, for any graphs $G$ and $H$,*

$$\|h(G) - h(H)\| \leq L \cdot \zeta\text{-TMD}^{T+1}(G, H),$$

*where $L = L_c 2^T \prod_{t=1}^{T} L_{f^{(t)}} L_{g^{(t)}}$ and $\| \cdot \|$ denotes the Euclidean vector norm.*

The Lipschitz property established in Theorem 3.4 plays a key role in deriving the generalization bounds for $\zeta$-MPNNs presented in Section 4. Next, we illustrate the Lipschitz continuity using the example of $\mathcal{F}$-MPNNs.

### 3.1 Example: $\mathcal{F}$-MPNNs

The $\mathcal{F}$-*Weisfeiler-Leman ($\mathcal{F}$-WL) test* generalizes the 1-WL test by incorporating features derived from a finite family of graphs, $\mathcal{F} \subset \mathcal{G}$. These features, often referred to as *motifs or patterns*, are used to enhance node representations.

Specifically, for each node $v$ in a graph $G$, the feature vector of $v$ is augmented with counts of patterns in $\mathcal{F}$ that include $v$. Formally, the augmented feature vector is defined as:

$$\tilde{x}_v = \left( x_v \, , \, \text{cnt}(P_1, G; v) \, , \, \ldots \, , \, \text{cnt}(P_{|\mathcal{F}|}, G; v) \right),$$

where $\text{cnt}(P, G; v)$ represents the number of occurrences of the pattern $P$ in $G$ such that $v$ is part of the pattern. These counts can for example be (injective) homomorphism counts (Bouritsas et al., 2023) or cycle basis counts (Yan et al., 2024).

If $\mathcal{F} \subset \tilde{\mathcal{F}}$, then $\tilde{\mathcal{F}}$-WL is more expressive than $\mathcal{F}$-WL (Barceló et al., 2021; Bouritsas et al., 2023).

Correspondingly, $\mathcal{F}$-*MPNNs* incorporate these motif counts into their message-passing scheme. Clearly, $\mathcal{F}$-WL can be strongly simulated via a transformed graph $R^{\mathcal{F}}(G)$ that includes the motif counts as node features.

**Corollary 3.5.** *Let $h$ be an $\mathcal{F}$-MPNN with $T$ layers. Then, there exists a constant $L$ such that for any graphs $G$ and $H$,*

$$\|h(G) - h(H)\| \leq L \cdot \mathcal{F}\text{-TMD}^{T+1}(G, H). \tag{2}$$

We emphasize that this is just one example; analogous definitions of $\zeta$-TMDs and corresponding versions of Corollary 3.5 can be derived for other GNN architectures, such as $k$-GNNs (Morris et al., 2019), see Appendix C.3.

# 4 GENERALIZATION BOUNDS WITH RESPECT TO TREE MOVER'S DISTANCE

In this section, we establish generalization bounds for GNNs using our generalized TMD framework.

## 4.1 PROBLEM SETUP AND ASSUMPTIONS

We consider a classification task where each data point is a graph equipped with node features. Formally, let $\mathcal{G}_{\mathrm{tr}}$ and $\mathcal{G}_{\mathrm{te}}$ denote the fixed sets of training and test graphs, respectively. We assume that there exists a constant $B > 0$ such that the norm of each node feature is bounded, i.e.,

$$\|x(G)_i\|_2 \leq B \quad \forall G \in \mathcal{G}_{\mathrm{tr}} \cup \mathcal{G}_{\mathrm{te}}, \forall i \in V(G).$$

Each graph $G$ is assigned a label $y_G \in \{1, \ldots, K\}$. We assume that, for each class $k$, there exists a Lipschitz continuous function $\eta_k : \mathcal{G} \to [0, 1]$ with respect to some pseudometric pm such that

$$\Pr(y_G = k \mid G) = \eta_k(G),$$

and set $C \coloneqq \max_{k \in \{1,\ldots,K\}} \mathrm{Lip}(\eta_k)$. If the labels are sampled according to such functions $\eta_k$, we say that the labels $y$ are *strongly correlated* with pm and write $y \sim \mathrm{pm}$.

Furthermore, we define $\xi_{\mathrm{pm}}$ as the distance between the training set and the test set:

$$\xi_{\mathrm{pm}} = \mathrm{pm}\left(\mathcal{G}_{\mathrm{te}}, \mathcal{G}_{\mathrm{tr}}\right) \coloneqq \max_{G \in \mathcal{G}_{\mathrm{te}}} \min_{H \in \mathcal{G}_{\mathrm{tr}}} \mathrm{pm}(G, H). \tag{3}$$

Given the set of labeled graphs $\mathcal{G}_{\mathrm{tr}}$ the task of graph-level supervised learning is to learn a classifier $h : \mathcal{G} \to \mathbb{R}^K$ from a function family $\mathcal{H}$. Given a classifier $h \in \mathcal{H}$, the classification for a graph $G$ is obtained by

$$\tilde{y}_G = \operatorname*{argmax}_{k \in \{1,\ldots,K\}} h(G)[k],$$

where $h(G)[k]$ refers to the $k$-th entry of $h(G)$.

For the observed graph labels $(y_G)_{G \in \mathcal{G}_{\mathrm{tr}}}$, the *empirical margin loss* of $h$ on $\mathcal{G}_{\mathrm{tr}}$ for a margin $\gamma \geq 0$ is defined as

$$\widehat{\mathcal{L}}_{tr}^{\gamma}(h) \coloneqq \frac{1}{N_{\mathrm{tr}}} \sum_{G \in \mathcal{G}_{\mathrm{tr}}} \mathbb{1}[h(G)[y_G] \leq (\gamma + \max_{k \neq y_G} h(G)[k])].$$

Here, $\mathbb{1}$ is the indicator function. The empirical margin loss of $h$ on $\mathcal{G}_{\mathrm{te}}$ is defined equivalently. The expected margin loss is defined as follows, $\mathcal{L}_{\mathrm{te}}^{\gamma}(h) \coloneqq \mathbb{E}_{y_G \sim \Pr(y_G | G), G \in G_{\mathrm{te}}}\left[\widehat{\mathcal{L}}_{\mathrm{te}}^{\gamma}(h)\right].$

## 4.2 MAIN RESULTS

In this section, we derive generalization bounds for GNNs, where we focus on the standard approach involving end-to-end trainable GNNs.

We consider the GNN to be the composition of two functions. Specifically, let $e : \mathcal{G} \to \mathbb{R}^b$ denote the graph embedding network, which maps a graph $G \in \mathcal{G}$ into a $b$-dimensional latent space, and let $c : \mathbb{R}^b \to \mathbb{R}^K$ denote the classifier, which maps embeddings to class scores. Consequently, the hypothesis space for these models can be written as:

$$\mathcal{H} = \mathcal{C} \circ \mathcal{E},$$

where $\mathcal{E}$ represents the space of graph embedding networks and $\mathcal{C}$ represents the class of MLP classifiers. For end-to-end learnable GNNs, both $\mathcal{E}$ and $\mathcal{C}$ are trainable. In contrast, for models with fixed encoders, only $\mathcal{C}$ is trainable, while $\mathcal{E}$ remains fixed.

### 4.2.1 END-TO-END LEARNABLE GNNS

Let $\zeta$ represent a CRA that can be strongly simulated by 1-WL. In this context, we consider the hypothesis space $\mathcal{H}_{\zeta} = \mathcal{C} \circ \mathcal{E}_{\zeta}$, where $\mathcal{E}_{\zeta}$ is the set of all $\zeta$-MPNNs of depth $T$, where each layer consists of a message function $g^{(t)}$ and an update function $f^{(t)}$. Both $g^{(t)}$ and $f^{(t)}$ are MLPs with a

maximum hidden dimension of $b$ and may contain an arbitrary number of layers. The weight matrices across all message and update functions are denoted by $\{W_i\}_{i=1}^P$.

Let $\mathcal{C}$ denote the set of MLP classifiers with $L$ layers and a maximum hidden dimension of $b$. The weight matrices in the MLP classifier are denoted by $\{\tilde{W}_l\}_{l=1}^L$.

We now present the generalization bound for end-to-end learnable GNNs.

**Theorem 4.1.** *Suppose that $y \sim \zeta\text{-TMD}^{T+1}$. Under mild assumptions (see Appendix D), for any $\gamma > 0$ and $0 < \alpha < \frac{1}{4}$, with probability at least $1 - \delta$ over the sample of training labels $y_{\mathrm{tr}}$, we have for any $\tilde{h} \in \mathcal{H}_\zeta$*

$$\mathcal{L}_{\mathrm{te}}^0(\tilde{h}) \leq \widehat{\mathcal{L}}_{\mathrm{tr}}^\gamma(\tilde{h}) + \mathcal{O}\left( \frac{b\left(\sum_i \|W_i\|_2^2 + \sum_l \|\tilde{W}_l\|_2^2\right)}{N_{\mathrm{tr}}^{2\alpha}(\gamma/8)^{2/D}} \xi_\zeta^{2/D} + \frac{b^2 \ln\left(2bDC(2dB)^{1/D}\right)}{N_{\mathrm{tr}}^{2\alpha}\gamma^{1/D}\delta} + CK\xi_\zeta \right),$$

*where $\xi_\zeta := \zeta\text{-TMD}^{T+1}(\mathcal{G}_{te}, \mathcal{G}_{tr})$ is defined in Equation (3), $D$ represents the total number of learnable weight matrices, $b$ denotes the maximum hidden dimension, $d$ is the maximum degree of nodes in the graphs, and $C$ serves as an upper bound on the spectral norm of all weight matrices.*

Theorem 4.1 highlights key factors influencing generalization in learnable graph classifiers, which are the structural similarity $\xi_\zeta$ under $\zeta$-TMD, model complexity $\left(D, b, \{\|W_i\|_2\}_i, \{\|\tilde{W}_l\|_2\}_l\right)$, graph properties such as maximum degree $d$, and training set size $N_{\mathrm{tr}}$. The structural similarity term $\xi_\zeta$ emphasizes the importance of diverse training data. This agrees with empirical observations by Southern et al. (2025), who demonstrated that augmenting graph representation to maximize TMD dissimilarity improves predictive performance and generalization. These findings underscore the significance of diverse training data (via task-specific augmentations) and balancing model complexity to achieve robust graph-based learning.

The proof of Theorem 4.1, given in Appendix E, follows a standard PAC-Bayesian approach (Neyshabur et al., 2018) and extends it to the correlated setting, following the approach of (Ma et al., 2021). It leverages the Lipschitz continuity of $\zeta$-MPNNs, established in Theorem 3.4, to derive the generalization bound. Throughout the remainder of this paper, we use the following simplified version of the bound from Theorem 4.1:

$$\mathcal{L}_{\mathrm{te}}^0(\tilde{h}) \;\leq\; \widehat{\mathcal{L}}_{\mathrm{tr}}^\gamma(\tilde{h}) \;+\; \mathcal{O}\Big( \underbrace{\tfrac{\mathrm{MC}(\mathcal{C}\circ\mathcal{E}_\zeta)}{N_{\mathrm{tr}}^{2\alpha}\,\gamma^{1/D}\,\delta}}_{\text{complexity term}} \;+\; \underbrace{C\,\xi_\zeta}_{\substack{\text{structural}\\\text{similarity term}}} \Big). \tag{4}$$

Here, $\mathrm{MC}(\mathcal{C} \circ \mathcal{E}_\zeta)$ captures the model complexity, i.e., spectral norms of all learnable weight matrices, hidden dimensions, and maximal node degree of the graphs. Notably, the model complexity may increase if the graph transformation $R^\zeta$ enlarges the size, maximum degree, or node feature dimension of the original input graphs.

The structural similarity term, $\xi_\zeta$, depends on the alignment between training and test graphs under $\zeta$-TMD. Whether $\xi_\zeta$ increases or decreases with more expressive networks depends on the task.

The same PAC-Bayes machinery yields analogous bounds when the graph encoder is frozen and only the MLP head is trained. We relegate the formal statement and proof to Appendix F to keep the main exposition focused on the end-to-end learnable case.

## 5 WHEN DOES MORE EXPRESSIVITY HURT?

In this section, we explore two scenarios within our framework: first, we identify conditions under which augmenting expressivity beyond task-specific requirements can negatively impact generalization. Second, we highlight conditions where increasing expressivity to accurately capture task-specific features does not degrade the bound in Theorem 4.1. This gives a possible explanation of why such graph classifiers achieve an optimal balance between expressivity and generalization, resulting in superior performance.

To formalize this, consider a classification task where the labels $y_G$ are strongly correlated with the pseudometric $\mathcal{F}\text{-TMD}^{T+1}$ for a specific set of substructures $\mathcal{F} \subset \mathcal{G}$. Assume that all MPNNs

considered in this section have $T$ layers. Let $\mathcal{F}' \subsetneq \mathcal{F} \subsetneq \tilde{\mathcal{F}} \subset \mathcal{G}$ be finite sets of graphs, where $\mathcal{F}'$-MPNNs are less expressive than $\mathcal{F}$-MPNNs, which in turn are less expressive than $\tilde{\mathcal{F}}$-MPNNs. Denote the corresponding hypothesis spaces as $\mathcal{H}_{\mathcal{F}'}, \mathcal{H}_{\mathcal{F}}$, and $\mathcal{H}_{\tilde{\mathcal{F}}}$.

Our analysis shows that increasing expressivity to the level required to capture task-relevant features (e.g., transitioning from $\mathcal{F}'$-MPNNs to $\mathcal{F}$-MPNNs) generally preserves generalization. However, further increasing expressivity beyond this necessary level (e.g., to $\tilde{\mathcal{F}}$-MPNNs) can degrade generalization significantly.

**Theorem 5.1.** *Consider the setting above. Then, for any $\gamma > 0$ and $\alpha < \frac{1}{4}$, with probability at least $1 - \delta$ over the sample of training labels $y_{\text{tr}}$,*

    *i)* *the test loss of any $\mathcal{F}'$-MPNN classifier $h'$, satisfies*

$$\mathcal{L}_{\text{te}}^0(h') \leq \mathcal{L}_{\text{tr}}^\gamma(h') + \mathcal{O}\left( \frac{\text{MC}(\mathcal{H}_{\mathcal{F}'})\xi_{\mathcal{F}}^{1/D}}{N_{tr}^{2\alpha}\gamma^{1/D}\delta} + C\xi_{\mathcal{F}} \right).$$

    *where $C$ is described in Theorem 4.1 and $\xi_{\mathcal{F}} = \mathcal{F}\text{-TMD}^{T+1}(\mathcal{G}_{\text{tr}}, \mathcal{G}_{\text{te}})$.*

    *ii)* *the test loss of any $\mathcal{F}$-MPNN classifier $h$, satisfies*

$$\mathcal{L}_{\text{te}}^0(h) \leq \mathcal{L}_{\text{tr}}^\gamma(h) + \mathcal{O}\left( \frac{\text{MC}(\mathcal{H}_{\mathcal{F}})\xi_{\mathcal{F}}^{1/D}}{N_{tr}^{2\alpha}\gamma^{1/D}\delta} + C\xi_{\mathcal{F}} \right).$$

    *iii)* *the test loss of any $\tilde{\mathcal{F}}$-MPNN classifier $\tilde{h}$, satisfies*

$$\mathcal{L}_{\text{te}}^0(\tilde{h}) \leq \mathcal{L}_{\text{tr}}^\gamma(\tilde{h}) + \mathcal{O}\left( \frac{\text{MC}(\mathcal{H}_{\tilde{\mathcal{F}}})\xi_{\tilde{\mathcal{F}}}^{1/D}}{N_{tr}^{2\alpha}\gamma^{1/D}\delta} + C\xi_{\tilde{\mathcal{F}}} \right).$$

    *where $\xi_{\tilde{\mathcal{F}}} = \tilde{\mathcal{F}}\text{-TMD}^{T+1}(\mathcal{G}_{\text{tr}}, \mathcal{G}_{\text{te}})$.*

Theorem 5.1 highlights the critical role of structural alignment between training and test graphs in determining generalization performance. In cases i) and ii), the same $\mathcal{F}$-TMD governs the similarity between training and test graphs, ensuring that the bound remains controlled. However, in case iii), introducing a more expressive GNN that captures features beyond those captured by $\mathcal{F}$-TMD (with which the labels are strongly correlated) leads to a higher structural discrepancy $\xi_{\tilde{\mathcal{F}}} \geq \xi_{\mathcal{F}}$, see Lemma G.1. These findings underscore the importance of aligning model expressivity with the structural requirements of the task, as excessive expressivity can increase our generalization bound in Theorem 4.1.

## 6 EXPERIMENTS

In this section, we evaluate GNNs on both synthetic and real-world graph datasets. We introduce two tasks to evaluate how structural similarity between training and test graphs, as well as task-relevant expressivity, impact classification performance and generalization. All experiments use 10-fold cross-validation, and we report the mean accuracy or error. Additional experiments and tasks, details and extended results are provided in Appendix I.

**Task 1: Median-Based Labeling with Cycle Counts.** We generate 3,000 random graphs from Erdős–Rényi, Barabási–Albert, and Stochastic Block Model distributions. The sum of 3-cycle and 4-cycle counts is computed for each graph, and graphs with counts below the dataset's median receive label 0, while those above receive label 1. We evaluate multiple GNN variants, including standard MPNNs, $\mathcal{F}_l$-MPNNs where $\mathcal{F}_l$ contains cycles up to length $l$, Subgraph GNNs (Sub-G), Local 2-GNNs (L-G), and Local Folklore 2-GNNs (LF-G). Model expressivity increases strictly in this order. We report training and test accuracies at both the final and the epoch with the best validation performance. Results can be found in Table 1, and more details and experiments in Appendix I.

Table 1: Test accuracy on Erdős–Rényi graphs for Task 1. All GNNs achieve a train accuracy greater than 0.99. More results in Table 4 in Appendix I.

| Model | Test Accuracy |
|---|---|
| LF-G | $0.8450 \pm 0.0135$ |
| L-G | $0.8543 \pm 0.0063$ |
| Sub-G | $0.8623 \pm 0.0058$ |
| $\mathcal{F}_7$-MPNN | $0.9660 \pm 0.0065$ |
| $\mathcal{F}_4$-MPNN | $\mathbf{0.9793 \pm 0.0068}$ |
| $\mathcal{F}_3$-MPNN | $0.8657 \pm 0.0085$ |
| MPNN | $0.8490 \pm 0.0045$ |

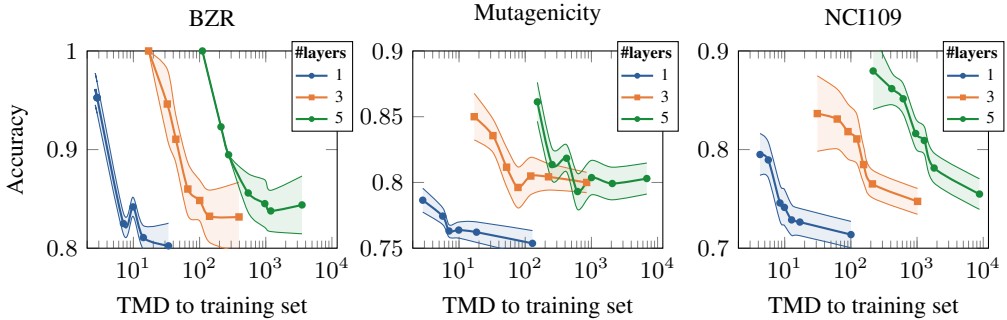

Figure 2: Accuracy of a GIN with 1, 3, and 5 layers versus TMD (log scale) to the training dataset.

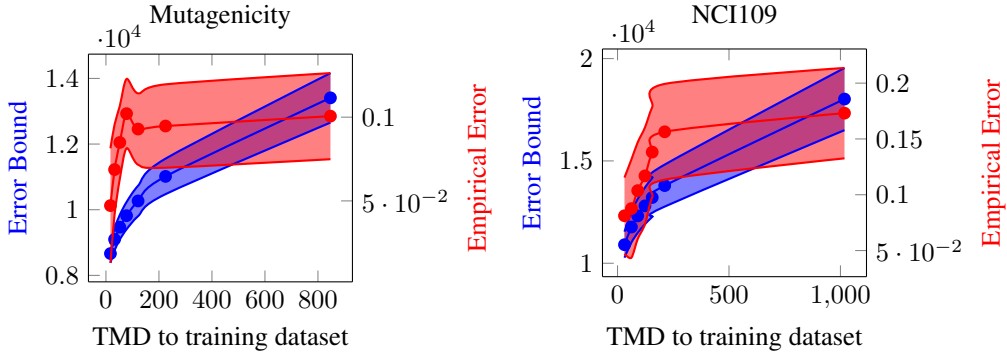

Figure 3: Error-bound curves for Mutagenicity, and NCI109. Each plot shows our theoretical bound (blue, left axis) and the empirical generalization error (red, right axis) as a function of TMD to the training set. Shaded areas indicate $\pm 1$ standard deviation across 10 random splits.

**Task 2: Real-World Datasets** We evaluate our framework on six graph classification datasets from the TUDataset (Morris et al., 2020a). Results on BZR, MUTAGENICITY, and NCI109 include: Figure 2, which plots test accuracy versus $\zeta$-TMD distance, and Figure 3, which compares our bound to the observed generalization gap. Additional results on PROTEINS, AIDS, COX2, and experiments on fixed encoders via molecular fingerprints (Gainza et al., 2019) appear in Appendix I.

**Results and Discussion.** In Task 1, GNNs that explicitly incorporate task-relevant cycle information, in particular $\mathcal{F}_4$-MPNNs, outperform MPNNs and expressive GNNs like Local Folklore 2-GNNs. Since the labels in Task 1 strongly correlated with $\mathcal{F}_4$-TMD, these results align with our theoretical findings in Section 5: GNNs that effectively leverage features strongly correlated with the task generalize better than more expressive models. On real-world datasets, we observe that classification accuracy declines as test graphs become more distant from the training set, in line with our theoretical insights in Theorem 4.1. Importantly, our bound closely aligns with the observed generalization gap across these datasets. By explicitly capturing structure–label correlation, our framework yields significantly tighter generalization bounds compared to standard PAC-Bayes bounds (Liao et al., 2021), which are often orders of magnitude larger (e.g., on the order of $10^{16}$ compared to our $10^4$).

## 7 CONCLUSION

We introduced a framework for analyzing GNN generalization in settings where graph labels correlate with different pseudometrics. Our analysis provided generalization bounds that emphasize the role of structural similarity between training and test data. We show that increasing expressivity does not necessarily degrade generalization if it aligns with the task. However, both theoretical and empirical results show that excessive expressivity can worsen generalization and predictive performance.

Empirical results confirm that GNNs whose embeddings align with task-relevant structures, i.e., those that are proven to be Lipschitz continuous with respect to pseudometrics strongly correlated with

the labels, achieve better generalization. We show that performance degrades for test graphs that are structurally distant from the training set, supporting our theoretical findings. Additionally, we identify cases where increased expressivity can either improve or hinder generalization, depending on task alignment, providing insights for GNN design.

**Limitations.** While our framework enables comparisons between $\mathcal{F}$-MPNNs, in general the dependence of our bound in Theorem 4.1 on different TMDs makes direct model comparisons challenging. These bounds serve as qualitative guidelines rather than precise estimates, as they are not tight and may not reflect practical performance. Moreover, the relevant pseudometric is often unknown and possibly expensive to compute, limiting direct applicability and leaving the identification of suitable metrics an open problem.

**Future Work.** Our generalization bounds improve with increased structural similarity between test and train graphs. Future work could explore augmentation techniques that generate synthetic graphs to improve this similarity, thereby boosting generalization.

## ETHICS STATEMENT

This work focuses on theoretical analysis of GNNs and does not involve experiments on human subjects, sensitive personal data, or applications with direct societal risks. The datasets referenced are publicly available benchmark graphs, and no private or restricted data was used. Potential ethical concerns related to misuse are minimal, as the contributions are mainly theoretical and methodological. We explicitly acknowledge that LLMs were used only for polishing sentence clarity and grammar, not for generating research ideas, proofs, or results.

## REPRODUCIBILITY STATEMENT

We have taken multiple steps to ensure reproducibility of our results. All theoretical claims are accompanied by rigorous proofs, presented in detail in the appendix. Assumptions underlying the theorems are explicitly stated, and definitions are given in full to allow independent verification. In addition, we provide open-source code to reproduce illustrative experiments and examples, which is available at GitHub.

## ACKNOWLEDGEMENTS

S. Maskey is funded by the NSF-Simons Research Collaboration on the Mathematical and Scientific Foundations of Deep Learning (MoDL) (NSF DMS 2031985). R. Paolino is funded by the Munich Center for Machine Learning (MCML). F. Jogl is funded by the Center for Artificial Intelligence and Machine Learning (CAIML) at TU Wien. G. Kutyniok acknowledges support by the DAAD programme Konrad Zuse Schools of Excellence in Artificial Intelligence, sponsored by the Federal Ministry of Education and Research. G. Kutyniok acknowledges support by the project "Genius Robot" (01IS24083), funded by the Federal Ministry of Education and Research (BMBF). G. Kutyniok acknowledges support by the gAIn project, which is funded by the Bavarian Ministry of Science and the Arts (StMWK Bayern) and the Saxon Ministry for Science, Culture and Tourism (SMWK Sachsen). G. Kutyniok acknowledges partial support by the Munich Center for Machine Learning (MCML), as well as the German Research Foundation under Grants DFG-SPP-2298, KU 1446/31-1 and KU 1446/32-1. Furthermore, G. Kutyniok is supported by LMUexcellent, funded by the Federal Ministry of Education and Research (BMBF) and the Free State of Bavaria under the Excellence Strategy of the Federal Government and the Länder as well as by the Hightech Agenda Bavaria. J. Lutzeyer is supported by the French National Research Agency (ANR) via the "GraspGNNs" JCJC grant (ANR-24-CE23-3888).

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

NOTATION

| | |
|---|---|
| $c$ | (MLP) classifier |
| $G$ | A graph $G = (V, E)$ |
| $\mathcal{G}$ | The set of all graphs |
| $\mathcal{N}(v)$ | Neighborhood of vertex $v$ in graph $G$ |
| $x_v$ | Feature of node $v$ |
| $\|\cdot\|_2$ | Frobenius norm |
| $\|\cdot\|$ | Spectral norm |
| $\mathrm{TD}_w$ | Tree Distance weighted by function $w$ |
| $\mathrm{TMD}_w^T$ | Tree Mover's Distance at depth $T$ with weight $w$ |
| $\mathcal{T}_G^T$ | Multiset of depth-$T$ computation trees for graph $G$ |
| $P$ | Prior distribution in PAC-Bayes framework |
| $Q$ | Posterior distribution in PAC-Bayes framework |
| $\mathcal{L}^\gamma$ | Expected margin loss with margin $\gamma$ |
| $\hat{\mathcal{L}}^\gamma$ | Empirical margin loss with margin $\gamma$ |
| $h$ | Classifier function (e.g., a MPNN + MLP) |
| $\mathcal{H}$ | Hypothesis space |
| $\mathcal{E}$ | Graph embedding networks (e.g., a MPNN) |
| $\mathcal{C}$ | Final classifer (e.g., a MLP) |
| $\mathrm{Pr}(\cdot)$ | Probability function |
| $\mathcal{G}_{\mathrm{tr}}$ | Training graphs |
| $\mathcal{G}_{\mathrm{te}}$ | Test graphs |
| $\mathrm{TMD}_w^L(\mathcal{G}_{\mathrm{tr}}, \mathcal{G}_{\mathrm{te}})$ | Distance between $\mathcal{G}_{\mathrm{tr}}$ and $\mathcal{G}_{\mathrm{te}}$ |
| $N_{\mathrm{tr}}$ | Number of training graphs |
| $N_{\mathrm{te}}$ | Number of test graphs |
| $b$ | Maximal hidden dimension of classifier function |
| $B$ | Maximum $L^2$-norm of input node features |
| $d$ | Maximum degree of all graphs in training or test set |
| $d(f)$ | Depth of MLP $f$ |
| $T$ | Number of MPNN layers |
| $C$ | Maximum Frobenius norm of any learnable weight matrix |
| $D$ | Number of learnable weight matrices in hypothesis space |
| $g^{(t)}$ | Message function in layer $t$ |
| $f^{(t)}$ | Update function in layer $t$ |

# A    DETAILED RELATED WORK

## A.1    EXPRESSIVITY OF GNNS

Expressivity in standard neural networks is often associated with their ability to approximate functions within a specific function class. For example, early works showed that MLPs can approximate any continuous function (Cybenko, 1989; Hornik et al., 1989). In the context of GNNs, however, expressivity is more commonly measured by the ability to distinguish between non-isomorphic graphs. This focus stems from computational challenges associated with achieving universality in GNNs and is further supported by the Stone-Weierstrass theorem: a GNN that can distinguish all graphs is also capable of approximating any continuous function on graphs (Chen et al., 2019; Dasoulas et al., 2020). Consequently, practical research often centers on characterizing the distinguishing power of specific GNN architectures (Xu et al., 2019; Morris et al., 2023b).

Xu et al. (2019); Morris et al. (2019) established that the expressive power of standard MPNNs is limited by the 1-WL test. To overcome this limitation, later works introduced higher-order GNNs based on $k$-WL and its local variants (Maron et al., 2018; Morris et al., 2020b; Geerts & Reutter, 2022). These models are theoretically universal (Maron et al., 2019b; Keriven & Peyré, 2019), meaning they can distinguish any non-isomorphic graphs and approximate continuous functions on graphs. However, their expressivity comes at the cost of exponential time and space complexity with respect to $k$, making them impractical for large-scale applications. To reduce this complexity, Morris et al. (2020b); Zhang et al. (2024) proposed local $k$-WL variants, while Abboud et al. (2022) introduced $k$-hop GNNs, which expand the receptive field to $k$-hop neighborhoods. Despite these improvements, the computational complexity of these approaches remains exponential in $k$.

Subgraph-based models further enhance expressivity by decomposing graphs into subgraphs and aggregating their information (Papp & Wattenhofer, 2022; Bevilacqua et al., 2021; You et al., 2021; Frasca et al., 2022; Huang et al., 2022). Although these models are more expressive than 1-WL, their power is bounded by 3-WL (Frasca et al., 2022). Moreover, subgraph GNNs increase computational complexity significantly, often scaling quadratically or cubically with the number of nodes $N$, which worsens the computational complexity of standard MPNNs by a factor of $N$.

Most subgraph-based GNNs associate a family of subgraphs with specific nodes or edges by either deleting or marking nodes. However, other strategies for subgraph representation have also been explored. For instance, Michel et al. (2023); Graziani et al. (2024); Paolino et al. (2024) focus on paths to enhance expressivity, while Tönshoff et al. (2023) leverage random walks for similar purposes.

Positional encodings (PEs) and structural encodings (SEs) have emerged as effective strategies to enhance the expressivity of MPNNs. PEs augment node representations with additional information, such as unique node identifiers (Vignac et al., 2020), random features (Abboud et al., 2021; Sato et al., 2021), or spectral features like eigenvectors (Lim et al., 2022; Maskey et al., 2022b). In contrast, SEs enrich MPNNs by embedding structural information about the graph. Examples of SEs include subgraph counts (Bouritsas et al., 2023) and homomorphism counts (Nguyen & Maehara, 2020; Barceló et al., 2021; Welke et al., 2023; Jin et al., 2024).

While PEs and SEs enhance the expressivity of MPNNs by modifying the initial node features, another line of research focuses on using computational graphs that differ from the input graph. For instance, Dimitrov et al. (2023) and Bause et al. (2023) propose graph transformations that enable MPNNs to achieve universality for specific classes of graphs, such as (outer-)planar graphs. More broadly, Jogl et al. (2024) demonstrate that many expressive GNN variants, including $k$-GNNs and subgraph GNNs, can be *simulated* by applying suitable graph transformations followed by standard message passing.

While incorporating SEs does not increase the forward-pass complexity of MPNNs, it introduces significant preprocessing overhead. For instance, computing homomorphism counts for graphs with treewidth $k$ requires $\mathcal{O}(N^k)$ operations, where $N$ is the number of nodes. This preprocessing complexity becomes exponential in $k$, making it computationally prohibitive for achieving expressivity beyond $k$-WL.

Most prior work evaluates GNN expressivity using the $k$-WL hierarchy, which provides a qualitative measure of distinguishing power but does not quantify the specific substructures a GNN can encode.

To address this gap, Zhang et al. (2024) proposed homomorphism counts as a quantitative measure of expressivity. Building on the work of Lovász (1967), they demonstrated that homomorphism counts are a complete graph invariant, meaning that two graphs are isomorphic if and only if their homomorphism counts are identical. Tinhofer (1986; 1991) showed that 1-WL is equivalent to counting homomorphisms from graphs with treewidth one, while Dell et al. (2018) extended this to prove that $k$-WL corresponds to counting homomorphisms from graphs with treewidth $k$.

Recent studies have further explored the relationship between homomorphism counts and GNN expressivity. For example, Barceló et al. (2021) showed that MPNNs augmented with homomorphism counts as initial node features can count homomorphisms of trees augmented with the included patterns. Similarly, Paolino et al. (2024) demonstrated that MPNNs enriched with specific path information can count homomorphisms of cactus graphs.

Recent work by Kemper et al. (2025) defines and studies a probabilistic version of the WL test, which allows them to draw similar conclusions to our work, in that model expressivity should not be blindly maximized, but rather carefully chosen to suit a given task and dataset.

## A.2 Generalization Bounds for GNNs

The generalization capabilities of Graph Neural Networks (GNNs) have been studied from various theoretical perspectives. Scarselli et al. (2018) provided an early understanding of GNN capacity by deriving generalization bounds for implicitly defined GNNs based on their VC-dimension. Building on this, Du et al. (2019) analyzed the generalization behavior of GNNs in the infinite-width limit using the Graph Neural Tangent Kernel (GNTK), offering insights into their asymptotic performance as network width grows unbounded.

Focusing on data-dependent approaches, Garg et al. (2020) and Liao et al. (2021) investigated the generalization properties of specific MPNNs with sum aggregation. By employing Rademacher complexity and PAC-Bayes methods, they established bounds that depend on the observed training data, shedding light on how factors like data distribution and architectural choices influence generalization.

Levie (2024) introduced the graphon-signal cut distance, a metric for measuring similarity between graph-signal distributions, and demonstrated that MPNNs are Lipschitz-continuous with respect to this distance. This insight enabled the derivation of generalization bounds for MPNNs in the context of arbitrary graph-signal distributions. However, these bounds exhibit a slow convergence rate of $O(1/\log\log(\sqrt{m}))$, where $m$ is the number of training graphs. This slow rate arises from the generality of their assumptions, which accommodate highly flexible graph-signal distributions.

The connection between GNN expressivity and generalization was further explored by Morris et al. (2023a), who showed that the number of graphs distinguishable by the 1-WL test is directly linked to the VC-dimension of GNNs. This result highlights the role of the Weisfeiler-Lehman hierarchy in understanding both the combinatorial expressivity and theoretical capacity of GNNs. In the restricted setting of linear separability, margin-based bounds have been proposed to partially bridge theory and practice (Franks et al., 2024), yet our broader understanding of how expressivity influences generalization remains incomplete.

Li et al. (2025) provide a novel perspective on the generalization behavior of graph neural networks by decoupling the representation learning component from the classification step. Their framework considers fixed graph encoders—such as MPNNs, $k$-WL, or homomorphism-based models—which map graphs into an embedding space. A separate, typically parametric, classifier (e.g., a softmax-based MLP) is then applied to these embeddings. This setting allows for a focused study of the generalization ability of classifiers conditioned on precomputed graph representations, shifting attention away from the learning dynamics of the GNN and toward the geometry and concentration properties of the induced embedding distributions.

Their analysis formalizes how generalization depends on two key factors: intra-class concentration, i.e., how tightly embeddings from the same class cluster, and inter-class separation, i.e., how well embeddings of different classes are separated. These are quantified using the 1-Wasserstein distance between class-conditional embedding distributions. The main theoretical result establishes a generalization bound on the classifier's margin loss, showing it can be upper-bounded in terms of these geometric quantities. Importantly, the bound incorporates the expressivity of the graph encoder through a Lipschitz constant that quantifies how much the embedding distribution of a more

expressive encoder (bounding in distinguishing power) can distort the geometry of a less expressive one (that is used for calculating the graph embeddings). This captures how changes in expressivity influence intra-class concentration and inter-class separation, and thus directly affect generalization.

Although the results elegantly characterize the trade-off between expressivity and generalization, a central limitation is that the graph encoders are *fixed*. They are fixed feature extractors, often derived from CRAs or fixed GNN architectures. As such, the framework does not directly reflect the behavior of trainable GNNs in practical deep learning pipelines, where representation learning and classifier fitting are tightly coupled. Additionally, in contrast to our correlation-based analysis, the generalization bounds in Li et al. (2025) require the existence of a fixed positive margin, which is a strong assumption that may not hold in practice. Our framework is more general, as it does not rely on margin separability and can quantify generalization even in the presence of label noise or overlapping classes. Nonetheless, the insights are valuable: they reveal conditions under which more expressive encoders can improve generalization—specifically, when expressivity increases intra-class concentration without excessively harming inter-class separation. In this way, the paper offers a principled theoretical basis for interpreting empirical phenomena observed in GNN performance across datasets and model classes.

Related to our work, Maskey et al. (2022a; 2025) analyzed scenarios where graph labels are correlated with random graph models, based on graphons. They demonstrated that MPNNs generalize better as the size of the sampled graphs increases, since the statistical properties of larger graphs more closely approximate those of the underlying random graph models.

The approach by Maskey et al. (2022a; 2025) is limited because it assumes labels are linked to random graph models, where many specific assumptions are made about the underlying graphon governing the data. These assumptions may not hold in practical scenarios, making their results less general and potentially less applicable to real-world tasks. In contrast, our framework accommodates arbitrary correlations between graph labels and structural features, as long as they can be described by a Lipschitz-continuous distribution. This broader scope makes our method suitable for analyzing a wide variety of graph datasets, including those where the graph generation process is not well understood or where random graph model assumptions are too restrictive.

The work most closely related to ours is (Ma et al., 2021), which studies generalization in a semi-supervised node classification setting. Their analysis considers a scenario where node labels are correlated with features derived from the node's local neighborhood and its attributes. Using a PAC-Bayes approach that heavily inspired our work, they show that generalization improves when the extracted features are similar between the training and test sets. However, their framework is limited by the assumption of a fixed, non-learnable graph encoder, and their results do not generalize to multi-graph settings or models with learnable parameters.

In contrast, our approach provides a general framework to analyze GNN generalization in settings where graph labels are correlated with structural features. This is achieved by introducing a pseudometric, such as the Tree Mover's Distance, which captures structural differences between graphs. Unlike prior works, our framework does not rely on restrictive assumptions about the underlying graph distribution. Instead, we assume only that the labels are generated by a Lipschitz-continuous probability distribution with respect to the pseudometric. This allows us to analyze a broad range of tasks where the graph structure plays a critical role in determining labels. By explicitly connecting generalization bounds to structural alignment between training and test graphs, our framework offers a flexible and robust method to study GNN performance in diverse applications.

Our framework overcomes these limitations by supporting learnable GNN architectures and extending naturally to multi-graph settings. This allows for a broader analysis of generalization, including GNNs beyond 1-WL expressivity. Moreover, our approach bridges the gap between theory and practice by providing insights into how the interplay between model capacity, structural similarity, and feature alignment affects performance.

## B  SIMULATABLE COLOR REFINE ALGORITHMS

It is possible to represent many different GNNs as MPNNs without loss in expressivity. For this, Jogl et al. (2024) developed the concept of *simulation*. Intuitively, a GNN with $t > 0$ layers can be simulated if we can map its input domain to the set of graphs and achieve the same expressivity

with a $t$-layer MPNN on this adapted set of graphs. To generalize to different types of GNNs, one represents GNNs $\phi$ as color refinement functions that iteratively refine a coloring on some relational structure $X$.

**Definition B.1.** Let $\phi$ be a color update function. Let $R$ be a mapping from the domain of $\phi$ to the set of graphs.[1] We consider two arbitrary relational structures from the domain of $\phi$, say $X$ and $X'$. We say $\phi$ can be *strongly simulated* under $R$ if for every $t \geq 0$ it holds that $\text{WL}^t(R(X)) = \text{WL}^t(R(X'))$ implies that $\phi^t(X) = \phi^t(X')$.

As an example, consider 3-WL from Section C.3. It can be seen as a color update function $c$ that operates on a set of labeled 3-tuples, i.e., in iteration $t > 0$ it refines the coloring of tuple $(v_1, v_2, v_3) \in V^3$ by utilizing the coloring $c^{(t-1)}$:

$$c_{(v_1,v_2,v_3)}^{(t)} = \text{HASH}\left(c_{(v_1,v_2,v_3)}^{(t-1)}, \left(C_1^{(t)}((v_1,v_2,v_3)), \ldots, C_3^{(t)}((v_1,v_2,v_3))\right)\right),$$

where

$$C_1^{(t)}((v_1,v_2,v_3)) = \text{HASH}\left(\{\{c_{(w,v_2,v_3)}^{(t-1)} \mid w \in V\}\}\right),$$

$$C_2^{(t)}((v_1,v_2,v_3)) = \text{HASH}\left(\{\{c_{(v_1,w,v_3)}^{(t-1)} \mid w \in V\}\}\right),$$

$$C_3^{(t)}((v_1,v_2,v_3)) = \text{HASH}\left(\{\{c_{(v_1,v_2,w)}^{(t-1)} \mid w \in V\}\}\right).$$

Instead of using 3-WL, we can create the graph $G^{\otimes 3}$ with vertices $(v_1, v_2, v_3) \in V^3$ and three types of edges

$$E_1 = \cup_{v_1,v_2,v_3 \in V}\{\{(v_1,v_2,v_3),(w,v_2,v_3)\} \mid w \in V\},$$
$$E_2 = \cup_{v_1,v_2,v_3 \in V}\{\{(v_1,v_2,v_3),(v_1,w,v_3)\} \mid w \in V\},$$
$$E_3 = \cup_{v_1,v_2,v_3 \in V}\{\{(v_1,v_2,v_3),(v_1,v_2,w)\} \mid w \in V\}.$$

Observe, that for a tuple $(v_1, v_2, v_3)$ the neighborhood $E_j$ contains exactly those tuples as aggregated in the definition of $C_j$. We can merge these three edge sets $E_1, E_2, E_3$ into a single edge set $E$ by using edge features that encode from which of the three sets the edge originates. Such a transformation $R$ allows for the strong simulation of 3-WL.

### B.1 OTHER STRONGLY-SIMULATABLE ARCHITECTURES

A GNN/WL variant is *strongly simulated* when there exists a structure-to-graph encoding $R$ such that applying $R$ to the input and running a depth–$t$ MPNN reproduces—*layer by layer*—the colors (or hidden states) produced by the original depth–$t$ model (Definition B.1). Table 3 summarizes some architectures that admit such a simulation and sketches the key transformation behind $R$; full proofs are in (Jogl et al., 2024).

## C  TREE MOVER'S DISTANCE

We summarize some important results from Chuang & Jegelka (2022) and Davidson & Dym (2025). We first start with the definition of the tree mover's distance which provides us with a tool to compare two graphs based on their computational trees quantitatively.

### C.1 TREE MOVER'S DISTANCE

The definition and notations in this section largely follow (Chuang & Jegelka, 2022).

**Definition C.1.** Let $G = (V, E)$ be a graph. We define the *depth-$T$ computational tree* $T_v^T$ of node $v$ recursively by connecting the neighbors of the leaf nodes of $T_v^{T-1}$ to the tree. We set $T_v^1 := v$ as the single node tree without any edges. The multiset of depth-$T$ computation trees defined by $G$ is denoted by $\mathcal{T}_G^T := \{\{T_v^T\}\}_{v \in V}$. Additionally, for a tree $T$ with root $r$, we denote by $\mathcal{T}_r$ the multiset of subtrees that root at the descendants of $r$.

---

[1] Jogl et al. (2024) introduces some additional restrictions on $R$ that we omit for the sake of simplicity.

Table 3: GNN families whose per-layer updates can be *exactly* reproduced by a 1-WL–equivalent MPNN on the transformed graph $R(G)$.

| Model / Algorithm | Graph transformation $R(G)$ |
|---|---|
| VVC-GNN (Sato et al., 2019) | For a given port ordering, write on each edge direction the port of the source and target node. |
| $k$-WL / $k$-GNN (Morris et al., 2019) | Compute all $k$-tuples of nodes. Create a node for each $k$-tuple and encode the isomorphism class of each tuple as a node feature. Connect tuples differing in one position encoding this postion as an edge feature. |
| $\delta$-$k$-(L)WL / GNN (Morris et al., 2020b) | Similar as the $k$-tuple graph above: when connecting tuples add a local/global flag encoding whether the nodes that differ in the tuple form an edge in the original graph. For $\delta$-$k$-LWL, only connect tuples when these nodes do form an edge. |
| $(k, s)$-LWL / SpeqNet (Morris et al., 2022) | Similar as $\delta$-$k$-WL above: keep only tuple-vertices whose induced subgraph has $\leq s$ components. |
| GSN-e / GSN-v (Bouritsas et al., 2023) | For pre-computed subgraph pattern counts, augment original node and edge features with these counts. |
| DS-WL / DS-GNN (Bevilacqua et al., 2021) | For a given policy $\pi$ to generate subgraphs, create a graph by taking the disjoint union of all extracted subgraphs $\pi(G)$. |
| $k$-OSWL / OSAN (Qian et al., 2022) | For each $k$ tuple of vertices in the graph ("$k$-ordered subgraphs"), create a copy of all vertices in the original graph and use node features to encode the atomic type of that node in this ordered subgraph. In each subgraph, either link all vertices or only neighbors in the original graph. |
| Mk-GNN (Papp & Wattenhofer, 2022) | For a given set of marked nodes, on each edge encode whether the target node is marked or unmarked. |
| GMP (Wijesinghe & Wang, 2022) | Attach structural coefficients as node or edge features. |
| Shortest-Path Nets (Abboud et al., 2022) | For $i \in 1, \ldots, k$ add an edge between every pair of nodes with shortest path distance $i$ and encode $i$ as a feature on that edge. |
| Generalised-Distance WL (Zhang et al., 2023) | For a given distance metric $d$ and graph $G$, add an edge between every pair of nodes $u, v$ and encode the metric $d_G(u, v)$ as a feature on this edge. |

In other words, the depth-$T$ computational tree $T_v^T$ of node $v$ is the 1-WL computational tree of node $v$ after $T - 1$ iterations. To compare two multisets of computational trees we need to *augments trees*.

**Definition C.2.** A *blank tree* $T_\emptyset$ is defined as a tree graph that contains a single node and no edge, where the node feature is the zero vector $\mathbf{0}_p \in \mathbb{R}^p$. We define $T_\emptyset^n$ as the multiset of $n$ blank trees.

**Definition C.3.** Let $\mathcal{T}_v, \mathcal{T}_u$ be two multisets of trees. We define $\rho$ as function that augments a pair of trees with blank trees as follows:

$$\rho : (\mathcal{T}_v, \mathcal{T}_u) \mapsto \left( \mathcal{T}_v \cup T_\emptyset^{\max(|\mathcal{T}_u| - |\mathcal{T}_v|, 0)}, \mathcal{T}_u \cup T_\emptyset^{\max(|\mathcal{T}_v| - |\mathcal{T}_u|, 0)} \right). \tag{5}$$

**Definition C.4.** Let $w : \mathbb{N} \to \mathbb{R}^+$ be a depth-dependent weighting function. For two trees $T_a, T_b$, we define the tree distance $\mathrm{TD}_w(T_a, T_b)$ between $T_a$ and $T_b$ recursively as

$$\mathrm{TD}_w(T_a, T_b) := \begin{cases} \|x_a - x_b\| + w(T) \cdot \mathrm{OT}_{\mathrm{TD}_w}(\rho(\mathcal{T}_a, \mathcal{T}_b)) & \text{if } T > 1 \\ \|x_a - x_b\| & \text{otherwise,} \end{cases} \tag{6}$$

where $T = \max(\mathrm{Depth}(T_a), \mathrm{Depth}(T_b))$.

We note that the optimal transport OT with respect to some metric $d$ between two multisets $\mathbf{x} = \{\{x_1, \ldots, x_n\}\}$ and $\mathbf{y} = \{\{y_1, \ldots, y_n\}\}$ of the same size $n$ is defined via

$$\mathrm{OT}_d(\mathbf{x}, \mathbf{y}) = \min_\sigma \sum_i d\left(x_i, y_{\sigma(i)}\right). \tag{7}$$

**Definition C.5.** Let $G, H \in \mathcal{G}$, $w : \mathbb{N} \to \mathbb{R}^+$, and $T \geq 0$. The *tree mover's distance* between $G$ and $H$ is defined as

$$\mathrm{TMD}_w^T(G, H) = \mathrm{OT}_{\mathrm{TD}_w}\left(\rho(\mathcal{T}_G^T, \mathcal{T}_H^T)\right), \tag{8}$$

where $\mathcal{T}_G^T$ and $\mathcal{T}_H^T$ are multisets of the depth-$T$ computation trees of graphs $G$ and $H$, respectively.

We note that, for simplicity, we omit the weighting function $w$ the corresponding subscript in the definition of $\mathrm{TMD}_w^T$ and simply write $\mathrm{TMD}^T$ in the main part of this manuscript.

## C.2 PROOFS IN SECTION 3

*Proof of Proposition 3.2.* By (Chuang & Jegelka, 2022, Theorem 6), $\text{TMD}_w^t$ is a pseudometric. It is easy to see that $\zeta\text{-TMD}_w^t$ is also a pseudometric. Given $G, H \in \mathcal{G}$, we have

$$
\begin{aligned}
\zeta\text{-TMD}(G, H) &= \text{TMD}(R^\zeta(G), R^\zeta(H)) \\
&= \text{TMD}(R^\zeta(H), R^\zeta(G)) \\
&= \zeta\text{-TMD}(H, G),
\end{aligned}
\tag{9}
$$

showing the symmetry of $\zeta\text{-TMD}_w^t$. □

*Proof of Proposition 3.3.* By (Chuang & Jegelka, 2022, Theorem 7), if two graphs $G'$ and $H'$ are determined to be non-isomorphic in WL iteration $T$ and $w(t) > 0$ for all $0 < t \leq T + 1$, then $\text{TMD}_w^{T+1}(G', H') > 0$.

If $\zeta$ distinguishes $G$ and $H$ after $T$ iterations, then WL determines $R^\zeta(G)$ and $R^\zeta(H)$ to be non-isomorphic, i.e., $\text{TMD}_w^{T+1}(R^\zeta(G), R^\zeta(H)) > 0$. Then,

$$
\zeta\text{-TMD}_w^{T+1}(G, H) > 0.
$$

□

We reformulate and prove a more general version of Theorem 3.4, where we consider general message functions on multisets that are Lipschitz continuous on the multiset domain, and update function that are Lipschitz on the standard Euclidean latent space. This follows the approach in (Davidson & Dym, 2025), where a similar version was proved in Theorem F.2. in their manuscript. However, in contrast, we calculate the explicit constant of the Lipschitz constant which we will need for later proofs.

Theorem 3.4 then follows as a corollary as permutations-invariant aggregation functions composed of a MLP followed by sum aggregation are Lipschitz continuous on multisets.

**Lemma C.6.** *Consider a MPNN of the form*

$$
x_v^{(t)} = f^{(t)}\left(x_v^{(t-1)}, g^{(t)}\left(\{\{x_u^{(t-1)}, \}\}_{u \in \mathcal{N}(v)}\right)\right)
\tag{10}
$$

*with message and update functions $\left(g^{(t)}, f^{(t)}\right)_{t=1}^T$. Consider also a global readout function of the form*

$$
h(G) = c\left(\{\{x_v^{(T)}\}\}_{v \in V(G)}\right).
$$

*Suppose that for all $t = 1, \ldots, T$, the message and update functions are Lipschitz continuous with Lipschitz constants bounded by $L_{g^{(t)}}$ and $L_{f^{(t)}}$, respectively. Suppose that the readout function is Lipschitz continuous with Lipschitz constants bounded by $L_c$. Then, for all layers $t = 0, 1, \ldots, T$ and for all pairs of graphs $G, H \in \mathcal{G}$ and all pairs of nodes $u \in V(G)$ and $v \in V(H)$, we have*

$$
\left\| x_u^{(t)} - x_v^{(t)} \right\| \leq \left(\prod_{t'=1}^t L_{g^{(t')}} L_{f^{(t')}}\right) 2^t TD\left(T_u^{(t+1)}, T_v^{(t+1)}\right).
\tag{11}
$$

*For the global output, we have*

$$
\|h(G) - h(H)\| \leq L_c \left(\prod_{t=1}^T L_{g^{(t)}} L_{f^{(t)}}\right) 2^T TMD^{T+1}(G, H).
\tag{12}
$$

*Proof.* Without loss of generality, suppose that $L_{f^{(t)}}, L_{g^{(t)}} \geq 1$. We show Equation (11) by induction. For $t = 0$ Equation (11) holds trivially.

**Step 1: Bound the difference between message function outputs.**

$$\left\| g^{(t)}\left(\{\{x_s^{(t-1)}\}\}_{s\in\mathcal{N}(u)}\right) - g^{(t)}\left(\{\{x_{s'}^{(t-1)}\}\}_{s'\in\mathcal{N}(u)}\right)\right\|$$

$$\leq L_{g^{(t)}}\mathrm{WD}_{\|\cdot\|}\left(\{\{x_s^{(t-1)}\}\}_{s\in\mathcal{N}(v)}, \{\{x_{s'}^{(t-1)}\}\}_{s'\in\mathcal{N}(u)}\right)$$

$$= L_{g^{(t)}}\min_{\tau\in S_n}\sum_s\left\| x_s^{(t-1)} - x_{\tau(s)}^{(t-1)}\right\|$$

$$\leq L_{g^{(t)}}\sum_s\left\| x_s^{(t-1)} - x_{\tau^*(s)}^{(t-1)}\right\| \tag{13}$$

$$= L_{g^{(t)}}\left(\prod_{t'=1}^{t-1} L_{g^{(t')}}L_{f^{(t')}}\right)2^{t-1}\sum_s\mathrm{TD}\left(T_s^{(t)}, T_{\tau^*(s)}^{(t)}\right)$$

$$= L_{g^{(t)}}\left(\prod_{t'=1}^{t-1} L_{g^{(t')}}L_{f^{(t')}}\right)2^{t-1}\mathrm{WD}_{\mathrm{TD}}\left(\{\{T_s^{(t)}\}\}_{s\in\mathcal{N}(v)}, \{\{T_{s'}^{(t)}\}\}_{s'\in\mathcal{N}(u)}\right),$$

where $\tau^*$ is the optimal permutation in the definition of $\mathrm{WD}_{\mathrm{TD}}\left(\{\{T_s^{(t)}\}\}_{s\in\mathcal{N}(v)}, \{\{T_{s'}^{(t)}\}\}_{s'\in\mathcal{N}(u)}\right)$.

**Step 2: Bound the difference between update function outputs.**

$$\left\| x_u^{(t)} - x_v^{(t)}\right\|$$

$$= \left\| f^{(t)}\left(x_s^{(t-1)}, g^{(t)}\left(\{\{x_s^{(t-1)}\}\}_{s\in\mathcal{N}(u)}\right)\right) - f^{(t)}\left(x_v^{(t-1)}, g^{(t)}\left(\{\{x_{s'}^{(t-1)}\}\}_{s'\in\mathcal{N}(v)}\right)\right)\right\|$$

$$\leq L_{f^{(t)}}\left(\left\| x_u^{(t-1)} - x_v^{(t-1)}\right\| + \left\| g^{(t)}\left(\{\{x_s^{(t-1)}\}\}_{s\in\mathcal{N}(u)}\right) - g^{(t)}\left(\{\{x_{s'}^{(t-1)}\}\}_{s'\in\mathcal{N}(v)}\right)\right\|\right)$$

$$\leq L_{f^{(t)}}\left(\left\| x_u^{(t-1)} - x_v^{(t-1)}\right\| + L_{g^{(t)}}\left(\prod_{t'=1}^{t-1} L_{g^{(t')}}L_{f^{(t')}}\right)2^{t-1}\mathrm{WD}_{\mathrm{TD}}\left(\{\{T_s^{(t)}\}\}_{s\in\mathcal{N}(v)}, \{\{T_{s'}^{(t)}\}\}_{s'\in\mathcal{N}(u)}\right)\right)$$

$$\leq L_{f^{(t)}}\left(\left(\prod_{t'=1}^{t-1} L_{g^{(t')}}L_{f^{(t')}}\right)2^{t-1}\mathrm{TD}\left(T_u^{(t)}, T_v^{(t)}\right)\right.$$

$$+ L_{g^{(t)}}\left(\prod_{t'=1}^{t-1} L_{g^{(t')}}L_{f^{(t')}}\right)2^{t-1}\mathrm{WD}_{\mathrm{TD}}\left(\{\{T_s^{(t)}\}\}_{s\in\mathcal{N}(v)}, \{\{T_{s'}^{(t)}\}\}_{s'\in\mathcal{N}(u)}\right)\right)$$

$$\leq \left(\prod_{t'=1}^{t} L_{g^{(t')}}L_{f^{(t')}}\right)2^t\mathrm{TD}\left(T_v^{(t+1)}, T_u^{(t+1)}\right).$$

$$\tag{14}$$

This finishes the proof of the first claim.

**Step 3: Bound the different between global outputs.** The second claim follows by calculating,

$$\|h(G) - h(H)\| = \left\| c\left(\{\{x_u^{(T)}\}\}_{u\in V(G)}\right) - c\left(\{\{x_v^{(T)}\}\}_{v\in V(G)}\right)\right\|$$

$$\leq L_c\mathrm{WD}_{\|\cdot\|}(\{\{x_u^{(T)}\}\}_{u\in V(G)}, \{\{x_v^{(T)}\}\}_{v\in V(G)})$$

$$= L_c\min_\tau\sum_s\|x_s^{(T)} - x_{\tau(s)}^{(T)}\| \tag{15}$$

$$\leq L_c\left(\prod_{t'=1}^{T} L_{g^{(t')}}L_{f^{(t')}}\right)2^T\sum_s\mathrm{TD}\left(T_s^{(T+1)}, T_{\tau^*(s)}^{(T+1)}\right)$$

$$= L_c\left(\prod_{t'=1}^{T} L_{g^{(t')}}L_{f^{(t')}}\right)2^T\mathrm{TMD}^{T+1}(G, H),$$

where $\tau^*$ is the optimal permutation in the definition of $\mathrm{TMD}^{T+1}(G, H)$. $\qquad\square$

We now consider the case where Lipschitz continuous functions on multisets are implemented via sum aggregation, followed by Lipschitz continuous functions on the Euclidean domain. This includes scenarios where the message, update, and readout functions are MLPs with Lipschitz continuous activation functions, such as ReLU.

**Corollary C.7.** *Consider a MPNN of the form*

$$x_v^{(t+1)} = f^{(t+1)} \left( x_v^{(t)}, \sum_{u \in \mathcal{N}(v)} g^{(t+1)} \left( x_u^{(t)} \right) \right),$$

*with message and update functions* $\left( g^{(t)}, f^{(t)} \right)_{t=1}^T$. *Consider also a global readout function* $e$ *of the form*

$$h(G) = e \left( \sum_{v \in V(G)} x_v^{(T)} \right).$$

*Suppose that for all* $t = 1, \dots, T$, *the message and update functions are Lipschitz continuous with Lipschitz constants bounded by* $L_{g^{(t)}}$ *and* $L_{f^{(t)}}$, *respectively. Suppose that the readout function is Lipschitz continuous with Lipschitz constants bounded by* $L_c$. *Then, for all layers* $t = 0, 1, \dots, T$ *and for all pairs of graphs* $G, H \in \mathcal{G}$ *and all pairs of nodes* $u \in V(G)$ *and* $v \in V(H)$, *we have*

$$\left\| x_u^{(t)} - x_v^{(t)} \right\| \leq \left( \prod_{t'=1}^t L_{g^{(t')}} L_{f^{(t')}} \right) 2^t TD \left( T_u^{(t+1)}, T_v^{(t+1)} \right). \tag{16}$$

*For the global output, we have*

$$\| h(G) - h(G) \| \leq L_c \left( \prod_{t=1}^T L_{g^{(t)}} L_{f^{(t)}} \right) 2^T TMD^{T+1} (G, H).$$

*Proof.* This follows directly from Lemma C.6, since for any Lipschitz continuous function $f$ with Lipschitz constant $L_f$, the induced multiset function $\tilde{f} : X \mapsto \sum_{x \in X} f(x)$ remains Lipschitz continuous with a Lipschitz constant bounded by $L_f$. $\square$

**Theorem C.8.** *Let* $\zeta$ *be a strongly simulatable CRA, and let* $h : \mathcal{G} \to \mathbb{R}^K$ *be a* $\zeta$-*MPNN with* $T$ *layers, where the message and update functions are Lipschitz continuous with Lipschitz constants bounded by* $L_{g^{(t)}}$ *and* $L_{f^{(t)}}$, *respectively. Suppose* $h$ *includes a global sum pooling layer followed by a Lipschitz continuous classifier* $c$ *with Lipschitz constant* $L_c$. *Then, for any graphs* $G$ *and* $H$,

$$\| h(G) - h(H) \| \leq L \cdot \zeta\text{-}TMD^{T+1}(G, H),$$

*where* $L = L_c 2^T \prod_{t=1}^T L_{f^{(t)}} L_{g^{(t)}}$.

*Proof.* Let $h'$ be the underlying MPNN of $h$. The proof of theorem follows from Corollary C.7 by

$$\| h(G) - h(G) \| = \left\| h' \left( R^\zeta(G) \right) - h' \left( R^\zeta(H) \right) \right\|$$

$$\leq L_c \left( \prod_{t=1}^T L_{g^{(t)}} L_{f^{(t)}} \right) 2^T TMD^{T+1} \left( R^\zeta(G), R^\zeta(H) \right)$$

$$= L_c \left( \prod_{t=1}^T L_{g^{(t)}} L_{f^{(t)}} \right) 2^T \zeta\text{-}TMD^{T+1} (G, H).$$

$\square$

## C.3 EXAMPLE II: $k$-GNNS

The *$k$-Weisfeiler-Leman ($k$-WL) test* enhances the expressive power of the 1-WL test by considering interactions between $k$-tuples of nodes. To strongly simulate $k$-WL, a *product graph* $G^{\otimes k}$ is constructed, where each node represents a $k$-tuple of nodes from the original graph $G$, and edges are defined based on the adjacency relationships in $G$.

Similarly, *$k$-MPNNs* operate directly on the product graph $G^{\otimes k}$, achieving the same level of expressivity as the $k$-WL.

**Corollary C.9.** *Let $h$ be an $k$-MPNN with $T$ layers. Then, there exists a constant $L$ such that for any graphs $G$ and $H$,*

$$\|h(G) - h(H)\| \leq L \cdot k\text{-}TMD^{T+1}(G, H). \tag{17}$$

## D  GRAPH CLASSIFICATION SETTING

We consider a classification problem with $K$ classes over a fixed set of training graphs $\mathcal{G}_{\text{tr}}$ and test graphs $\mathcal{G}_{\text{te}}$. Each graph $G$ is equipped with node features $x$, and we assume there exists a constant $B > 0$ such that $\|x_i\| \leq B$ for every node $i \in V(G)$ and every $G \in \mathcal{G}_{\text{tr}} \cup \mathcal{G}_{\text{te}}$. Let $N_{\text{tr}} = |\mathcal{G}_{\text{tr}}|$ and $N_{\text{te}} = |\mathcal{G}_{\text{te}}|$. Each graph $G$ has a label $y$ sampled from an unknown distribution $p$. Moreover, we assume that graphs closer in a given pseudometric $\text{pm}$ are more likely to share the same label. Concretely, for each class $k \in \{1, \ldots, K\}$, there is a Lipschitz continuous function $\eta_k$ (with respect to some pseudometric $\text{pm}$) such that

$$\Pr(y = k \mid G) = \eta_k(G),$$

and we denote by $C := \max_k \text{Lip}(\eta_k)$ the maximum Lipschitz constant among all $\{\eta_k\}$. Let $\xi$ be an upper bound on the distance (with respect to $\text{pm}$) between any training graph in $\mathcal{G}_{\text{tr}}$ and any test graph in $\mathcal{G}_{\text{te}}$.

We now introduce a PAC-Bayes framework to derive a generalization bound for classifiers under this setting. We consider two different setups for learning on graphs:

1. *$\zeta$-MPNNs:* Let $\zeta$ be a strongly simulatable color-refinement algorithm (see Definition B.1). A $\zeta$-MPNN has a fixed depth $T$ and a final MLP $e$ as the classifier. Each layer $t \in \{1, \ldots, T\}$ has message and update functions denoted by $g^{(t)}$ and $f^{(t)}$, each with fixed depths $d(g^{(t)})$ and $d(f^{(t)})$, respectively. The corresponding weight matrices are written as $\{w(g^{(t)}, s)\}_{s=1}^{d(g^{(t)})}$ and $\{w(f^{(t)}, s)\}_{s=1}^{d(f^{(t)})}$. The final MLP classifier $e$ has parameters $\{w(e, s)\}_{s=1}^{d(e)}$. Define $b$ to be the maximum hidden dimension across these modules, and let $\mathcal{H}_\zeta$ be the class of all such $\zeta$-MPNN classifiers.

2. *MLPs on non-learnable graph embeddings:* Here, the final MLP classifier $c$ has the same notations for its parameters as above, and we denote the hypothesis set by $\mathcal{H}_{\text{latent}}$.

We assume that all non-linearities in the MLPs are Lipschitz continuous and homogeneous.

Finally, we adapt the following assumption from Ma et al. (2021).

**Assumption D.1** (Assumption on Concentrated Expected Loss Difference.)**.** Let $P$ be a distribution over $\mathcal{H}$ obtained by sampling $D$ (vectorized) trainable weight matrices from $\mathcal{N}(0, \sigma^2 I)$, with

$$\sigma^2 \leq \frac{\left(\gamma/(8\xi)\right)^{2/D}}{2\,b\left(\lambda\,N_{\text{tr}}^{-\alpha} + \ln(2\,b\,D)\right)}.$$

For any classifier $h \in \mathcal{H}$ with model parameters $\{w_j\}_j$, define $T_h := \max_j \|w_j\|_2$. Assume there exists an $0 < \alpha < \frac{1}{4}$ such that

$$\Pr_{h \sim P}\left(\mathcal{L}_{\text{te}}^{\gamma/4}(h) - \mathcal{L}_{\text{tr}}^{\gamma/2}(h) > N_{\text{tr}}^{-\alpha} + c\,K\,\xi \;\middle|\; T_h^D\,\xi > \frac{\gamma}{8}\right) \leq e^{-N_{\text{tr}}^{2\alpha}}.$$

This assumption posits that when the model parameters $h$ sampled from $P$ do not exceed a certain norm threshold (i.e., $T_h^D \xi \leq \frac{\gamma}{8}$) and the number of training samples $N_{\text{tr}}$ is sufficiently large, then the expected margin loss on the test set will not exceed that on the training set by more than $N_{\text{tr}}^{-\alpha} + cK\xi$. Informally, once the training set grows large enough (and model weights are not too large), the test performance cannot deviate significantly from the training performance. This property becomes trivially true if all samples in $\mathcal{G}_{\text{tr}}$ and $\mathcal{G}_{\text{te}}$ are i.i.d. since $\mathcal{L}_{\text{te}}^{\gamma/2} - \mathcal{L}_{\text{tr}}^{\gamma/2} \leq 0$ in that case.

## E  PROOFS OF THE RESULTS SECTION 4

In this section, we provide detailed proofs of the theoretical results presented in the Section 4.

We note that the proof of Theorem 4.1 follows the proof of Theorem 3 in (Ma et al., 2021), carefully adapted for graph classification tasks and learnable graph encoders.

Specifically, the proof leverages the PAC-Bayes framework, properties of the chosen pseudometric, and structural constraints on the hypothesis space. For clarity, we decompose it into a series of steps, each contributing to the final result.

In Appendix E.1, we present intermediate results that are independent of the choice of pseudometric and hypothesis space. To emphasize this generality, we denote the pseudometric by $\mathrm{pm}$ and the hypothesis space by $\mathcal{H}$. This ensures that the results in Appendix E.1 apply regardless of whether the pseudometric is defined in the graph space or latent space, and whether the classifier is an end-to-end learnable GNN or a deterministic graph encoder followed by a learnable MLP classifier. This approach allows us to establish in Appendix E.2 the generalization bound for end-to-end learnable GNNs, and in the following subsection, the bound for graph classifiers with fixed encoders–both derived from the results in Appendix E.1.

We note that if $\mathrm{pm}$ is defined on $\mathcal{G}$, the corresponding classifier $h : \mathcal{G} \to \{1, \ldots, K\}$ is assumed to be Lipschitz with respect to $\mathrm{pm}$. If instead $\mathrm{pm}$ is defined on $\mathbb{R}^b$, the classifier $h : \mathbb{R}^b \to \{1, \ldots, K\}$, typically an MLP, is assumed to be Lipschitz with respect to $\mathrm{pm}$. In the latter case, classification is performed via $h \circ e(G)$, where $e : \mathcal{G} \to \mathbb{R}^b$ is a deterministic graph embedding network.

### E.1 PRELIMINARIES FOR THE PROOFS IN SECTION 4

**Step 1: Deterministic Bound via PAC-Bayes.** We begin by establishing a deterministic bound on the test loss in terms of the training loss and the Kullback-Leibler (KL) divergence between the posterior and prior distributions over the hypothesis space $\mathcal{H}$.

**Lemma E.1.** *Let $\tilde{h} \in \mathcal{H}$ be any classifier and let $P$ be a prior distribution on $\mathcal{H}$ independent of the training data. For any $\lambda > 0$ and $\gamma \geq 0$, with probability at least $1 - \delta$ over the training sample $y_{\mathrm{tr}}$, for any posterior distribution $Q$ on $\mathcal{H}$ satisfying*

$$\mathbb{P}_{h \sim Q} \left( \max_{G \in \mathcal{G}_{\mathrm{tr}} \cup \mathcal{G}_{\mathrm{te}}} \|h(G) - \tilde{h}(G)\|_\infty < \frac{\gamma}{8} \right) \geq \frac{1}{2},$$

*the following bound holds:*

$$\mathcal{L}^0_{\mathrm{te}}(\tilde{h}) \leq \mathcal{L}^\gamma_{\mathrm{tr}}(\tilde{h}) + \frac{1}{\lambda} \left( 2 \left( D_{\mathrm{KL}}(Q\|P) + 1 \right) + \ln \frac{1}{\delta} + \frac{\lambda^2}{4N_{\mathrm{tr}}} + D^{\gamma/2}_{\mathrm{te,tr}}(P; \lambda) \right), \quad (18)$$

*where $D^{\gamma/2}_{\mathrm{te,tr}}(P; \lambda) = \ln \mathbb{E}_{h \sim P} \exp \left( \lambda \left( \mathcal{L}^{\gamma/2}_{\mathrm{te}}(h) - \mathcal{L}^\gamma_{\mathrm{tr}}(h) \right) \right)$.*

*Proof of Lemma E.1.* The proof adapts Theorem 2 from Ma et al. (2021) from the node to the graph classification setting. We present the proof for completeness.

First, define a subset $\mathcal{H}_{\tilde{h}} \subset \mathcal{H}$ as:

$$\mathcal{H}_{\tilde{h}} = \left\{ h \in \mathcal{H} \mid \max_{G \in \mathcal{G}_{\mathrm{tr}} \cup \mathcal{G}_{\mathrm{te}}} \|h(G) - \tilde{h}(G)\|_\infty \leq \frac{\gamma}{8} \right\}. \quad (19)$$

Using this subset $\mathcal{H}_{\tilde{h}}$, define the modified distribution $Q'$ over $\mathcal{H}_{\tilde{h}}$ as:

$$Q'(h) = \begin{cases} \frac{1}{\mathbb{P}_{h \sim Q}\left(h \in \mathcal{H}_{\tilde{h}}\right)} Q(h), & \text{if } h \in \mathcal{H}_{\tilde{h}}, \\ 0, & \text{otherwise.} \end{cases} \quad (20)$$

We aim to show

$$\mathcal{L}^0_{\mathrm{te}}(\tilde{h}) \leq \mathcal{L}^{\gamma/4}_{\mathrm{te}}(h) \quad \text{and} \quad \hat{\mathcal{L}}^{\gamma/2}_{\mathrm{tr}}(h) \leq \hat{\mathcal{L}}^\gamma_{\mathrm{tr}}(\tilde{h}). \quad (21)$$

The first inequality holds since

$$\mathcal{L}^0_{\mathrm{te}}(\tilde{h}) - \mathcal{L}^{\gamma/4}_{\mathrm{te}}(h)$$

$$= \mathbb{E}_{y_i \sim \mathrm{Pr}(y|G_i), G_i \in \mathcal{G}_{\mathrm{te}}} \left[ \hat{\mathcal{L}}^0_{\mathrm{te}}(\tilde{h}) \right] - \mathbb{E}_{y_i \sim \mathrm{Pr}(y|g_i), G_i \in \mathcal{G}_{\mathrm{te}}} \left[ \hat{\mathcal{L}}^{\gamma/4}_{\mathrm{te}}(h) \right]$$

$$= \mathbb{E}_{y_i \sim \Pr(y|G_i), G_i \in \mathcal{G}_{\text{te}}} \left[ \frac{1}{N_{\text{te}}} \sum_{G_i \in \mathcal{G}_{\text{te}}} \mathbb{1} \left[ \tilde{h}(G_i)[y_i] \leq \left( 0 + \max_{k \neq y_{G_i}} \tilde{h}(G_i)[k] \right) \right] \right]$$

$$- \mathbb{E}_{y_i \sim \Pr(y|G_i), G_i \in \mathcal{G}_{\text{te}}} \left[ \frac{1}{N_{\text{te}}} \sum_{G_i \in \mathcal{G}_{\text{te}}} \mathbb{1} \left[ h(G_i)[y_i] \leq \left( \gamma/4 + \max_{k \neq y_{G_i}} h(G_i)[k] \right) \right] \right]$$

$$= \mathbb{E}_{y_i \sim \Pr(y|G_i), G_i \in \mathcal{G}_{\text{te}}} \left[ \frac{1}{N_{\text{te}}} \sum_{G_i \in \mathcal{G}_{\text{te}}} \mathbb{1} \left[ \tilde{h}(G_i)[y_i] \leq \left( 0 + \max_{k \neq y_{G_i}} \tilde{h}(G_i)[k] \right) \right] \right]$$

$$- \mathbb{1} \left[ h(G_i)[y_i] \leq \left( \gamma/4 + \max_{k \neq y_{G_i}} h(G_i)[k] \right) \right] \Bigg] ,$$

i.e., it remains to show that if $\tilde{h}(G_i)[y_i] \leq \left( 0 + \max_{k \neq y_{G_i}} \tilde{h}(G_i)[k] \right)$ holds, then also $h(G_i)[y_i] \leq \left( \gamma/4 + \max_{k \neq y_{G_i}} h(G_i)[k] \right)$. This is clear by Equation (19),

$$h(G_i)[y_i] \leq \gamma/8 + \tilde{h}(G_i)[y_i]$$
$$\leq \gamma/8 + \max_{k \neq y_{G_i}} \tilde{h}(G_i)[k]$$
$$\leq \gamma/4 + \max_{k \neq y_{G_i}} h(G_i)[k].$$

Similarly, one can prove the second inequality in Equation (21).

Therefore, with probability at least $1 - \delta$ over the samples $y_{\text{tr}}$, we get

$$\mathcal{L}_{\text{te}}^0(\tilde{h}) \leq \mathbb{E}_{h \sim Q'} \hat{\mathcal{L}}_{\text{te}}^{\gamma/4}(h)$$
$$\leq \mathbb{E}_{h \sim Q'} \hat{\mathcal{L}}_{\text{tr}}^{\gamma/2}(h) + \frac{1}{\lambda} \left( D_{\text{KL}}(Q'||P) + \ln \frac{1}{\delta} + \frac{\lambda^2}{4N_{\text{tr}}} + D_{\text{te,tr}}^{\gamma/2}(P; \lambda) \right)$$
$$\leq \mathcal{L}_{\text{tr}}^{\gamma}(\tilde{h}) + \frac{1}{\lambda} \left( D_{\text{KL}}(Q'||P) + \ln \frac{1}{\delta} + \frac{\lambda^2}{4N_{\text{tr}}} + D_{\text{te,tr}}^{\gamma/2}(P; \lambda) \right).$$

The second inequality applies Theorem 1 from Ma et al. (2021), while the first and last inequalities follow directly from (21) and the definitions of $\mathcal{H}_{\tilde{h}}$ and $Q'$. The remaining steps align with the proof of Theorem 2 from Ma et al. (2021). □

**Step 2: Bounding the discrepancy term.** Next in Step 3, we will bound the term $D_{\text{te,tr}}^{\gamma/2}(P; \lambda)$ from Equation (18) in Lemma E.1, which captures the difference in expected losses between the test and training distributions under the prior $P$. We begin by bounding $\mathcal{L}_{\text{te}}^{\gamma/2}(h) - \mathcal{L}_{\text{tr}}^{\gamma}(h)$ in this step.

**Lemma E.2.** *Let $h \in \mathcal{H}$ be any classifier with Lipschitz constant $L$ with respect to the pseudometric* pm. *For any $\gamma \geq 0$, if $L\xi \leq \frac{\gamma}{4}$, then*

$$\mathcal{L}_{\text{te}}^{\gamma/2}(h) - \mathcal{L}_{\text{tr}}^{\gamma}(h) \leq CK\xi,$$

*where $K$ is the number of classes and $C$ is the maximum Lipschitz constant of the functions $\eta_k$.*

*Proof of Lemma E.2.* We set

$$\mathcal{L}^{\gamma}(G, y) := \mathbb{1} \left[ h(G)[y] \leq \gamma + \max_{k \neq y} h(G)[k] \right],$$

where $h(G)[k]$ denotes the output score for class $k$.

Then, we can write

$$\mathcal{L}_{\text{te}}^{\gamma/2}(h) - \mathcal{L}_{\text{tr}}^{\gamma}(h)$$

$$= \mathbb{E}_{y^{\text{te}}} \left[ \sum_{G_j \in \mathcal{G}_{\text{te}}} \frac{1}{N_{\text{te}}} \mathcal{L}^{\gamma/2}(G_j, y_j) \right] - \mathbb{E}_{y^{\text{tr}}} \left[ \sum_{G_i \in \mathcal{G}_{\text{tr}}} \frac{1}{N_{\text{tr}}} \mathcal{L}^{\gamma}(G_i, y_i) \right]$$

$$= \frac{1}{N_{\text{te}}} \sum_{G_j \in \mathcal{G}_{\text{te}}} \sum_{k=1}^{K} \eta_k(G_j) \mathcal{L}^{\gamma/2}(G_j, k) - \frac{1}{N_{\text{tr}}} \sum_{G_i \in \mathcal{G}_{\text{tr}}} \sum_{k=1}^{K} \eta_k(G_i) \mathcal{L}^{\gamma}(G_i, k)$$

$$= \frac{1}{N_{\text{te}}} \frac{1}{N_{\text{tr}}} \sum_{G_i \in \mathcal{G}_{\text{tr}}} \sum_{G_j \in \mathcal{G}_{\text{te}}} \sum_{k=1}^{K} \eta_k(G_j) \mathcal{L}^{\gamma/2}(G_j, k) - \frac{1}{N_{\text{tr}}} \frac{1}{N_{\text{te}}} \sum_{G_j \in \mathcal{G}_{\text{te}}} \sum_{G_i \in \mathcal{G}_{\text{tr}}} \sum_{k=1}^{K} \eta_k(G_i) \mathcal{L}^{\gamma}(G_i, k)$$

$$= \frac{1}{N_{\text{te}} N_{\text{tr}}} \sum_{G_i \in \mathcal{G}_{\text{tr}}} \sum_{G_j \in \mathcal{G}_{\text{te}}} \sum_{k=1}^{K} \left( \eta_k(G_j) \mathcal{L}^{\gamma/2}(G_j, k) - \eta_k(G_i) \mathcal{L}^{\gamma}(G_i, k) \right)$$

$$= \frac{1}{N_{\text{te}} N_{\text{tr}}} \sum_{G_i \in \mathcal{G}_{\text{tr}}} \sum_{G_j \in \mathcal{G}_{\text{te}}} \sum_{k=1}^{K} \left( \eta_k(G_j) \mathcal{L}^{\gamma/2}(G_j, k) - \eta_k(G_j) \mathcal{L}^{\gamma}(G_i, k) \right)$$
$$+ \left( \eta_k(G_j) \mathcal{L}^{\gamma}(G_i, k) - \eta_k(G_i) \mathcal{L}^{\gamma}(G_i, k) \right)$$

$$= \frac{1}{N_{\text{te}} N_{\text{tr}}} \sum_{G_i \in \mathcal{G}_{\text{tr}}} \sum_{G_j \in \mathcal{G}_{\text{te}}} \sum_{k=1}^{K} \eta_k(G_j) \left( \mathcal{L}^{\gamma/2}(G_j, k) - \mathcal{L}^{\gamma}(G_i, k) \right) + \mathcal{L}^{\gamma}(G_i, k) \left( \eta_k(G_j) - \eta_k(G_i) \right)$$

$$\leq \frac{1}{N_{\text{te}} N_{\text{tr}}} \sum_{G_i \in \mathcal{G}_{\text{tr}}} \sum_{G_j \in \mathcal{G}_{\text{te}}} \sum_{k=1}^{K} \left( \mathcal{L}^{\gamma/2}(G_j, k) - \mathcal{L}^{\gamma}(G_i, k) \right) + \left( \eta_k(G_j) - \eta_k(G_i) \right).$$

$$(22)$$

The last inequality holds since $\mathcal{L}^{\gamma}$ and $\eta_k$ are bounded by 1. We have, by assumption on the probability distribution,

$$\eta_k(G_j) - \eta_k(G_i) \leq C \cdot \text{pm}(G_j, G_i) \leq C\xi. \tag{23}$$

Furthermore, by assumption,

$$\|h(G_i) - h(G_j)\|_{\infty} \leq L \cdot \text{pm}(G_j, G_i) \leq L\xi \leq \frac{\gamma}{4}. \tag{24}$$

Then, for any $k = 1, \ldots, K$,

$$\mathcal{L}^{\gamma/2}(h(G_j), k) \leq \mathcal{L}^{\gamma}(h(G_i), k). \tag{25}$$

This is true since both $\mathcal{L}^{\gamma/2}(h(G_j), k)$ and $\mathcal{L}^{\gamma}(h(G_i), k)$ are 0-1-valued. If $\mathcal{L}^{\gamma/2}(h(G_j), k) = 1$, we have by Equation (24),

$$h(G_i)[k] \leq \gamma/4 + h(G_j)[k]$$
$$\leq \gamma/4 + \gamma/2 + \max_{l=1,\ldots,K} h(G_j)[l]$$
$$\leq \gamma/4 + \gamma/2 + \gamma/4 + \max_{l=1,\ldots,K} h(G_i)[l]$$
$$= \gamma + \max_{l=1,\ldots,K} h(G_i)[l],$$

i.e., $\mathcal{L}^{\gamma}(h(G_i), k) = 1$ as well.

Finally, we continue with the calculation in Equation (22),

$$\mathcal{L}_{\text{te}}^{\gamma/2}(h) - \mathcal{L}_{\text{tr}}^{\gamma}(h) \leq \frac{1}{N_{\text{te}} N_{\text{tr}}} \sum_{G_i \in \mathcal{G}_{\text{tr}}} \sum_{G_j \in \mathcal{G}_{\text{te}}} \sum_{k=1}^{K} \left( \mathcal{L}^{\gamma/2}(G_j, k) - \mathcal{L}^{\gamma}(G_i, k) \right) + \left( \eta_k(G_j) - \eta_k(G_i) \right)$$

$$\leq \frac{1}{N_{\text{tr}}} \sum_{G_i \in \mathcal{G}_{\text{tr}}} \frac{1}{N_{\text{te}}} \sum_{G_j \in \mathcal{G}_{\text{te}}} \sum_{k=1}^{K} (0 + C\xi)$$

$$= CK\xi,$$

$$(26)$$

where the second inequality holds by Equation (23) and Equation (25). □

**Step 3: Bounding the discrepancy term $D_{\mathrm{tr,te}}^{\gamma/2}(P; \lambda)$.**

**Lemma E.3.** *Let $\alpha > 0$. For any $0 < \lambda \leq N^{2\alpha}$ and $\gamma \geq 0$, assume the "prior" $P$ on $\mathcal{H}$ is defined by sampling the vectorized trainable weight matrices from $\mathcal{N}(0, \sigma^2 I)$ for some $\sigma^2 \leq \frac{(\gamma/8\xi)^{2/D}}{2b(\lambda N_{\mathrm{tr}}^{-\alpha} + \ln 2bD)}$. We have*

$$D_{\mathrm{tr,te}}^{\gamma/2}(P; \lambda) \leq \ln 3 + \lambda C K \xi, \tag{27}$$

*where $D_{\mathrm{te,tr}}^{\gamma/2}(P; \lambda) = \ln \mathbb{E}_{h \sim P} e^{\lambda\left(\mathcal{L}_{\mathrm{te}}^{\gamma/2}(h) - \mathcal{L}_{\mathrm{tr}}^{\gamma}(h)\right)}$.*

*Proof.* First, set $T_h := \max_{j=1,\ldots,D} \|w_j\|_2$. We prove this lemma by partitioning $\mathcal{H}$ into two events: one with high probability, where the spectral norms of the model parameters satisfy the conditions in Lemma E.2, and its complement. For the latter event, we use Assumption D.1.

For any $j = 1, \ldots, D$, we have, by (Tropp, 2015), for any $t > 0$,

$$\Pr\left(\|w_j\|_2 \geq t\right) \leq 2be^{-\frac{t^2}{2b\sigma^2}},$$

where $b$ is the maximum width of all hidden layers of the considered classifier. We set $t = \left(\frac{\gamma}{8\xi}\right)^{1/D}$. Applying a union bound leads to

$$\Pr\left(T_h^D \xi > \frac{\gamma}{8}\right) = \Pr\left(T_h > \left(\frac{\gamma}{8\xi}\right)^{1/D}\right) \leq 2bDe^{-\frac{(\gamma/8\varepsilon)^{2/D}}{2b\sigma^2}} \leq e^{-\lambda N_{\mathrm{tr}}^{-\alpha}}, \tag{28}$$

where the last inequality uses the condition $\sigma^2 \leq \frac{(\gamma/8\xi)^{2/D}}{2b(\lambda N_{\mathrm{tr}}^{-\alpha} + \ln 2bD)}$.

For any $h$ satisfying $T_h^D \xi \leq \frac{\gamma}{8}$, by Lemma E.2, we have $e^{\lambda\left(\mathcal{L}_{\mathrm{te}}^{\gamma/4}(h) - \mathcal{L}_{\mathrm{tr}}^{\gamma/2}(h)\right)} \leq e^{\lambda C K \xi}$.

The complement event, i.e., $T_h^D \xi > \frac{\gamma}{8}$ occurs with probability at most $e^{-\lambda N_{\mathrm{tr}}^{-\alpha}}$. We decompose $D_{\mathrm{tr,te}}^{\gamma/2}(P; \lambda)$ as follows,

$$
\begin{aligned}
D_{\mathrm{tr,te}}^{\gamma/2}(P; \lambda) &= \ln \mathbb{E}_{h \sim P} e^{\lambda\left(\mathcal{L}_{\mathrm{te}}^{\gamma/4}(h) - \mathcal{L}_{\mathrm{tr}}^{\gamma/2}(h)\right)} \\
&\leq \ln\left(\Pr\left(T_h^D \xi \leq \frac{\gamma}{8}\right) e^{\lambda C K \xi} + \Pr\left(T_h^D \xi > \frac{\gamma}{8}\right) \mathbb{E}_{h \sim P \mid T_h^D \xi > \frac{\gamma}{8}} e^{\lambda\left(\mathcal{L}_{\mathrm{te}}^{\gamma/4}(h) - \mathcal{L}_{\mathrm{tr}}^{\gamma/2}(h)\right)}\right) \\
&\leq \ln\left(e^{\lambda C K \xi} + e^{-\lambda N_{\mathrm{tr}}^{-\alpha}} \mathbb{E}_{h \sim P \mid T_h^D \xi > \frac{\gamma}{8}} e^{\lambda\left(\mathcal{L}_{\mathrm{te}}^{\gamma/4}(h) - \mathcal{L}_{\mathrm{tr}}^{\gamma/2}(h)\right)}\right) \\
&\leq \ln\left(e^{\lambda C K \xi} + e^{-\lambda N_{\mathrm{tr}}^{-\alpha}} \left(e^{-N_{\mathrm{tr}}^{2\alpha}} \cdot e^{\lambda} + \left(1 - e^{-N_{\mathrm{tr}}^{2\alpha}}\right) \cdot e^{\lambda N_{\mathrm{tr}}^{-\alpha} + \lambda C K \xi}\right)\right) \\
&= \ln\left(e^{\lambda C K \xi} + e^{\lambda - N_{\mathrm{tr}}^{2\alpha} - \lambda N_{\mathrm{tr}}^{-\alpha}} + e^{-\lambda N_{\mathrm{tr}}^{-\alpha}} \left(\left(1 - e^{-N^{2\alpha}}\right) \cdot e^{\lambda N_{\mathrm{tr}}^{-\alpha} + \lambda C K \xi}\right)\right) \\
&= \ln\left(e^{\lambda C K \xi} + e^{\lambda - N_{\mathrm{tr}}^{2\alpha} - \lambda N_{\mathrm{tr}}^{-\alpha}} + \left(\left(1 - e^{-N_{\mathrm{tr}}^{2\alpha}}\right) \cdot e^{\lambda C K \xi}\right)\right) \\
&\leq \ln\left(2e^{\lambda C K \xi} + 1\right) \\
&\leq \ln 3 + \lambda C K \xi.
\end{aligned}
$$
$$\tag{29}$$

The first inequality holds by decomposing the domain over the expectation/integral into the event in which $T_h^D \xi \leq \frac{\gamma}{8}$ holds and its complement. The second inequality holds by $\Pr\left(T_h^D \xi \leq \frac{\gamma}{8}\right) \leq 1$ and Equation (28). The third inequality holds by Assumption D.1. The second-to-last inequality holds by the assumption $0 < \lambda \leq N^{2\alpha}$. The remaining equations and inequalities are algebraic reformulations. □

### E.2 PROOF OF THEOREM 4.1

We first present auxiliary results for the proof of Theorem 4.1, focusing on MPNNs, which may be end-to-end learnable. At the end of this chapter, we provide the full proof of Theorem 4.1, building on the results from this and the previous section. For simplicity, we derive the results for MPNNs; the corresponding results for $\zeta$-MPNNs applied to a graph $G$ follow by applying the derived results for standard MPNNs to the strong simulation $\zeta$. Let us denote the hypothesis space of MPNNs by $\mathcal{H}_{\mathrm{mpnn}}$, where the parameters follow the assumptions in Theorem 4.1.

**Step 4: MPNNs are stable under weight pertubations.** We proceed by proving the following result that shows that MPNNs are stable under small pertubations of their weights.

**Lemma E.4.** *Let $\tilde{h}$ be any classifier in $\mathcal{H}_{\mathrm{mpnn}}$ with learnable weight matrices $\mathbf{w} = \left\{ \{w(f^{(t)}, s)\}_{s=1}^{d(f^{(t)})}, \{w(g^{(t)}, s)\}_{s=1}^{d(g^{(t)})}, \{w(c, s)\}_{s=1}^{d(c)} \right\}$ and $\tilde{\beta} > 0$. Let $\mathcal{G}_{B,d}$ be the set of graphs with maximum degree $d$ and input node features in a ball of radius $B$. Then,*

$$
\max_{G \in \mathcal{G}_{B,d}} |\tilde{h}_{\mathbf{w}+\mathbf{u}}(G) - \tilde{h}_{\mathbf{w}}(G)|
$$

$$
\leq e \left( \prod_{s=1}^{d(c)} \|w(c, s)\|_2 \right) \left( \prod_{k=0}^{T-1} \left( \prod_{s}^{d(f^{(k+1)})} \|w(f^{(k+1)}, s)\| \right) \left( 1 + d \left( \prod_{s}^{d(g^{(k+1)})} \|w(g^{(k+1)}, s)\| \right) \right) \right) B
$$

$$
\cdot \sum_{t=0}^{T-1} \left( \sum_s \frac{\|u(g^{(t+1)}, s)\|}{\|w(g^{(t+1)}, s)\|} + \sum_s \frac{\|u(f^{(t+1)}, s)\|}{\|w(f^{(t+1)}, s)\|} \right).
$$

(30)

*Proof.* We denote by $x_v^{(t)}$ the representation of node $v$ after $t$ layers of the MPNN $\tilde{h}_{\mathbf{w}}$ with standard weights $\mathbf{w}$ and $\tilde{x}_v^{(t)}$ is the representation of node $v$ after $t$ layers of the MPNN $\tilde{h}_{\mathbf{w}+\mathbf{u}}$ with perturbed weights $\mathbf{w} + \mathbf{u}$. We similarly define the message terms $m_v^{(t)}$ and $\tilde{m}_v^{(t)}$. We further define

$$
\delta_{t+1} := \max_v \left\| \tilde{x}_v^{(t+1)} - x_v^{(t+1)} \right\|.
$$

We calculate

$$
\begin{aligned}
\delta_{t+1} &= \max_v \left\| \tilde{x}_v^{(t+1)} - x_v^{(t+1)} \right\| \\
&= \max_v \left\| f_{w+u}^{(t+1)}(\tilde{x}_v^{(t)}, \tilde{m}_v^{(t+1)}) - f_w^{(t+1)}(x_v^{(t)}, m_v^{(t+1)}) \right\| \\
&\leq \left\| f_{w+u}^{(t+1)}(\tilde{x}_{v^*}^{(t)}, \tilde{m}_{v^*}^{(t+1)}) - f_{w+u}^{(t+1)}(x_{v^*}^{(t)}, m_{v^*}^{(t+1)}) \right\| \\
&\quad + \left\| f_{w+u}^{(t+1)}(x_{v^*}^{(t)}, m_{v^*}^{(t+1)}) - f_w^{(t+1)}(x_{v^*}^{(t)}, m_{v^*}^{(t+1)}) \right\| \\
&= (A) + (B),
\end{aligned}
$$

where $v^*$ is the node where the maximum is taken.

**Bound** $(A)$. We begin by bounding the first term $(A)$. It holds

$$
\begin{aligned}
(A) &\leq L(f_{w+u}^{(t+1)}) \left( \|\tilde{x}_v^{(t)} - x_v^{(t)}\| + \|\tilde{m}_v^{(t)} - m_v^{(t)}\| \right) \\
&\leq L(f_{w+u}^{(t+1)}) \left( \delta_t + \left\| \sum_{\tilde{u} \in \mathcal{N}(v)} g_{w+u}^{(t+1)}(\tilde{x}_{\tilde{u}}^{(t)}) - g_w^{(t+1)}(x_{\tilde{u}}^{(t)}) \right\| \right) \\
&\leq L(f_{w+u}^{(t+1)}) \left( \delta_t + d \max_{u^* \in \mathcal{N}(v)} \left\| g_{w+u}^{(t+1)}(\tilde{x}_{u^*}^{(t)}) - g_w^{(t+1)}(x_{u^*}^{(t)}) \right\| \right).
\end{aligned}
$$

(31)

We calculate,

$$
\begin{aligned}
&\|g_{w+u}^{(t+1)}(\tilde{x}_{u^*}^{(t)}) - g_w^{(t+1)}(x_{u^*}^{(t)})\| \\
&\leq \|g_{w+u}^{(t+1)}(\tilde{x}_{u^*}^{(t)}) - g_{w+u}^{(t+1)}(x_{u^*}^{(t)})\| + \|g_{w+u}^{(t+1)}(x_{u^*}^{(t)}) - g_w^{(t+1)}(x_{u^*}^{(t)})\| \\
&\leq \left(1 + \frac{1}{D}\right)^{d(g^{(t+1)})} \left(\prod_s \|w(g^{(t+1)}, s)\|\right) \max_{u^* \in \mathcal{N}(v)} \|x_{u^*}^{(t)}\| \sum_s \frac{\|u(g^{(t+1)}, s)\|}{\|w(g^{(t+1)}, s)\|} \\
&\quad + L(g_{w+u}^{(t+1)}) \max_{u^* \in \mathcal{N}(v)} \|\tilde{x}_{u^*}^{(t)} - x_{u^*}^{(t)}\|.
\end{aligned}
\tag{32}
$$

Hence,

$$
\begin{aligned}
(A) \leq L(f_{w+u}^{(t+1)}) \Bigg( &\delta_t + d \left(1 + \frac{1}{D}\right)^{d(g^{(t+1)})} \left(\prod_s \|w(g^{(t+1)}, s)\|\right) \|x^{(t)}\| \sum_s \frac{\|u(g^{(t+1)}, s)\|}{\|w(g^{(t+1)}, s)\|} \\
&+ L(g_{w+u}^{(t+1)})\delta_t \Bigg)
\end{aligned}
\tag{33}
$$

**Bound** $(B)$. We continue by bounding the first term $(B)$. It holds

$$
(B) \leq \left(1 + \frac{1}{D}\right)^{d(f^{(t+1)})} \left(\prod_s \|w(f^{(t+1)}, s)\|\right) \left(\|x_{v^*}^{(t)}\| + \|m_{v^*}^{(t+1)}\|\right) \sum_s \frac{\|u(f^{(t+1)}, s)\|}{\|w(f^{(t+1)}, s)\|}.
\tag{34}
$$

Together, we have

$$
\begin{aligned}
\delta_{t+1} \leq\ & L(f_{w+u}^{(t+1)}) \Bigg(\delta_t + d \left(1 + \frac{1}{D}\right)^{d(g^{(t+1)})} \left(\prod_s \|w(g^{(t+1)}, s)\|\right) \|x^{(t)}\| \sum_s \frac{\|u(g^{(t+1)}, s)\|}{\|w(g^{(t+1)}, s)\|} + L(g_{w+u}^{(t+1)})\delta_t \Bigg) \\
&+ \left(1 + \frac{1}{D}\right)^{d(f^{(t+1)})} \left(\prod_s \|w(f^{(t+1)}, s)\|\right) \left(\|x_{v^*}^{(t)}\| + \|m_{v^*}^{(t+1)}\|\right) \sum_s \frac{\|u(f^{(t+1)}, s)\|}{\|w(f^{(t+1)}, s)\|} \\
=\ & L(f_{w+u}^{(t+1)}) \Big((1 + L(g_{w+u}^{(t+1)}))\Big)\delta_t \\
&+ L(f_{w+u}^{(t+1)}) \left(d \left(1 + \frac{1}{D}\right)^{d(g^{(t+1)})} \left(\prod_s \|w(g^{(t+1)}, s)\|\right) \|x^{(t)}\| \sum_s \frac{\|u(g^{(t+1)}, s)\|}{\|w(g^{(t+1)}, s)\|}\right) \\
&+ \left(1 + \frac{1}{D}\right)^{d(f^{(t+1)})} \left(\prod_s \|w(f^{(t+1)}, s)\|\right) \left(\|x_{v^*}^{(t)}\| + \|m_{v^*}^{(t+1)}\|\right) \sum_s \frac{\|u(f^{(t+1)}, s)\|}{\|w(f^{(t+1)}, s)\|}.
\end{aligned}
\tag{35}
$$

We solve this recurrence relation to get

$$
\delta_T \leq \sum_{t=0}^{T-1} \left(\prod_{k=t+1}^{T-1} A_k\right) B_t,
$$

where

$$
A_t = L(f_{w+u}^{(t+1)})\Big(1 + L(g_{w+u}^{(t+1)})\Big),
\tag{36}
$$

$$
\begin{aligned}
B_t &= L(f_{w+u}^{(t+1)}) \left( d \left(1 + \tfrac{1}{D}\right)^{d(g^{(t+1)})} \left( \prod_{s}^{d(g^{(t+1)})} \| w(g^{(t+1)}, s) \| \right) \right. \\
&\quad \cdot \left( \prod_{k=1}^{t} L(f_w^{(k)}) \left(1 + dL(g_w^{(k)})\right) \right) B \sum_{s} \frac{\| u(g^{(t+1)}, s) \|}{\| w(g^{(t+1)}, s) \|} \right) \\
&\quad + \left(1 + \tfrac{1}{D}\right)^{d(f^{(t+1)})} \left( \prod_{s}^{d(f^{(t+1)})} \| w(f^{(t+1)}, s) \| \right) \\
&\quad \cdot \left( (1 + dL(g_w^{(t+1)})) \left( \prod_{k=1}^{t} L(f_w^{(k)}) \left(1 + dL(g_w^{(k)})\right) \right) B \right) \sum_{s} \frac{\| u(f^{(t+1)}, s) \|}{\| w(f^{(t+1)}, s) \|},
\end{aligned}
\tag{37}
$$

where

$$
L(f_{w+u}^{(t+1)}) = \left( \prod_{s=1}^{d(f^{(t+1)})} \| w(f^{(t)}, s) + u(f^{(t)}, s) \| \right),
\tag{38}
$$

$$
L(g_{w+u}^{(t+1)}) = \left( \prod_{s=1}^{d(g^{(t+1)})} \| w(g^{(t)}, s) + u(g^{(t)}, s) \| \right),
\tag{39}
$$

and the product $\prod_{k=t+1}^{T-1} A_k$ is taken to be $1$ when $t = T$ (empty product).

**Final step.** As a final step we need to incorporate the MLP classifier after the $T$ message passing layers. For this we calculate,

$$
\begin{aligned}
\delta_{T+1} &= \left\| f_{w+u} \left( \frac{1}{N} \sum_{v=1}^{N} \tilde{x}_v^{(T)} \right) - f_w \left( \frac{1}{N} \sum_{v=1}^{N} x_v^{(T)} \right) \right\| \\
&\leq \left\| f_{w+u} \left( \frac{1}{N} \sum_{v=1}^{N} \tilde{x}_v^{(T)} \right) - f_{w+u} \left( \frac{1}{N} \sum_{v=1}^{N} x_v^{(T)} \right) \right\| \\
&\quad + \left\| f_{w+u} \left( \frac{1}{N} \sum_{i=1}^{N} x_v^{(T)} \right) - f_w \left( \frac{1}{N} \sum_{v=1}^{N} x_v^{(T)} \right) \right\| \\
&\leq \left(1 + \frac{1}{D}\right)^{d(f)} \left( \prod_{s=1}^{d(f)} \| w(c, s) \|_2 \right) \delta_T + \left(1 + \frac{1}{D}\right)^{d(f)} \left( \prod_{s=1}^{d(f)} \| w(c, s) \|_2 \right) \left\| x^{(T)} \right\|_2 \sum_{i=1}^{d(f)} \frac{\| u(f, s) \|_2}{\| w(c, s) \|_2} \\
&\leq \left(1 + \frac{1}{D}\right)^{d(f)} \left( \prod_{s=1}^{d(f)} \| w(c, s) \|_2 \right) \sum_{t=0}^{T-1} \left( \prod_{k=t+1}^{T-1} A_k \right) B_t \\
&\quad + \left(1 + \frac{1}{D}\right)^{d(f)} \left( \prod_{s=1}^{d(f)} \| w(c, s) \|_2 \right) \left( \prod_{t=1}^{T} L(f_w^{(t)}) \left(1 + dL(g_w^{(t)})\right) \right) B \sum_{i=1}^{d(f)} \frac{\| u(f, s) \|_2}{\| w(c, s) \|_2} \\
&= (C) + (D).
\end{aligned}
\tag{40}
$$

For the term $(C)$, we calculate

$$\left(1 + \frac{1}{D}\right)^{d(c)} \left(\prod_{s=1}^{d(c)} \|w(c,s)\|_2\right) \sum_{t=0}^{T-1} \left(\prod_{k=t+1}^{T-1} A_k\right) B_t$$

$$\leq \left(1 + \frac{1}{D}\right)^{d(c)} \left(\prod_{s=1}^{d(c)} \|w(c,s)\|_2\right) \sum_{t=0}^{T-1} \left(\prod_{k=t+1}^{T-1} L(f_{w+u}^{(k+1)})\left(1 + L(g_{w+u}^{(k+1)})\right)\right) B_t$$

$$= \left(1 + \frac{1}{D}\right)^{d(c)} \left(\prod_{s=1}^{d(c)} \|w(c,s)\|_2\right) \sum_{t=0}^{T-1} \left(\prod_{k=t+1}^{T-1} \left(1 + \frac{1}{D}\right)^{d(f^{(k+1)})} \left(\prod_s^{d(f^{(k+1)})} \|w(f^{(k+1)}, s)\|\right)\right.$$

$$\left. \cdot \left(1 + \left(1 + \frac{1}{D}\right)^{d(g^{(k+1)})} \left(\prod_s^{d(g^{(k+1)})} \|w(g^{(k+1)}, s)\|\right)\right)\right) \cdot B_t$$

$$(41)$$

For $B_t$, we calculate

$$B_t = L(f_{w+u}^{(t+1)})\left(d\left(1 + \frac{1}{D}\right)^{d(g^{(t+1)})} \left(\prod_s^{d(g^{(t+1)})} \|w(g^{(t+1)}, s)\|\right)\right.$$

$$\cdot \left(\prod_{k=1}^t L(f_w^{(k)})\left(1 + dL(g_w^{(k)})\right)\right) D \sum_s \frac{\|u(g^{(t+1)}, s)\|}{\|w(g^{(t+1)}, s)\|}\right)$$

$$+ \left(1 + \frac{1}{D}\right)^{d(f^{(t+1)})} \left(\prod_s^{d(f^{(t+1)})} \|w(f^{(t+1)}, s)\|\right)$$

$$\cdot \left((1 + dL(g_w^{(t+1)}))\left(\prod_{k=1}^t L(f_w^{(k)})\left(1 + dL(g_w^{(k)})\right)\right) D\right) \sum_s \frac{\|u(f^{(t+1)}, s)\|}{\|w(f^{(t+1)}, s)\|}$$

$$= \left(1 + \frac{1}{D}\right)^{d(f^{(t+1)})} \left(\prod_s^{d(f^{(t+1)})} \|w(f^{(t+1)}, s)\|\right) \left(\prod_{k=1}^t L(f_w^{(k)})\left(1 + dL(g_w^{(k)})\right)\right) B$$

$$\cdot \left(d\left(1 + \frac{1}{D}\right)^{d(g^{(t+1)})} \left(\prod_s^{d(g^{(t+1)})} \|w(g^{(t+1)}, s)\|\right) \sum_s \frac{\|u(g^{(t+1)}, s)\|}{\|w(g^{(t+1)}, s)\|}\right.$$

$$\left. + \left(1 + d\left(\prod_s^{d(f^{(t+1)})} \|w(f^{(t+1)}, s)\|\right)\right) \sum_s \frac{\|u(f^{(t+1)}, s)\|}{\|w(f^{(t+1)}, s)\|}\right)$$

$$\leq \left(1 + \frac{1}{D}\right)^{d(f^{(t+1)})} \left(\prod_s^{d(f^{(t+1)})} \|w(f^{(t+1)}, s)\|\right) \left(\prod_{k=1}^t L(f_w^{(k)})\left(1 + dL(g_w^{(k)})\right)\right)$$

$$\cdot B \left(1 + d\left(1 + \frac{1}{D}\right)^{d(g^{(t+1)})} \left(\prod_s^{d(g^{(t+1)})} \|w(g^{(t+1)}, s)\|\right)\right) \cdot \left(\sum_s \frac{\|u(g^{(t+1)}, s)\|}{\|w(g^{(t+1)}, s)\|} + \sum_s \frac{\|u(f^{(t+1)}, s)\|}{\|w(f^{(t+1)}, s)\|}\right)$$

$$\leq \left(1 + \frac{1}{D}\right)^{d(f^{(t+1)}) + d(g^{(t+1)})} \left(\prod_{k=1}^{t+1} L(f_w^{(k)})\left(1 + dL(g_w^{(k)})\right)\right) B \cdot \left(\sum_s \frac{\|u(g^{(t+1)}, s)\|}{\|w(g^{(t+1)}, s)\|} + \sum_s \frac{\|u(f^{(t+1)}, s)\|}{\|w(f^{(t+1)}, s)\|}\right),$$

$$(42)$$

since $L(g_w^{(t+1)}) = \prod_s^{d(g^{(t+1)})} \|w(g^{(t+1)}, s)\|$ and $L(f_w^{(t+1)}) = \prod_s^{d(f^{(t+1)})} \|w(f^{(t+1)}, s)\|$, respectively.

Together, we get

$$
\begin{aligned}
(C) &\leq \left(1+\frac{1}{D}\right)^{D} \left(\prod_{s=1}^{d(c)} \|w(c,s)\|_2\right) \sum_{t=0}^{T-1} \left(\prod_{k=t+1}^{T-1} \left(\prod_s^{d(f^{(k+1)})} \|w(f^{(k+1)},s)\|\right) \cdot \left(1 + \left(\prod_s^{d(g^{(k+1)})} \|w(g^{(k+1)},s)\|\right)\right)\right) \\
&\quad \cdot \left(1+\frac{1}{D}\right)^{d(f^{(t+1)})+d(g^{(t+1)})} \left(\prod_{k=1}^{t+1} L(f_w^{(k)})\left(1+dL(g_w^{(k)})\right)\right) B \cdot \left(\sum_s \frac{\|u(g^{(t+1)},s)\|}{\|w(g^{(t+1)},s)\|} + \sum_s \frac{\|u(f^{(t+1)},s)\|}{\|w(f^{(t+1)},s)\|}\right) \\
&\leq \left(1+\frac{1}{D}\right)^{D} \left(\prod_{s=1}^{d(c)} \|w(c,s)\|_2\right) \left(\prod_{k=0}^{T-1} \left(\prod_s^{d(f^{(k+1)})} \|w(f^{(k+1)},s)\|\right) \cdot \left(1 + d\left(\prod_s^{d(g^{(k+1)})} \|w(g^{(k+1)},s)\|\right)\right)\right) B \\
&\quad \cdot \sum_{t=0}^{T-1} \left(\sum_s \frac{\|u(g^{(t+1)},s)\|}{\|w(g^{(t+1)},s)\|} + \sum_s \frac{\|u(f^{(t+1)},s)\|}{\|w(f^{(t+1)},s)\|}\right) \\
&\leq e \left(\prod_{s=1}^{d(c)} \|w(c,s)\|_2\right) \left(\prod_{k=0}^{T-1} \left(\prod_s^{d(f^{(k+1)})} \|w(f^{(k+1)},s)\|\right) \cdot \left(1 + d\left(\prod_s^{d(g^{(k+1)})} \|w(g^{(k+1)},s)\|\right)\right)\right) B \\
&\quad \cdot \sum_{t=0}^{T-1} \left(\sum_s \frac{\|u(g^{(t+1)},s)\|}{\|w(g^{(t+1)},s)\|} + \sum_s \frac{\|u(f^{(t+1)},s)\|}{\|w(f^{(t+1)},s)\|}\right),
\end{aligned}
\tag{43}
$$

where $D = d(f) + \sum_{k=1}^{T} \left(d\left(f^{(k)}\right) + d\left(g^{(k)}\right)\right)$. Hence,

$$
\begin{aligned}
\delta_{T+1} &\leq (C) + (D) \\
&\quad e \left(\prod_{s=1}^{d(c)} \|w(c,s)\|_2\right) \left(\prod_{k=0}^{T-1} \left(\prod_s^{d(f^{(k+1)})} \|w(f^{(k+1)},s)\|\right) \cdot \left(1 + d\left(\prod_s^{d(g^{(k+1)})} \|w(g^{(k+1)},s)\|\right)\right)\right) B \\
&\quad \cdot \sum_{t=0}^{T-1} \left(\sum_s \frac{\|u(g^{(t+1)},s)\|}{\|w(g^{(t+1)},s)\|} + \sum_s \frac{\|u(f^{(t+1)},s)\|}{\|w(f^{(t+1)},s)\|}\right) \\
&\quad + \left(1+\frac{1}{D}\right)^{d(f)} \left(\prod_{s=1}^{d(f)} \|w(c,s)\|_2\right) \left(\prod_{t=1}^{T} L(f_w^{(t)})\left(1+dL(g_w^{(t)})\right)\right) B \sum_{i=1}^{d(f)} \frac{\|u(f,s)\|_2}{\|w(c,s)\|_2} \\
&\leq e \left(\prod_{s=1}^{d(c)} \|w(c,s)\|_2\right) \left(\prod_{k=0}^{T-1} \left(\prod_s^{d(f^{(k+1)})} \|w(f^{(k+1)},s)\|\right) \cdot \left(1 + d\left(\prod_s^{d(g^{(k+1)})} \|w(g^{(k+1)},s)\|\right)\right)\right) B \\
&\quad \cdot \left(\sum_{t=0}^{T-1} \left(\sum_s \frac{\|u(g^{(t+1)},s)\|}{\|w(g^{(t+1)},s)\|} + \sum_s \frac{\|u(f^{(t+1)},s)\|}{\|w(f^{(t+1)},s)\|}\right) + \sum_{i=1}^{d(f)} \frac{\|u(f,s)\|_2}{\|w(c,s)\|_2}\right).
\end{aligned}
\tag{44}
$$

$\square$

**Step 5: Derive sufficient conditions for Lemma E.1.** We proceed with the following lemma that gives sufficient conditions under which the conditions of Lemma E.1 are satisfied.

**Lemma E.5.** *Let $\tilde{h}$ be any classifier in $\mathcal{H}_{mpnn}$ with learnable weight matrices $\mathbf{w} = \left\{\{w(f^{(t)},s)\}_{s=1}^{d(f^{(t)})}, \{w(g^{(t)},s)\}_{s=1}^{d(g^{(t)})}, \{w(c,s)\}_{s=1}^{d(c)}\right\}$ and $\tilde{\beta} > 0$. Consider random perturbations to $\mathbf{w}$ given by $\mathbf{u} = \left\{\{u(f^{(t)},s)\}_{s=1}^{d(f^{(t)})}, \{u(g^{(t)},s)\}_{s=1}^{d(g^{(t)})}, \{u(c,s)\}_{s=1}^{d(c)}\right\}$, where each perturbation follows an independent Gaussian distribution $\mathcal{N}(0,\sigma^2 I)$. Suppose the following conditions hold:*

- *The variance of the perturbation satisfies*

$$
\sigma \leq \frac{\gamma}{e^2 B \left(\tilde{\beta}^{D-(\sum_{k=1}^{T} d(g^{(k)}))-1} + d^{T-1}\tilde{\beta}^{D-1}\right)\sqrt{2h\ln(4Dh)}}.
\tag{45}
$$

- *All weights $w \in \mathbf{w}$ satisfy $\|w\|_2 = \beta$, with $|\tilde{\beta} - \beta| \leq \frac{\tilde{\beta}}{D}$.*

*Then, with respect to the random draw of $\mathbf{u}$,*

$$\Pr\left(\max_{G \in \mathcal{G}_{\mathrm{tr}} \cup \mathcal{G}_{\mathrm{te}}} \|\tilde{h}_{\mathbf{w}}(G) - \tilde{h}_{\mathbf{w}+\mathbf{u}}(G)\|_\infty < \frac{\gamma}{8}\right) \geq \frac{1}{2}.$$

*Proof.* By $|\beta - \tilde{\beta}| \leq \frac{1}{D}\tilde{\beta}$, we get

$$\frac{1}{e}\beta^{D-1} \leq \tilde{\beta}^{D-1} \leq e\beta^{D-1}.$$

We can bound the spectral norm of each perturbation matrix $u \in \mathbf{u}$, by Tropp (2015), as follows:

$$\Pr\left(\|u\|_2 > t\right) \leq 2b \exp\left(-\frac{t^2}{2b\sigma^2}\right),$$

where $b$ represents the hidden dimension. Applying a union bound over all layers, we obtain that with probability at least $1/2$, the spectral norm of each perturbation $u \in \mathbf{u}$ is bounded by

$$\sigma\sqrt{2b\ln(4Db)}.$$

Substituting this spectral norm bound into Lemma E.4, we have with probability at least $1/2$,

$$\max_{G \in \mathcal{G}_{\mathrm{tr}} \cup \mathcal{G}_{\mathrm{te}}} |\tilde{h}_{\mathbf{w}+\mathbf{u}}(G) - \tilde{h}_{\mathbf{w}}(G)|$$

$$\leq e \left( \prod_{s=1}^{d(c)} \|w(c,s)\|_2 \right) \prod_{k=0}^{T-1} \left( \prod_{s}^{d(f^{(k+1)})} \|w(f^{(k+1)},s)\| \right) \cdot \left( 1 + d \prod_{s}^{d(g^{(k+1)})} \|w(g^{(k+1)},s)\| \right) B$$

$$\cdot \left( \sum_{t=0}^{T-1} \left( \sum_s \frac{\|u(g^{(t+1)},s)\|}{\|w(g^{(t+1)},s)\|} + \sum_s \frac{\|u(f^{(t+1)},s)\|}{\|w(f^{(t+1)},s)\|} \right) + \sum_{i=1}^{d(f)} \frac{\|u(f,s)\|_2}{\|w(c,s)\|_2} \right)$$

$$\leq e \left( \prod_{s=1}^{d(c)} \|w(c,s)\|_2 \right) \prod_{k=0}^{T-1} \left( \prod_{s}^{d(f^{(k+1)})} \|w(f^{(k+1)},s)\| \right) \cdot d \prod_{s}^{d(g^{(k+1)})} \|w(g^{(k+1)},s)\| B$$

$$\cdot \left( \sum_{t=0}^{T-1} \left( \sum_s \frac{\|u(g^{(t+1)},s)\|}{\|w(g^{(t+1)},s)\|} + \sum_s \frac{\|u(f^{(t+1)},s)\|}{\|w(f^{(t+1)},s)\|} \right) + \sum_{i=1}^{d(f)} \frac{\|u(f,s)\|_2}{\|w(c,s)\|_2} \right)$$

$$+ e \left( \prod_{s=1}^{d(c)} \|w(c,s)\|_2 \right) \prod_{k=0}^{T-1} \left( \prod_{s}^{d(f^{(k+1)})} \|w(f^{(k+1)},s)\| \right) B$$

$$\cdot \left( \sum_{t=0}^{T-1} \left( \sum_s \frac{\|u(g^{(t+1)},s)\|}{\|w(g^{(t+1)},s)\|} + \sum_s \frac{\|u(f^{(t+1)},s)\|}{\|w(f^{(t+1)},s)\|} \right) + \sum_{i=1}^{d(f)} \frac{\|u(f,s)\|_2}{\|w(c,s)\|_2} \right)$$

$$= e\beta^{D-1}d^{T-1}B \cdot \left( \sum_{t=0}^{T-1} \left( \sum_s \|u(g^{(t+1)},s)\| + \sum_s \|u(f^{(t+1)},s)\| \right) + \sum_{s=1}^{d(f)} \|u(f,s)\| \right)$$

$$+ e\beta^{D-(\sum_{k=1}^{T} d(g^{(k)}))-1}B \cdot \left( \sum_{t=0}^{T-1} \left( \sum_s \|u(g^{(t+1)},s)\| + \sum_s \|u(f^{(t+1)},s)\| \right) + \sum_{s=1}^{d(f)} \|u(f,s)\| \right)$$

$$\leq e^2 \tilde{\beta}^{D-1}d^{T-1}B \cdot \left( \sum_{t=0}^{T-1} \left( \sum_s \|u(g^{(t+1)},s)\| + \sum_s \|u(f^{(t+1)},s)\| \right) + \sum_{s=1}^{d(f)} \|u(f,s)\| \right)$$

$$+ e^2 \tilde{\beta}^{D-(\sum_{k=1}^{T} d(g^{(k)}))-1}B \cdot \left( \sum_{t=0}^{T-1} \left( \sum_s \|u(g^{(t+1)},s)\| + \sum_s \|u(f^{(t+1)},s)\| \right) + \sum_{s=1}^{d(f)} \|u(f,s)\| \right)$$

$$= e^2 \left( \tilde{\beta}^{D-(\sum_{k=1}^{T} d(g^{(k)}))-1} + d^{T-1}\tilde{\beta}^{D-1} \right) B \cdot \left( \sum_{t=0}^{T-1} \left( \sum_s \|u(g^{(t+1)},s)\| + \sum_s \|u(f^{(t+1)},s)\| \right) + \sum_{s=1}^{d(f)} \|u(f,s)\| \right)$$

$$\leq e^2 \left( \tilde{\beta}^{D-(\sum_{k=1}^{T} d(g^{(k)}))-1} + d^{T-1}\tilde{\beta}^{D-1} \right) B \sigma \sqrt{2b \ln(4Db)}$$

$$\leq \frac{\gamma}{4},$$

where for the last inequality we used Equation (45). $\qquad\square$

**Step 6: Putting everything together.** We finish this section by reformulating and proving Theorem 4.1.

**Theorem E.6.** *Let $\tilde{h}$ be any classifier in $\mathcal{H}_{mpnn}$ with parameters $\{w_i\}_{i=1}^{D}$. For any $\gamma \geq 0$, $\alpha \geq 1/4$ and large enough $N_{\mathrm{tr}}$, with probability at least $1 - \delta$ over the sample of $y_{\mathrm{tr}}$, we have*

$$\mathcal{L}_{\mathrm{te}}^0(\tilde{h}) - \hat{\mathcal{L}}_{\mathrm{tr}}^\gamma(\tilde{h}) \leq \mathcal{O} \left( \frac{b \sum_i \|w_i\|_F^2}{N_{\mathrm{tr}}^{2\alpha} (\gamma/8)^{2/D}} \xi^{2/D} + \frac{1}{N_{\mathrm{tr}}^{2\alpha}} h^2 \ln 2h \frac{DC (2dB)^{1/D}}{\gamma^{1/D}\delta} + \frac{1}{N_{\mathrm{tr}}^{1-2\alpha}} + CK\xi \right).$$

$$(46)$$

*Proof.* The proof follows the proof of Theorem 1 in (Neyshabur et al., 2018) Theorem 6 in (Ma et al., 2021).

There are two main steps in the proof. In the first step, for a given constant $\beta > 0$, we first define the "prior" $P$ and the "posterior" $Q$ on $\mathcal{H}$ in a way complying with conditions in Lemma E.1, Lemma E.3, and Lemma E.5. Without loss of generality (due to the homogeneity of the activation function), we can assume that $\|w_i^{(j)}\|_2 = \beta$ for some $\beta \geq 0$. Then, for all classifiers with parameters satisfying $|\beta - \tilde{\beta}| \leq \frac{\tilde{\beta}}{D}$ and $\tilde{\beta}$ being some value on a predefined grid in the parameters space, we can derive a generalization bound by applying Lemma E.1, Lemma E.3, and Lemma E.5.

In the second step, we investigate the number of $\tilde{\beta}$ we need to cover all possible relevant classifier parameters and apply a union bound to get the final bound. The two steps are essentially the same as Neyshabur et al. (2018) with the first step differing by the need for incorporating Lemma E.1.

**Step 1.** We begin by showing the first step. Given a choice of $\tilde{\beta}$ independent of the training data, let

$$\sigma = \min \left( \frac{(\gamma/8\xi)^{1/D}}{\sqrt{2b\left(\lambda N_{\text{tr}}^{-\alpha} + \ln 2bD\right)}}, \quad \frac{\gamma}{e^2 B \left(\tilde{\beta}^{D-(\sum_{k=1}^T d(g^{(k)}))-1} + d^{T-1}\tilde{\beta}^{D-1}\right)\sqrt{2h\ln(4Dh)}} \right). \tag{47}$$

Assume the "prior" $P$ on $\mathcal{H}$ is defined by sampling the vectorized MLP parameters from $\mathcal{N}(0, \sigma^2 I)$. The "posterior" $Q$ on $\mathcal{H}$ is defined by first sampling a set of random perturbations $\{u_i\}_{i=1}^D$ and then adding them to $\{w_i\}_{i=1}^D$. By Lemma E.5, we have

$$\Pr \left( \max_{G \in \mathcal{G}_{\text{tr}} \cup \mathcal{G}_{\text{te}}} \|\tilde{h}_{\mathbf{w}}(G) - \tilde{h}_{\mathbf{w}+\mathbf{u}}(G)\|_\infty < \frac{\gamma}{8} \right) \geq \frac{1}{2}. \tag{48}$$

Therefore, by applying Lemma E.1, we get with probability at least $1 - \delta$,

$$\begin{aligned} \mathcal{L}_{\text{te}}^0(\tilde{h}) - \hat{\mathcal{L}}_{\text{tr}}^\gamma(\tilde{h}) &\leq \frac{1}{\lambda}\left( 2\left(D_{\text{KL}}(Q\|P) + 1\right) + \ln\frac{1}{\delta} + \frac{\lambda^2}{4N_{\text{tr}}} + D_{\text{te,tr}}^{\gamma/2}(P;\lambda)\right) \\ &\leq \frac{1}{\lambda}\left( 2\left(D_{\text{KL}}(Q\|P) + 1\right) + \ln\frac{1}{\delta} + \frac{\lambda^2}{4N_{\text{tr}}} + (\ln 3 + \lambda C K \xi)\right) \\ &\leq \frac{2}{N_{\text{tr}}^{2\alpha}}D_{\text{KL}}(Q\|P) + \frac{1}{N_{\text{tr}}^{2\alpha}}\ln\frac{1}{\delta} + \frac{1}{4N_{\text{tr}}^{1-2\alpha}} + \frac{2 + \ln 3}{N_{\text{tr}}^{2\alpha}} + C K \xi. \end{aligned} \tag{49}$$

where we chose $\lambda = N_{\text{tr}}^{2\alpha}$.

Moreover, since both $P$ and $Q$ are normal distributions, we know that

$$D_{\text{KL}}(Q\|P) \leq \sum_i \frac{\|w_i\|_2^2}{2\sigma^2}. \tag{50}$$

Per assumption, both $B$ and $D$ are constant with respect to $N_{\text{tr}}$. Hence, for large enough $N_{\text{tr}}$,

$$\frac{(\gamma/8\xi)^{1/D}}{\sqrt{2b\left(\lambda N_{\text{tr}}^{-\alpha} + \ln 2bD\right)}} < \frac{\gamma}{4(e^2+1)e^2\, d^J B\, \beta^{D-1}\sqrt{2h\ln(4Dh)}}, \tag{51}$$

which implies,

$$\sigma = \frac{(\gamma/8\xi)^{1/D}}{\sqrt{2b\left(\lambda N_{\text{tr}}^{-\alpha} + \ln 2bD\right)}} \tag{52}$$

and hence

$$D_{\text{KL}}(Q\|P) \leq \frac{b\left(N_{\text{tr}}^\alpha + \ln 2bD\right)\sum_i \|w_i\|_2^2}{(\gamma/8)^{2/D}}\xi^{2/D}. \tag{53}$$

Therefore, with probability at least $1 - \delta$,

$$\begin{aligned} \mathcal{L}_{\text{te}}^0(\tilde{h}) - \mathcal{L}_{\text{tr}}^\gamma(\tilde{h}) &\leq \frac{2}{N_{\text{tr}}^{2\alpha}}D_{\text{KL}}(Q\|P) + \frac{1}{N_{\text{tr}}^{2\alpha}}\ln\frac{1}{\delta} + \frac{1}{4N_{\text{tr}}^{1-2\alpha}} + \frac{2 + \ln 3}{N_{\text{tr}}^{2\alpha}} + C K \xi \\ &\leq \mathcal{O}\left( \frac{b\sum_i \|w_i\|_2^2}{N_{\text{tr}}^\alpha (\gamma/8)^{2/D}}\xi^{2/D} + \frac{1}{N_{\text{tr}}^{2\alpha}}\ln\frac{1}{\delta} + \frac{1}{N_{\text{tr}}^{1-2\alpha}} + C K \xi \right). \end{aligned} \tag{54}$$

**Step 2.** Then we show the second step, i.e., finding out the number of $\tilde{\beta}$ we need to cover all possible relevant classifier parameters. Similarly as Neyshabur et al. (2018), we will show that we only need to consider $\left(\frac{\gamma}{2B}\right)^{1/D} \leq \beta \leq C$ (recall that the spectral norm of all weight matrices is bounded by $C$). If $\beta < \left(\frac{\gamma}{2B}\right)^{1/D}$, then for any graph $G \in \mathcal{G}_{\mathrm{tr}} \cup \mathcal{G}_{\mathrm{te}}$, we get $\|\tilde{h}(G)\|_\infty \leq \frac{\gamma}{2}$, which implies that the bound trivially holds. Since we only consider $\beta$ in the above range, a sufficient condition to make $|\beta - \tilde{\beta}| \leq \frac{\beta}{D}$ hold would be $|\beta - \tilde{\beta}| \leq \frac{1}{D}\left(\frac{\gamma}{2B}\right)^{1/D}$. Therefore, it suffices to find a covering for all possible weight matrices with radius $\frac{1}{D}\left(\frac{\gamma}{2B}\right)^{1/D}$ for a ball in $\mathbb{R}^{b \times b}$ of radius $C$. This can be satisfied by $\left(2\frac{DCb(2B)^{1/D}}{\gamma^{1/D}}\right)^{b^2}$ balls. Taking a union bound, we get, with probability at least $1 - \delta$,

$$\mathcal{L}_{\mathrm{te}}^0(\tilde{h}) - \mathcal{L}_{\mathrm{tr}}^\gamma(\tilde{h}) \leq \mathcal{O}\left(\frac{b\sum_j \sum_i \|w_i\|_2^2}{N_{\mathrm{tr}}^{2\alpha}(\gamma/8)^{2/D}}\xi^{2/D} + \frac{1}{N_{\mathrm{tr}}^{2\alpha}}b^2\ln 2b\frac{DC(2B)^{1/D}}{\gamma^{1/D}\delta} + \frac{1}{N_{\mathrm{tr}}^{1-2\alpha}} + CK\xi\right). \tag{55}$$

$\square$

## F    FIXED GRAPH ENCODERS

Next, we apply our analysis to GNNs with fixed encoders. Here, the embedding function $e : \mathcal{G} \to \mathbb{R}^b$ is fixed, and only the classifier $c : \mathbb{R}^b \to \mathbb{R}^K$ is trainable. Let $\mathcal{H} = \mathcal{C} \circ \mathcal{E}$, where $\mathcal{E}$ now represents a fixed graph embedding network. The generalization bound in this setting simplifies to the following.

**Corollary F.1.** *Let* lat *be any pseudometric in the latent space $\mathbb{R}^d$. Under mild assumptions (see Appendix D), for any $\gamma > 0$ and $0 < \alpha < \frac{1}{4}$, with probability at least $1 - \delta$ over the sample of training labels $y_{\mathrm{tr}}$, the test loss of any classifier $\tilde{h} \in \mathcal{H}$*

$$\mathcal{L}_{\mathrm{te}}^0(\tilde{h}) \;\leq\; \widehat{\mathcal{L}}_{\mathrm{tr}}^\gamma(\tilde{h}) \;+\; \mathcal{O}\Big(\underbrace{\tfrac{\mathrm{MC}(\mathcal{C})}{N_{tr}^{2\alpha}\,\gamma^{1/D}\,\delta}}_{\substack{\text{complexity term}}} \;+\; \underbrace{C\,\xi_{\mathrm{lat}}}_{\substack{\text{structural}\\\text{similarity term}}}\Big), \tag{56}$$

*where $\xi_{\mathrm{lat}} = \max_{G \in \mathcal{G}_{te}} \min_{H \in \mathcal{G}_{tr}} \mathrm{lat}\left(g(G), g(H)\right)$ and the other constants are defined in Theorem 4.1.*

*Proof.* The proof follows the same steps as in the proof of Theorem E.6. For example, Lemma E.4 can be shown for MLPs by simply assuming that there are no MPNN layers in Lemma E.4. Theorem E.6 can then be proved for the hypothesis space $\mathcal{H}$ by defining the variance as

$$\sigma = \min\left(\frac{(\gamma/8\xi)^{1/D}}{\sqrt{2b\left(\lambda N_{\mathrm{tr}}^{-\alpha} + \ln 2bD\right)}}, \quad \frac{\gamma}{e^2 B\,\tilde{\beta}^{D-1}\sqrt{2b\ln(4Db)}}\right).$$

$\square$

In this case, the complexity of the graph embedding model no longer contributes to the bound, and only the complexity of the MLP classifier affects generalization. This reduced complexity underscores promise of fixed graph encoders.

## G    PROOF OF LEMMA ON COMPARISON OF TMD AND $\mathcal{F}$-TMD

**Lemma G.1.** *For any $G, H \in \mathcal{G}$ and any non-empty family of features $\mathcal{F}' \subset \mathcal{F} \subset \mathcal{G}$, we have*

$$\mathcal{F}'\text{-}TMD(G, H) \leq \mathcal{F}\text{-}TMD(G, H).$$

*Proof.* Starting from the definitions, we have:

$$\mathcal{F}'\text{-}TMD(G, H) = \min_{\sigma \in S_n} \sum_{j=1}^n \mathrm{TD}\left(T_j^{G^{\mathcal{F}'}}, T_{\sigma(j)}^{H^{\mathcal{F}'}}\right),$$

$$\mathcal{F}\text{-TMD}(G, H) = \min_{\sigma \in S_n} \sum_{j=1}^{n} \text{TD}\left(T_j^{G^{\mathcal{F}}}, T_{\sigma(j)}^{H^{\mathcal{F}}}\right),$$

where $S_n$ is the set of all permutations of the node set $\{1, 2, \ldots, n\}$, and $T_j^{G^{\mathcal{F}}}$ denotes the subtree of $G^{\mathcal{F}}$ (the graph $G$ augmented with additional features $\mathcal{F}$) at node $j$.

Let $\sigma'$ be the optimal assignment for $\mathcal{F}\text{-TMD}(G, H)$, i.e.,

$$\mathcal{F}\text{-TMD}(G, H) = \sum_{j=1}^{n} \text{TD}\left(T_j^{G^{\mathcal{F}}}, T_{\sigma'(j)}^{H^{\mathcal{F}}}\right).$$

Since $\mathcal{F}'\text{-TMD}(G, H)$ optimizes over all assignments, it follows that

$$\mathcal{F}'\text{-TMD}(G, H) \le \sum_{j=1}^{n} \text{TD}\left(T_j^{G^{\mathcal{F}'}}, T_{\sigma'(j)}^{H^{\mathcal{F}'}}\right). \tag{57}$$

Our goal is to show that for every $j$:

$$\text{TD}\left(T_j^{G^{\mathcal{F}'}}, T_{\sigma'(j)}^{H^{\mathcal{F}'}}\right) \le \text{TD}\left(T_j^{G^{\mathcal{F}}}, T_{\sigma'(j)}^{H^{\mathcal{F}}}\right). \tag{58}$$

Together, Equation (57) and Equation (58), lead to

$$\begin{aligned}
\mathcal{F}'\text{-TMD}(G, H) &\le \sum_{j=1}^{n} \text{TD}\left(T_j^{G^{\mathcal{F}'}}, T_{\sigma'(j)}^{H^{\mathcal{F}'}}\right) \\
&\le \sum_{j=1}^{n} \text{TD}\left(T_j^{G^{\mathcal{F}}}, T_{\sigma'(j)}^{H^{\mathcal{F}}}\right) \\
&= \mathcal{F}\text{-TMD}(G, H).
\end{aligned}$$

We proceed to prove Equation (58) via induction. In fact, we show a more general version: for every assignment $\rho$ between nodes of $G$ and $H$ and any node $j$, we have

$$\text{TD}\left(T_j^{G^{\mathcal{F}'}}, T_{\rho(j)}^{H^{\mathcal{F}'}}\right) \le \text{TD}\left(T_j^{G^{\mathcal{F}}}, T_{\rho(j)}^{H^{\mathcal{F}}}\right).$$

**Base Case ($t = 0$).**

At depth $0$, the trees consist only of root nodes. The tree distance is the difference between node features, i.e.,

$$\begin{aligned}
\text{TD}\left(T_j^{G^{\mathcal{F}'}}, T_{\rho(j)}^{H^{\mathcal{F}'}}\right) &= \|x'_j - x'_{\rho(j)}\| \\
&\le \|\tilde{x}_j - \tilde{x}_{\rho(j)}\| \\
&= \text{TD}\left(T_j^{G^{\mathcal{F}}}, T_{\rho(j)}^{H^{\mathcal{F}}}\right),
\end{aligned}$$

where $x'_j = (x_j, c_j(F_1), \ldots, c_j(F_{|\mathcal{F}'|}))$ and $\tilde{x}_j = (x_j, c_j(F_1), \ldots, c_j(F_{|\mathcal{F}|}))$ include the additional features.

**Induction Step ($t - 1 \mapsto t$).**

Assume that for every tree of depth $t - 1$, we have, for every assignment $\rho$ and every $j = 1, \ldots, n$,

$$\text{TD}\left(T_j^{G^{\mathcal{F}'}}, T_{\rho(j)}^{H^{\mathcal{F}'}}\right) \le \text{TD}\left(T_j^{G^{\mathcal{F}}}, T_{\rho(j)}^{H^{\mathcal{F}}}\right).$$

Let $\sigma'$ be any assignment of trees and let $\tau$ be the optimal assignment in $W_{\text{TD}}\left(\mathcal{T}_j^{\mathcal{F}}, \mathcal{T}_{\sigma'(j)}^{\mathcal{F}}\right)$.

Then, for any $j = 1, \dots, n$

$$
\begin{aligned}
\mathrm{TD}\left(T_j^{G^{\mathcal{F}'}}, T_{\sigma'(j)}^{H^{\mathcal{F}'}}\right) &= \|x_j' - x_{\sigma'(j)}'\| + \min_{\tau' \in S_n} \sum_{j=1}^n \mathrm{TD}\left(T_j^{G^{\mathcal{F}'}}, T_{\tau'(j)}^{H^{\mathcal{F}'}}\right) \\
&\leq \|x_j' - x_{\sigma'(j)}'\| + \sum_{j=1}^n \mathrm{TD}\left(T_j^{G^{\mathcal{F}'}}, T_{\tau(j)}^{H^{\mathcal{F}'}}\right) \\
&\leq \|x_j' - x_{\sigma'(j)}'\| + \sum_{j=1}^n \mathrm{TD}\left(T_j^{G^{\mathcal{F}}}, T_{\tau(j)}^{H^{\mathcal{F}}}\right) \qquad (59) \\
&= \|x_j' - x_{\sigma'(j)}'\| + W_{\mathrm{TD}}\left(\rho\left(\mathcal{T}_j^{\mathcal{F}}, \mathcal{T}_{\sigma'(j)}^{\mathcal{F}}\right)\right) \\
&\leq \|\tilde{x}_j - \tilde{x}_{\sigma'(j)}\| + W_{\mathrm{TD}}\left(\rho\left(\mathcal{T}_j^{\mathcal{F}}, \mathcal{T}_{\sigma'(j)}^{\mathcal{F}}\right)\right) \\
&= \mathrm{TD}\left(T_j^{G^{\mathcal{F}}}, T_{\sigma'(j)}^{H^{\mathcal{F}}}\right).
\end{aligned}
$$

The second inequality follows by the induction hypothesis as the considered trees are of depth $t-1$. Hence, by induction, the inequality $\mathrm{TD}\left(T_j^{G^{\mathcal{F}'}}, T_{\sigma'(j)}^{H^{\mathcal{F}'}}\right) \leq \mathrm{TD}\left(T_j^{G^{\mathcal{F}}}, T_{\rho(j)}^{H^{\mathcal{F}}}\right)$ holds, completing the proof. $\qquad \square$

*Proof of Theorem 6.1.* Assume that $y \sim \mathcal{F}$-TMD.

1. Since any $\mathcal{F}$-GIN classifier $h$ is Lipschitz continuous with respect to $\mathcal{F}$-TMD and $h$ is stable with respect to weight perturbations, by Theorem 4.1, we have

$$
\mathcal{L}_{\mathrm{te}}^0(\tilde{h}) \leq \mathcal{L}_{\mathrm{tr}}^\gamma(\tilde{h}) + \mathcal{O}\left( \frac{b \sum_j \sum_i \|W_i^{(j)}\|_F^2}{N_{\mathrm{tr}}^{2\alpha}(\gamma/8)^{2/D}} \xi^{2/D} + \frac{h^2 \ln(2h) DC(2dB)^{1/D}}{N_{\mathrm{tr}}^{2\alpha} \gamma^{1/D} \delta} + \frac{1}{N_{\mathrm{tr}}^{1-2\alpha}} + CK\xi \right),
$$

   with $\xi := \max_{G_{\mathrm{tr}} \in \mathcal{G}_{\mathrm{tr}}, G_{\mathrm{te}} \in \mathcal{G}_{\mathrm{te}}} \mathcal{F}\text{-TMD}_w^{L+1}(G_{\mathrm{tr}}, G_{\mathrm{te}})$ and $B = \max_G \|X(R^{\mathcal{F}}(G))[i,:]\|_2$.

2. By Lemma G.1, every $\mathcal{F}'$-GIN classifier $h'$ is Lipschitz continuous with respect to $\mathcal{F}$-TMD, as
$$
\|h'(G) - h'(H)\| \leq L' \mathcal{F}'\text{-TMD}(G, H) \leq L' \mathcal{F}\text{-TMD}(G, H). \qquad (60)
$$
   Hence, applying Theorem 4.1, we have

$$
\mathcal{L}_{\mathrm{te}}^0(\tilde{h}) \leq \mathcal{L}_{\mathrm{tr}}^\gamma(\tilde{h}) + \mathcal{O}\left( \frac{b \sum_j \sum_i \|W_i^{(j)}\|_F^2}{N_{\mathrm{tr}}^{2\alpha}(\gamma/8)^{2/D}} \xi^{2/D} + \frac{h^2 \ln(2h) DC(2dB')^{1/D}}{N_{\mathrm{tr}}^{2\alpha} \gamma^{1/D} \delta} + \frac{1}{N_{\mathrm{tr}}^{1-2\alpha}} + CK\xi \right),
$$

   where $B' = \max_G \|X(R^{\mathcal{F}'}(G))[i,:]\|_2$. Note that $B'$ is the only difference compared to the previous bound.

3. Assume $y \sim \mathcal{F}$-TMD, but apply a $\tilde{\mathcal{F}}$-GIN classifier $\tilde{h}$. Since $\tilde{h}$ is not necessarily Lipschitz continuous with respect to $\tilde{\mathcal{F}}$-TMD, we cannot directly apply Theorem 4.1.

   However, if $y \sim \mathcal{F}$-TMD, then $y \sim \tilde{\mathcal{F}}$-TMD as well. Specifically, if $y \sim \mathcal{F}$-TMD, there exists for every class $k$ some $\eta_k$ such that $\eta_k(G) = \Pr(y_G = k \mid G)$, and $\eta_k$ is Lipschitz continuous with constant $L_{\eta^k}$. Then, by Lemma G.1,
$$
|\eta_k(G) - \eta_k(H)| \leq L_{\eta^k} \cdot \mathcal{F}\text{-TMD}(G, H) \leq L_{\eta^k} \cdot \tilde{\mathcal{F}}\text{-TMD}(G, H). \qquad (61)
$$
   Consequently, we can now apply Theorem 4.1 to obtain

$$
\mathcal{L}_{\mathrm{te}}^0(\tilde{h}) \leq \mathcal{L}_{\mathrm{tr}}^\gamma(\tilde{h}) + \mathcal{O}\left( \frac{b \sum_j \sum_i \|W_i^{(j)}\|_F^2}{N_{\mathrm{tr}}^{2\alpha}(\gamma/8)^{2/D}} \tilde{\xi}^{2/D} + \frac{h^2 \ln(2h) DC(2d\tilde{B})^{1/D}}{N_{\mathrm{tr}}^{2\alpha} \gamma^{1/D} \delta} + \frac{1}{N_{\mathrm{tr}}^{1-2\alpha}} + CK\tilde{\xi} \right),
$$

   where $\tilde{\xi} := \max_{G_{\mathrm{tr}} \in \mathcal{G}_{\mathrm{tr}}, G_{\mathrm{te}} \in \mathcal{G}_{\mathrm{te}}} \tilde{\mathcal{F}}\text{-TMD}_w^{L+1}(G_{\mathrm{tr}}, G_{\mathrm{te}})$ and $\tilde{B} = \max_G \|X(R^{\tilde{\mathcal{F}}}(G))[i,:]\|_2$. Note that $\tilde{B}$ *and* $\tilde{\xi}$ are now different compared to the previous bounds in Item 1 and 2.

$\qquad \square$

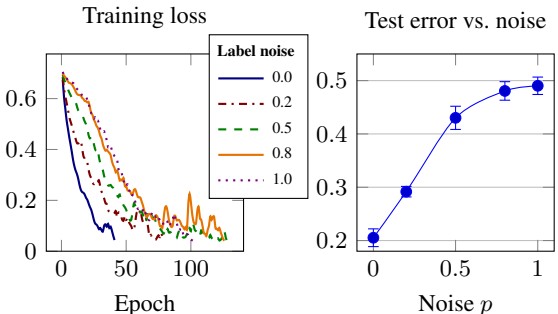 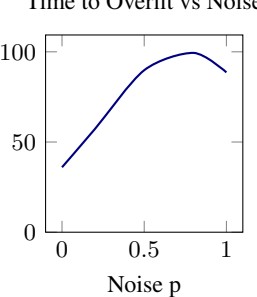

Figure 4: **Left:** Training-loss trajectories of a GIN on MUTAGENICITY under increasing label noise $p$. **Center:** Corresponding test errors on MUTAGENICITYrises sharply as label–structure correlation is essential for generalization (mean $\pm$ standard deviation across five seeds). **Right:** Number of epochs to overfit, i.e., reach 99% training accuracy under increasing label noise $p$.

## H   OTHER RESULTS

The following lemmata are easy to prove.

**Lemma H.1.** *Given a MPNN with $t$ layers such that the message and update functions $g^{(k)}$ and $f^{(k)}$ are Lipschitz continuos with Lipschitz constant $L(f^{(k)})$ and $L(g^{(k)})$, respectively. Then for any graph $G$ with maximum node degree $v$, we have*

$$\max_{v \in V(G)} \|x_v^{(t)}\| \leq \left( \prod_{k=1}^{t} L(f^{(k)}) \left( 1 + dL(g^{(k)}) \right) \right) B, \tag{62}$$

*where $x_v^{(t)}$ denotes the output of the MPNN after $t$ layers at node $v$.*

**Lemma H.2.** *Given a MPNN with $t+1$ layers such that the message and update functions $g^{(k)}$ and $f^{(k)}$ are Lipschitz continuos with Lipschitz constant $L(f^{(k)}$ and $L(g^{(k)})$, respectively. Then for any graph $G$ with maximum node degree $v$, we have*

$$\max_v \|m_v^{(t+1)}\| \leq dL_{g^{(t+1)}} \left( \prod_{k=1}^{t} L(f_w^{(k)}) \left( 1 + dL(g_w^{(k)}) \right) \right) B, \tag{63}$$

*where $m_v^{(t)}$ denotes the message of the MPNN at the $(t+1)$'th layer at node $v$.*

**Lemma H.3** (Lemma 2 in (Neyshabur et al., 2018)). *Let $f_{\mathbf{w}} : \mathbb{R}^n \to \mathbb{R}^k$ be a neural network with Lipschitz continuous and homogenous activations and $d(f)$ layers For any $D$, Then for any $\mathbf{w}$, $\mathbf{x} \in \mathcal{X}_{B,n}$, and any perturbation $\mathbf{u} = \mathrm{vec}(\{\mathbf{U}_i\}_{i=1}^{d})$ such that $\|\mathbf{U}_i\|_2 \leq \frac{1}{d}\|\mathbf{W}_i\|_2$, the change in the output of the network can be bounded as follows:*

$$\|f_{\mathbf{w}+\mathbf{u}}(\mathbf{x}) - f_{\mathbf{w}}(\mathbf{x})\|_2 \leq eB \left( \prod_{i=1}^{d} \|\mathbf{W}_i\|_2 \right) \sum_{i=1}^{d} \frac{\|\mathbf{U}_i\|_2}{\|\mathbf{W}_i\|_2}.$$

## I   ADDITIONAL EXPERIMENTAL RESULTS AND DETAILS

This appendix provides a comprehensive overview of our experimental setup, including dataset generation, model architectures, training procedures, evaluation metrics, and additional analyses. All experiments were run on an internal cluster with Intel Xeon CPUs (28 cores, 192GB RAM) and GeForce RTX 3090 Ti GPUs (4 units, 24GB memory each), as well as Intel Xeon CPUs (32 cores, 192GB RAM) and NVIDIA RTX A6000 GPUs (3 units, 48GB memory each). Each subsection corresponds to one of the three experimental tasks.

### I.1   LABEL NOISE EXPERIMENTS

To investigate how MPNNs behave under noisy supervision, we conducted controlled label corruption experiments on three benchmark molecular graph classification datasets from the TUDataset collection

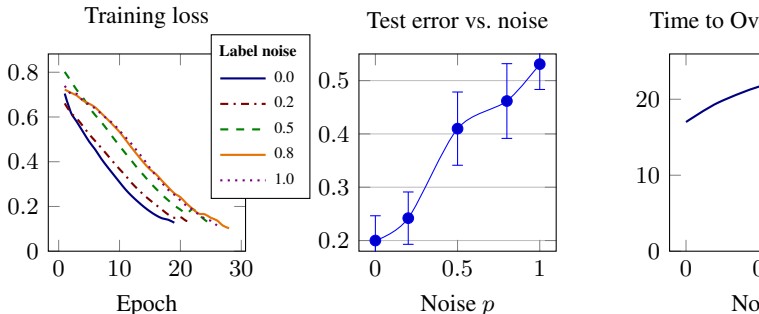

Figure 5: **Left:** Training-loss trajectories of a GIN on BZR under increasing label noise $p$. **Center:** Corresponding test errors on BZRrises sharply as label–structure correlation is essential for generalization (mean $\pm$ standard deviation across five seeds). **Right:** Number of epochs to overfit, i.e., reach 99% training accuracy under increasing label noise $p$.

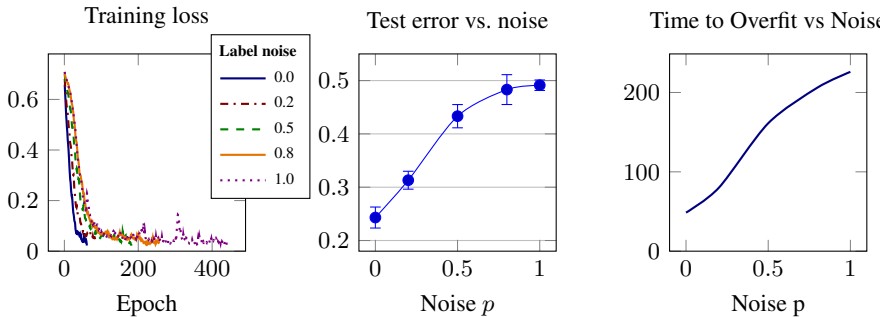

Figure 6: **Left:** Training-loss trajectories of a GIN on NCI109 under increasing label noise $p$. **Center:** Corresponding test errors on NCI109 rises sharply as label–structure correlation is essential for generalization (mean $\pm$ standard deviation across five seeds). **Right:** Number of epochs to overfit, i.e., reach 99% training accuracy under increasing label noise $p$.

(Morris et al., 2020a): MUTAGENICITY, NCI109, and BZR. For each dataset, we randomly corrupted a fixed proportion of the training labels by replacing them with uniformly sampled class labels.

We employed a MPNN with four layers, ReLU activations, GraphNorm, and a two-layer MLP head. Each message and update function in the MPNN is given by an MLP with two layers. The models were trained for up to 5000 epochs with early stopping once 99% training accuracy was achieved. Label noise levels were varied in $0.0, 0.2, 0.4, 0.6, 0.8, 1.0$, and results were averaged over five random seeds per setting.

We track the per-epoch training and test accuracy/loss, the number of epochs until memorization, and total training time. This setup closely follows the protocol of Zhang et al. (2017), adapted to the graph setting, and enables us to study not only final generalization but also how memorization unfolds over time under increasing label noise.

Figure 4, 5, and 6 visualize the results on MUTAGENICITY, BZR, and NCI109, respectively. Each figure presents from left to right: (left) the full training curves, (middle) final test accuracy versus noise level, and (right) the number of epochs required to reach 99% training accuracy. These plots collectively illustrate the sharp transition from generalization to memorization and the dataset-specific sensitivity of MPNNs to label corruption.

As the label noise increases, the time required to reach 99% training accuracy increases moderately—suggesting that fitting corrupted labels is harder, but still feasible. However, all models eventually reach near-perfect training accuracy across all noise levels, even for fully randomized labels ($p = 1.0$), underscoring the high memorization capacity of GNNs. In stark contrast, the test accuracy consistently deteriorates with increasing noise and converges to chance level at $p = 1.0$,

where labels are entirely uninformative. This gap between training and test performance confirms that GNNs can overfit to pure noise and emphasizes the need for principled regularization and early stopping to preserve generalization.

## I.2  TASK 1: MEDIAN-BASED LABELING WITH CYCLE COUNTS

To investigate the role of local structural patterns in graph classification, we generate 3,000 random graphs using three common models: Erdős–Rényi (ER), Barabási–Albert (BA), and Stochastic Block Model (SBM). Each graph's label is related to the sum of its 3-cycle and 4-cycle counts, which are computed using NetworkX (Hagberg et al., 2008). The total cycle count is then used to assign labels: graphs with counts below the dataset median are labeled 0, while those above receive label 1.

For all synthetic graphs, we sample the number of nodes randomly between 35 and 55. For each random graph model, we chose the following parameters.

- Erdős–Rényi (ER) Graphs: We generate an ER graph with edge probability $p = 0.1$.
- Barabási–Albert (BA) Graphs: Each new node is connected to $m = 2$ existing nodes following the BA preferential attachment process.
- Stochastic Block Model (SBM) Graphs: We randomly select between 3 and 6 blocks, ensuring each block has at least 3 nodes. The probability of an edge within the same block is sampled uniformly between $[0.1, 0.3]$, while inter-block connections have a probability in the range $[0.001, 0.02]$.

We evaluate multiple GNN variants to assess the impact of different levels of expressivity:

- MPNN: A standard Message Passing Neural Network (Gilmer et al., 2017).
- $\mathcal{F}_l$-MPNN: MPNNs enriched with cycle counts of cycles up to length $l$ (Bouritsas et al., 2023; Barceló et al., 2021).
- Subgraph GNN: Incorporates subgraph structures (Frasca et al., 2022).
- Local 2-GNN: A local variant of 2-GNN (Morris et al., 2019).
- Local Folklore 2-GNN: A local variant of 2-Folklore-GNN (Zhang et al., 2024).

Each model is implemented in PyTorch Geometric (Fey & Lenssen, 2019), using the implementation details of Zhang et al. (2024). We trained the models using the Adam optimizer (Kingma & Ba, 2015) with a learning rate of $10^{-3}$ for 100 epochs and cosine scheduler. We fix the test set, and we perform 10-fold cross-validation, reporting mean accuracy (ACC) and standard deviation. Performance is measured at both the final epoch and the best validation epoch.

We present additional experimental results in Table 4–6 in this appendix. To further illustrate the strong correlation between labels and $\mathcal{F}_4$-TMD, we provide a qualitative visualization of the dataset structure (see Figure 7). Specifically, we select 50 graphs of the ER dataset and project them into a two-dimensional space using Multidimensional Scaling (MDS), with distances computed based on standard TMD, $\mathcal{F}_3$-TMD, $\mathcal{F}_4$-TMD, and $\mathcal{F}_{12}$-TMD. Among these, the projection using $\mathcal{F}_4$-TMD exhibits the clearest class separation, visually reinforcing its alignment with the classification task.

## I.3  TASK 2: REAL-WORLD DATASETS

We evaluate our generalization framework on six real-world datasets from the TU Dataset collection (Morris et al., 2020a), spanning both biological and chemical graph classification tasks:

- **Mutagenicity**: 4,337 molecular graphs labeled as mutagenic or non-mutagenic.
- **PROTEINS**: 1,113 protein graphs classified by their enzymatic function.
- **BZR**: 405 graphs representing benzodiazepine receptor ligands labeled for activity.
- **COX2**: 467 graphs labeled based on activity against the COX2 enzyme.
- **NCI109**: 4,127 graphs representing chemical compounds screened for anti-cancer activity.
- **AIDS**: 2,000 graphs with binary activity labels relevant to AIDS antiviral screening.

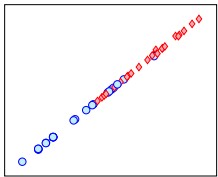 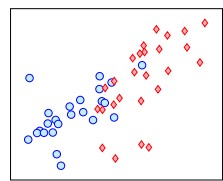 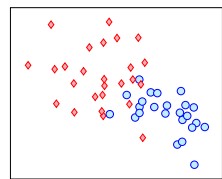 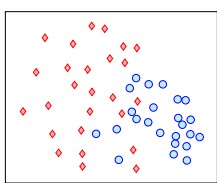

Figure 7: Low-dimensional embeddings via MDS for Task 1 in Section 6. MDS projects the graphs into $\mathbb{R}^2$ while preserving the pairwise $\mathcal{F}_l$-TMDs between graphs in the dataset. From left to right: TMD, $\mathcal{F}_3$-TMD, $\mathcal{F}_4$-TMD, and $\mathcal{F}_{12}$-TMD. Visually, $\mathcal{F}_4$-TMD achieves the best class separation, highlighting that the labels strongly correlate with $\mathcal{F}_4$-TMD. As a result, $\mathcal{F}_4$-MPNNs provide the best generalization and predictive performance.

To assess whether generalization in MPNNs and MLPs with fixed feature extractors depends on the structural similarity between test and training graphs, as predicted by our generalization bound in Theorem 4.1, we conduct two complementary evaluations:

(i) **GIN with TMD**: We train GINs (Xu et al., 2019) and measure distances between graphs using the TMD.

(ii) **Fixed encoder with Hamming distance**: We compute molecular fingerprints (Gainza et al., 2019) for each graph, apply an MLP classifier, and measure distances in the resulting feature space via Hamming distance.

We analyze two key properties:

**Error vs. TMD Distance.** To quantify the relationship between structural proximity and model performance, we report cumulative accuracy. Test graphs are ordered by increasing distance to the training set, and we compute the cumulative average accuracy. Specifically, given the ordered test set $G_1, \ldots, G_{N_{\text{te}}}$, we define

$$\tilde{y}_i = \frac{1}{i} \sum_{j=1}^{i} \mathbb{1} \left[ h(G_i)[y_{G_i}] \leq \max_{k \neq y_{G_i}} h(G)[k] \right],$$

where $\tilde{y}_i$ represents the average accuracy over the first $i$ graphs. For end-to-end trained GIN models, Figure 2 and Figure 8 illustrate that test accuracy consistently decreases as the structural distance from the training distribution increases. A similar trend is observed for MLP+fingerprint models in Figure 9. These empirical results align with our theoretical insights presented in Theorem 4.1, which predict this performance degradation.

**Theoretical vs. Empirical Generalization Bound.** We further examine how well our bound from Theorem 4.1 predicts the actual generalization behavior. For each dataset and model, we compute the theoretical bound using the empirical training loss and the graph distances to the training set as prescribed by Theorem 4.1. We compare this bound to the observed test error across all datasets. Results are presented in Figure 3 and in the appendix (Figure 10). While our bound is not tight, it does track the empirical error trends well across datasets, substantiating our claim that structural similarity with respect to the right pseudometric governs generalization in graph learning.

I.4 TASK 3: MDS-BASED LABELING VIA $\mathcal{F}_5$-TMD DISTANCES

For the second synthetic task, we construct 500 random graphs and compute pairwise distances using the $\mathcal{F}_5$-TMD pseudometric, where $\mathcal{F}_5$ includes cycles up to length 5. Labels are assigned using a clustering algorithm, ensuring that graphs closer in $\mathcal{F}_5$-TMD likely share the same label. We test different GNNs and summarize the results in Figure 11.

**Details.** For the second synthetic task, we generate 500 random graphs between 15 and 35 nodes using ER graphs with edge probability $p = 0.1$. We use the $\mathcal{F}_5$-TMD pseudometric, where $\mathcal{F}_5$ consists of cycles up to length 5. We compute pairwise graph distances with respect to $\mathcal{F}_5$-TMD

Table 4: Train and test accuracy, Erdős–Rényi graphs. The task is to predict if the count of cycles of length at most $4$ in the cycle basis of each graph is above or below the median of the whole dataset. The node features are augmented with ($\text{hom}_N$) homomorphism-counts of cycles up to length N, ($\text{sub}_N$) subgraph-counts of cycles up to length N, and ($\text{bas}_N$) number of cycle graphs up to length N in the cycle basis.

|  | (a) w/ early stopping. | | | (b) w/o early stopping. | | |
|---|---|---|---|---|---|---|
|  | Num. layers | | | Num. layers | | |
| Model | 1 | 3 | 5 | 1 | 3 | 5 |
| L-G | $0.9180 \pm 0.0179$ | $0.9216 \pm 0.0379$ | $0.9226 \pm 0.0318$ | $0.9904 \pm 0.0041$ | $0.9998 \pm 0.0004$ | $1.0000 \pm 0.0001$ |
|  | $0.8707 \pm 0.0105$ | $0.8707 \pm 0.0110$ | $0.8637 \pm 0.0106$ | $0.8543 \pm 0.0063$ | $0.8520 \pm 0.0073$ | $0.8553 \pm 0.0102$ |
| LF-G | $0.9162 \pm 0.0165$ | $0.9148 \pm 0.0409$ | $0.9226 \pm 0.0391$ | $0.9868 \pm 0.0051$ | $0.9999 \pm 0.0003$ | $1.0000 \pm 0.0001$ |
|  | $0.8647 \pm 0.0095$ | $0.8683 \pm 0.0086$ | $0.8623 \pm 0.0112$ | $0.8450 \pm 0.0135$ | $0.8540 \pm 0.0121$ | $0.8543 \pm 0.0116$ |
| MP | $0.9050 \pm 0.0094$ | $0.9135 \pm 0.0256$ | $0.9255 \pm 0.0403$ | $0.9906 \pm 0.0033$ | $1.0000 \pm 0.0001$ | $0.9998 \pm 0.0005$ |
|  | $0.8737 \pm 0.0091$ | $0.8694 \pm 0.0165$ | $0.8597 \pm 0.0126$ | $0.8490 \pm 0.0045$ | $0.8567 \pm 0.0087$ | $0.8637 \pm 0.0138$ |
| MP+hom$_3$ | $0.8993 \pm 0.0122$ | $0.9115 \pm 0.0316$ | $0.9335 \pm 0.0462$ | $0.9886 \pm 0.0039$ | $0.9999 \pm 0.0002$ | $1.0000 \pm 0.0000$ |
|  | $0.8783 \pm 0.0105$ | $0.8710 \pm 0.0116$ | $0.8750 \pm 0.0098$ | $0.8480 \pm 0.0108$ | $0.8537 \pm 0.0107$ | $0.8673 \pm 0.0141$ |
| MP+hom$_4$ | $0.8993 \pm 0.0121$ | $0.8998 \pm 0.0120$ | $0.9185 \pm 0.0408$ | $0.9302 \pm 0.0039$ | $0.9978 \pm 0.0018$ | $0.9990 \pm 0.0011$ |
|  | $0.8783 \pm 0.0089$ | $0.8767 \pm 0.0063$ | $0.8710 \pm 0.0125$ | $0.8720 \pm 0.0088$ | $0.8520 \pm 0.0138$ | $0.8540 \pm 0.0102$ |
| MP+hom$_7$ | $0.8913 \pm 0.0117$ | $0.9002 \pm 0.0198$ | $0.8915 \pm 0.0286$ | $0.9091 \pm 0.0081$ | $0.9795 \pm 0.0040$ | $0.9897 \pm 0.0058$ |
|  | $0.8740 \pm 0.0065$ | $0.8693 \pm 0.0178$ | $0.8700 \pm 0.0097$ | $0.8693 \pm 0.0099$ | $0.8450 \pm 0.0123$ | $0.8390 \pm 0.0145$ |
| MP+bas$_3$ | $0.9210 \pm 0.0201$ | $0.9465 \pm 0.0376$ | $0.9436 \pm 0.0393$ | $0.9956 \pm 0.0028$ | $0.9999 \pm 0.0002$ | $1.0000 \pm 0.0000$ |
|  | $0.8717 \pm 0.0173$ | $0.8773 \pm 0.0168$ | $0.8783 \pm 0.0096$ | $0.8657 \pm 0.0085$ | $0.8670 \pm 0.0097$ | $0.8723 \pm 0.0049$ |
| MP+bas$_4$ | $0.9754 \pm 0.0061$ | $0.9761 \pm 0.0066$ | $0.9841 \pm 0.0081$ | $0.9904 \pm 0.0034$ | $0.9985 \pm 0.0021$ | $0.9997 \pm 0.0004$ |
|  | $0.9870 \pm 0.0046$ | $0.9700 \pm 0.0154$ | $0.9603 \pm 0.0048$ | $0.9793 \pm 0.0068$ | $0.9587 \pm 0.0072$ | $0.9523 \pm 0.0102$ |
| MP+bas$_7$ | $0.9782 \pm 0.0087$ | $0.9804 \pm 0.0051$ | $0.9850 \pm 0.0155$ | $0.9958 \pm 0.0026$ | $0.9999 \pm 0.0003$ | $0.9998 \pm 0.0005$ |
|  | $0.9720 \pm 0.0111$ | $0.9573 \pm 0.0077$ | $0.9490 \pm 0.0093$ | $0.9660 \pm 0.0065$ | $0.9550 \pm 0.0062$ | $0.9537 \pm 0.0072$ |
| MP+sub$_3$ | $0.8979 \pm 0.0174$ | $0.9340 \pm 0.0473$ | $0.9166 \pm 0.0320$ | $0.9933 \pm 0.0029$ | $1.0000 \pm 0.0001$ | $1.0000 \pm 0.0000$ |
|  | $0.8827 \pm 0.0123$ | $0.8697 \pm .0135$ | $0.8747 \pm 0.0144$ | $0.8510 \pm 0.0078$ | $0.8547 \pm 0.0075$ | $0.8683 \pm 0.0080$ |
| MP+sub$_4$ | $0.8976 \pm 0.0111$ | $0.9090 \pm 0.0270$ | $0.9233 \pm 0.0287$ | $0.9853 \pm 0.0034$ | $0.9995 \pm 0.0009$ | $0.9998 \pm 0.0004$ |
|  | $0.8780 \pm 0.0031$ | $0.8717 \pm 0.0098$ | $0.8733 \pm 0.0109$ | $0.8487 \pm 0.0155$ | $0.8613 \pm 0.0127$ | $0.8623 \pm 0.0130$ |
| MP+sub$_7$ | $0.8963 \pm 0.0059$ | $0.9140 \pm 0.0174$ | $0.9312 \pm 0.0304$ | $0.9018 \pm 0.0035$ | $0.9670 \pm 0.0070$ | $0.9815 \pm 0.0090$ |
|  | $0.8790 \pm 0.0067$ | $0.8760 \pm 0.0092$ | $0.8657 \pm 0.0075$ | $0.8803 \pm 0.0070$ | $0.8680 \pm 0.0121$ | $0.8553 \pm 0.0135$ |
| Sub-G | $0.9194 \pm 0.0238$ | $0.9060 \pm 0.0338$ | $0.9561 \pm 0.0411$ | $0.9887 \pm 0.0048$ | $1.0000 \pm 0.0001$ | $1.0000 \pm 0.0000$ |
|  | $0.8710 \pm 0.0088$ | $0.8663 \pm 0.0074$ | $0.8640 \pm 0.0099$ | $0.8623 \pm 0.0058$ | $0.8533 \pm 0.0088$ | $0.8680 \pm 0.0100$ |

Table 5: Train and test accuracy, Barabasi-Albert graphs. The task is to predict if the count of cycles of length at most $4$ in the cycle basis of each graph is above or below the median of the whole dataset. The node features are augmented with ($\text{hom}_N$) homomorphism-counts of cycles up to length N, ($\text{sub}_N$) subgraph-counts of cycles up to length N, and ($\text{bas}_N$) number of cycle graphs up to length N in the cycle basis.

(a) w/ early stopping.

| Model | Num. layers 1 | 2 | 5 |
|---|---|---|---|
| L-G | $0.819 \pm 0.013$ 
 $0.767 \pm 0.014$ | $0.794 \pm 0.017$ 
 $0.751 \pm 0.013$ | $0.793 \pm 0.019$ 
 $0.747 \pm 0.010$ |
| LF-G | $0.818 \pm 0.018$ 
 $0.775 \pm 0.008$ | $0.807 \pm 0.021$ 
 $0.770 \pm 0.012$ | $0.822 \pm 0.051$ 
 $0.747 \pm 0.015$ |
| MP | $0.811 \pm 0.015$ 
 $0.771 \pm 0.015$ | $0.802 \pm 0.013$ 
 $0.767 \pm 0.011$ | $0.798 \pm 0.022$ 
 $0.757 \pm 0.009$ |
| MP+hom$_4$ | $0.777 \pm 0.010$ 
 $0.774 \pm 0.019$ | $0.750 \pm 0.010$ 
 $0.749 \pm 0.021$ | $0.789 \pm 0.019$ 
 $0.772 \pm 0.01$ |
| MP+bas$_3$ | $0.840 \pm 0.010$ 
 $0.809 \pm 0.020$ | $0.847 \pm 0.017$ 
 $0.804 \pm 0.021$ | $0.863 \pm 0.039$ 
 $0.798 \pm 0.016$ |
| MP+bas$_4$ | $0.963 \pm 0.012$ 
 $0.995 \pm 0.003$ | $0.970 \pm 0.005$ 
 $0.994 \pm 0.005$ | $0.963 \pm 0.008$ 
 $0.980 \pm 0.008$ |
| MP+bas$_8$ | $0.978 \pm 0.008$ 
 $0.989 \pm 0.005$ | $0.978 \pm 0.014$ 
 $0.977 \pm 0.007$ | $0.981 \pm 0.016$ 
 $0.953 \pm 0.006$ |
| Sub-G | $0.795 \pm 0.013$ 
 $0.760 \pm 0.015$ | $0.814 \pm 0.030$ 
 $0.730 \pm 0.015$ | $0.821 \pm 0.060$ 
 $0.733 \pm 0.016$ |

(b) w/o early stopping.

| Model | Num. layers 1 | 2 | 5 |
|---|---|---|---|
| L-G | $0.928 \pm 0.009$ 
 $0.747 \pm 0.007$ | $0.827 \pm 0.006$ 
 $0.757 \pm 0.014$ | $0.993 \pm 0.003$ 
 $0.719 \pm 0.011$ |
| LF-G | $0.846 \pm 0.008$ 
 $0.773 \pm 0.013$ | $0.902 \pm 0.006$ 
 $0.745 \pm 0.010$ | $0.978 \pm 0.006$ 
 $0.730 \pm 0.018$ |
| MP | $0.849 \pm 0.004$ 
 $0.762 \pm 0.009$ | $0.909 \pm 0.011$ 
 $0.740 \pm 0.011$ | $1.000 \pm 0.000$ 
 $0.751 \pm 0.025$ |
| MP+hom$_4$ | $0.787 \pm 0.004$ 
 $0.778 \pm 0.006$ | $0.760 \pm 0.005$ 
 $0.759 \pm 0.011$ | $0.908 \pm 0.012$ 
 $0.737 \pm 0.015$ |
| MP+bas$_3$ | $0.962 \pm 0.003$ 
 $0.809 \pm 0.016$ | $0.982 \pm 0.005$ 
 $0.807 \pm 0.015$ | $0.998 \pm 0.001$ 
 $0.807 \pm 0.008$ |
| MP+bas$_4$ | $0.972 \pm 0.005$ 
 $0.994 \pm 0.004$ | $0.977 \pm 0.007$ 
 $0.991 \pm 0.006$ | $0.992 \pm 0.005$ 
 $0.973 \pm 0.003$ |
| MP+bas$_8$ | $0.982 \pm 0.004$ 
 $0.989 \pm 0.007$ | $0.994 \pm 0.003$ 
 $0.971 \pm 0.009$ | $0.998 \pm 0.001$ 
 $0.954 \pm 0.004$ |
| Sub-G | $0.918 \pm 0.004$ 
 $0.755 \pm 0.015$ | $0.878 \pm 0.011$ 
 $0.728 \pm 0.011$ | $0.992 \pm 0.004$ 
 $0.712 \pm 0.025$ |

Table 6: Train and test accuracy, Stochastic Block Model graphs. The task is to predict if the count of cycles of length at most $4$ in the cycle basis of each graph is above or below the median of the whole dataset. The node features are augmented with ($\text{hom}_N$) homomorphism-counts of cycles up to length N, ($\text{sub}_N$) subgraph-counts of cycles up to length N, and ($\text{bas}_N$) number of cycle graphs up to length N in the cycle basis.

(a) w/ early stopping.

| Model | Num. layers 1 | 2 | 5 |
|---|---|---|---|
| L-G | $0.961 \pm 0.007$ 
 $0.961 \pm 0.003$ | $0.972 \pm 0.009$ 
 $0.949 \pm 0.003$ | $0.960 \pm 0.011$ 
 $0.943 \pm 0.010$ |
| LF-G | $0.965 \pm 0.005$ 
 $0.948 \pm 0.005$ | $0.959 \pm 0.020$ 
 $0.942 \pm 0.004$ | $0.962 \pm 0.020$ 
 $0.953 \pm 0.011$ |
| MP | $0.967 \pm 0.006$ 
 $0.955 \pm 0.006$ | $0.973 \pm 0.004$ 
 $0.962 \pm 0.007$ | $0.967 \pm 0.004$ 
 $0.955 \pm 0.006$ |
| MP+hom$_4$ | $0.947 \pm 0.005$ 
 $0.953 \pm 0.006$ | $0.950 \pm 0.007$ 
 $0.946 \pm 0.006$ | $0.963 \pm 0.013$ 
 $0.953 \pm 0.004$ |
| MP+bas$_3$ | $0.966 \pm 0.007$ 
 $0.958 \pm 0.012$ | $0.958 \pm 0.007$ 
 $0.960 \pm 0.005$ | $0.973 \pm 0.012$ 
 $0.959 \pm 0.007$ |
| MP+bas$_4$ | $0.957 \pm 0.011$ 
 $0.981 \pm 0.005$ | $0.959 \pm 0.007$ 
 $0.977 \pm 0.011$ | $0.979 \pm 0.012$ 
 $0.975 \pm 0.005$ |
| MP+bas$_8$ | $0.964 \pm 0.006$ 
 $0.982 \pm 0.007$ | $0.959 \pm 0.010$ 
 $0.973 \pm 0.004$ | $0.955 \pm 0.005$ 
 $0.975 \pm 0.008$ |
| Sub-G | $0.957 \pm 0.006$ 
 $0.951 \pm 0.012$ | $0.959 \pm 0.010$ 
 $0.943 \pm 0.006$ | $0.977 \pm 0.006$ 
 $0.934 \pm 0.008$ |

(b) w/o early stopping.

| Model | Num. layers 1 | 2 | 5 |
|---|---|---|---|
| L-G | $0.968 \pm 0.003$ 
 $0.961 \pm 0.005$ | $0.981 \pm 0.002$ 
 $0.948 \pm 0.005$ | $0.998 \pm 0.001$ 
 $0.923 \pm 0.004$ |
| LF-G | $0.968 \pm 0.003$ 
 $0.957 \pm 0.002$ | $0.979 \pm 0.005$ 
 $0.941 \pm 0.006$ | $0.999 \pm 0.001$ 
 $0.937 \pm 0.005$ |
| MP | $0.973 \pm 0.006$ 
 $0.951 \pm 0.005$ | $0.975 \pm 0.003$ 
 $0.963 \pm 0.005$ | $0.991 \pm 0.002$ 
 $0.955 \pm 0.007$ |
| MP+hom$_4$ | $0.958 \pm 0.003$ 
 $0.953 \pm 0.008$ | $0.963 \pm 0.002$ 
 $0.939 \pm 0.008$ | $0.973 \pm 0.006$ 
 $0.953 \pm 0.006$ |
| MP+bas$_3$ | $0.979 \pm 0.003$ 
 $0.960 \pm 0.005$ | $0.991 \pm 0.001$ 
 $0.962 \pm 0.007$ | $0.994 \pm 0.003$ 
 $0.959 \pm 0.007$ |
| MP+bas$_4$ | $0.980 \pm 0.001$ 
 $0.989 \pm 0.003$ | $0.989 \pm 0.003$ 
 $0.984 \pm 0.003$ | $0.995 \pm 0.002$ 
 $0.971 \pm 0.010$ |
| MP+bas$_8$ | $0.982 \pm 0.004$ 
 $0.978 \pm 0.003$ | $0.995 \pm 0.001$ 
 $0.967 \pm 0.003$ | $0.999 \pm 0.001$ 
 $0.966 \pm 0.006$ |
| Sub-G | $0.970 \pm 0.006$ 
 $0.954 \pm 0.004$ | $0.975 \pm 0.002$ 
 $0.948 \pm 0.009$ | $0.994 \pm 0.003$ 
 $0.931 \pm 0.009$ |

and embed the graphs into a two-dimensional space via Multidimensional Scaling (MDS) (Kruskal, 1964). MDS embeds the graphs into a two-dimensional space while preserving the initial $\mathcal{F}_5$-Tree Mover's Distances from the raw graph space, i.e., one can formulate it as the optimization problem given by

$$\arg \min_{x_1,\ldots,x_n \in \mathbb{R}^2} \sum_{i<j} \left( \|x_i - x_j\| - \zeta\text{-TMD}(G_i, G_j) \right)^2 .$$

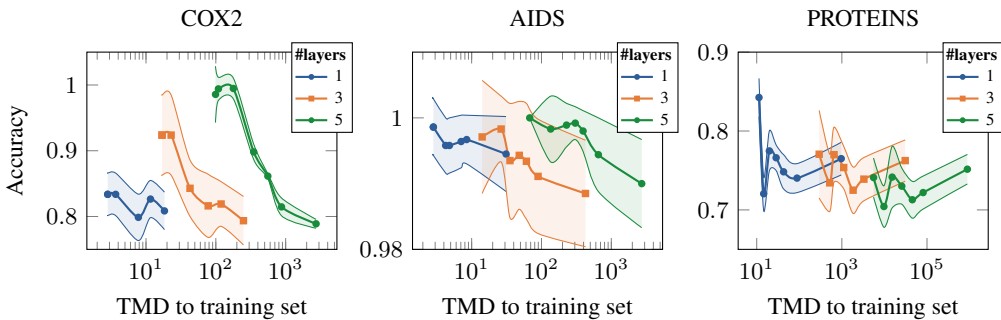

Figure 8: Accuracy of a GIN with 1, 3, and 5 layers versus Tree Mover's Distance (log scale) to the training dataset.

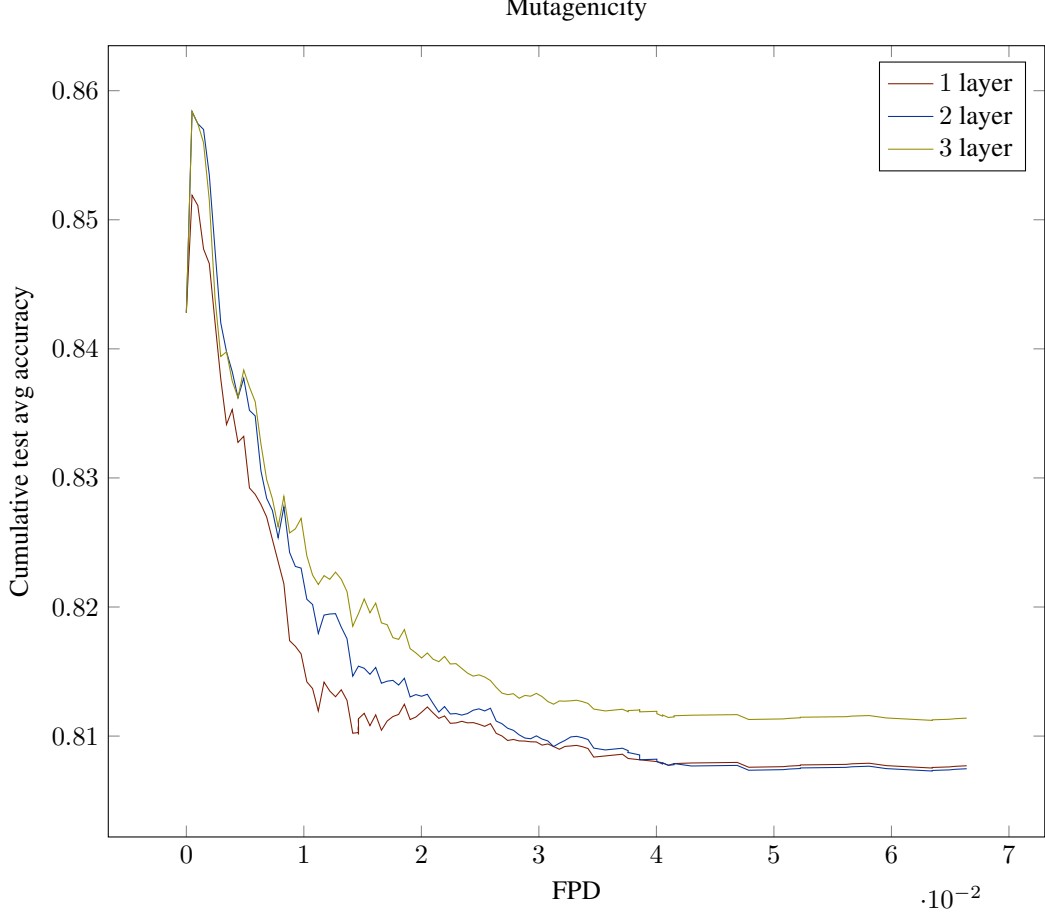

Figure 9: Cumulative test accuracy of MLP vs. Fingerprint Distance (FPD).

Labels are assigned using 2-means clustering, ensuring that structurally similar graphs remain in the same class. MDS and 2-means clustering are implemented via scikit-learn (Buitinck et al., 2013).

The experimental setup is identical to Task 1, with models trained and evaluated under the same protocol: We perform 10-fold cross validation with a training/validation/test set with 80/10/10 splits. We report the test accuracy at the epoch with the highest validation accuracy. This task tests the ability of different GNN architectures to generalize. The task is generated such that labels strongly correlate with $\mathcal{F}_5$-TMD. We present classification performance, showing that $\mathcal{F}_5$-MPNNs perform better than MPNNs and more expressive GNN variants.

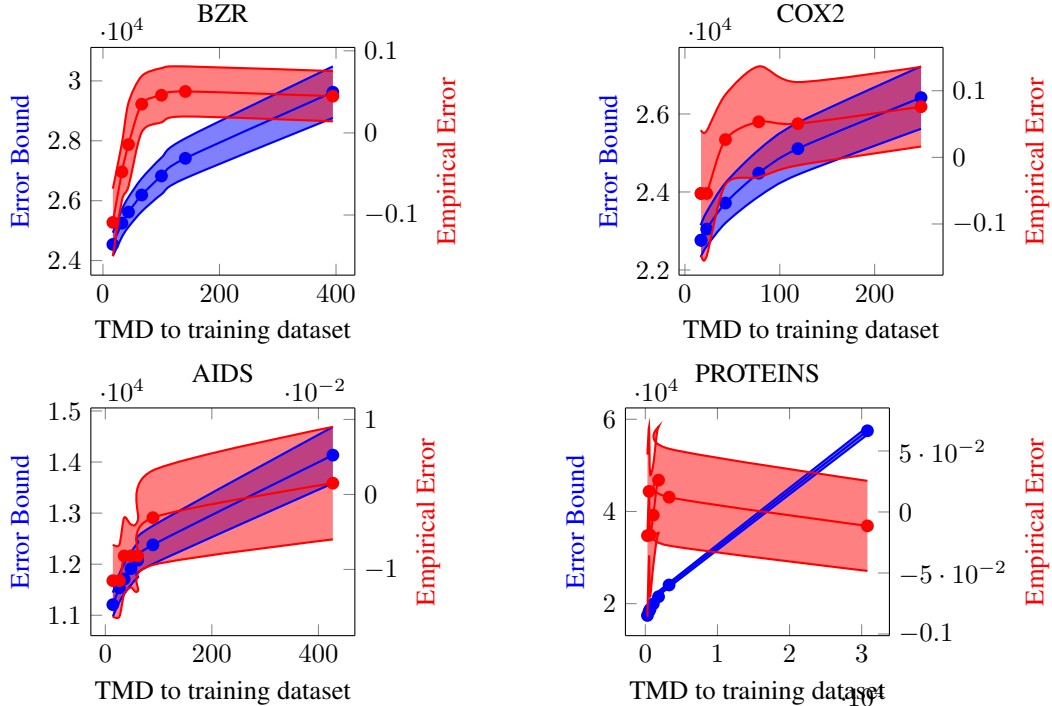

Figure 10: Error-bound curves for BZR, COX2, AIDS, and PROTEINS. Each plot shows our theoretical bound (blue, left axis) and the empirical generalization error (red, right axis) as a function of TMD to the training set. Shaded areas indicate $\pm 1$ standard deviation across 10 random splits.

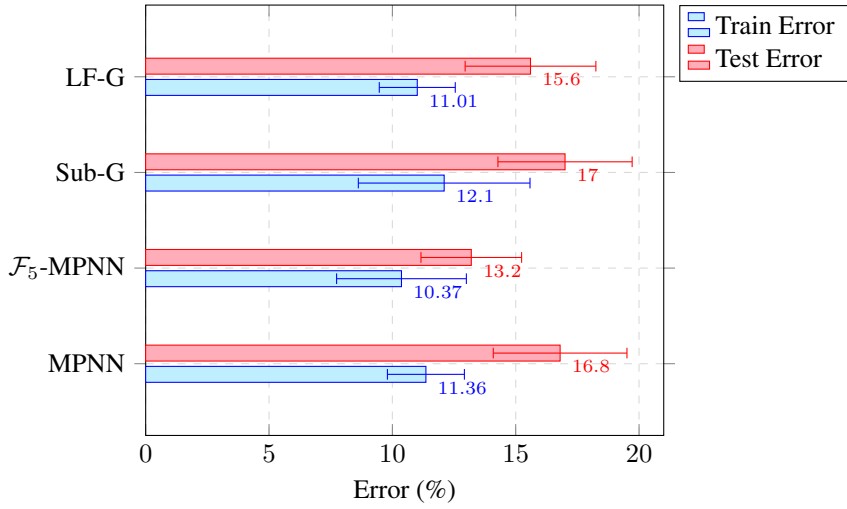

Figure 11: Performance of different GNNs for $y \sim \mathcal{F}_5\text{-TMD}^3$, where $\mathcal{F}_5$ includes cycles up to length 5. $\mathcal{F}_5$-MPNN achieves the best performance, supporting our claim that incorporating task-relevant features outperforms excessive expressivity.

