# OpenReview forum: "Graph Representational Learning: When Does More Expressivity Hurt Generalization?"
_ICLR.cc/2026/Conference — ICLR 2026 Poster_

### Official Review · Reviewer_iyDd · 2025-10-26

**Soundness:** 3
**Presentation:** 3
**Contribution:** 3
**Rating:** 6
**Confidence:** 3

**Summary:**

This paper conducts an theoretical analysis on the tradeoffs between model expressiveness in the graph neural network context and empirical performance (seen through the lens of generalization). To this end, the paper studies the generalization of GNNs vis-a-vis their expressiveness by introducing and basing their theoretical analysis on variants of a tree-movers distance, which indicates correlations between structural features learnable / detectable by models and the ultimate classification output. The paper introduces the necessary concepts early in the paper, and explains how expressiveness affects TMDs (with more expressive models increasing them, potentially harmfully to performance), and them produces a bound combining model complexity and structural similarity.

From this bound, the authors produce a simplified version that more accessibly explains the main contributors to worse generalization between training and testing. In particular, the paper highlights model complexity as a negative factor, relating this to overfitting, as well as worsening similarity between training and testing sets and worsening TMDs, which decreases model alignment.

Finally, the paper conducts an empirical analysis of GNNs of varying expressiveness against both synthetic and real-world baselines, showing empirically that expressiveness indeed offers a "sweet spot", in which too little expressiveness can undermine performance, but excessive, uncorrelated expressiveness can lead to overfitting.

**Strengths:**

- The paper's methodology is sound and its arguments and experiments are well-designed.
- The exposition of the paper is balanced, despite the highly involved nature of the proofs and of the subject matter as a whole.
- The authors do well in extracting key insights from the paper and maintaining a high-level narrative despite the high technicality of the work.

**Weaknesses:**

- The results derived in this paper need significant contextualization and connection to practical phenomena to become more beneficial to the community. Naturally, I appreciate the derivations and constructions behind the theorems being proven in this paper. However, the connection, say, between $\zeta$-TMDs as a theoretical construct and the choices of datasets and synthetic targets in the paper are not clear. I fully recognize that this is not a simple concept or connection to make, but I feel it is essential to better understand the underlying mechanics that are qualitatively being presented. It would also substantially strengthen the takeaway message of the paper.

- At a high level, the insights drawn in this paper align with intuition, but, in keeping with my point above, lack accompanying guidance or support to be more widely applicable. For instance, $\zeta$-TMDs and structural similarity are reasonable and plausible (if not perfectly quantifiable) metrics to want to estimate when selecting a model for a graph representation learning task, but it's not clear where to start. This may be a somewhat trivial exercise compared to the theoretical derivations themselves, but I feel that more curated examples with concrete choices for the pseudometrics and guidance on identifying appropriate choices more broadly would substantially increase the appeal of this work, and take it from being a theoretical result confirming widely held intuitions to the start of a framework or heuristic that informs model selection, much like algorithmic alignment [1]

- Speaking of algorithmic alignment, I feel that that nuance is missing in this work. This is not a fundamental limitation, as the work in itself is already substantial, but I do still think that a comparison with intuitive alignment arguments would be beneficial for grounding your intuitions and conclusions, and would add good color. Most prominently, adding features that align with the task being learned, e.g., cycle counting, is well explained as a performance improvement through algorithmic alignment, and is compatible with your results. However, alignment captures a complementary subtlely, namely that being able to learn cycles as an expressive model is typically worse for generalization versus having explicit cycling representations. This also lines up with your work: A more expressive model capturing cycles among other cycles would weaken the correlation with the target signal, worsening generalization, whereas an aligned cycle-only model wouldn't. I would add some discussion on this as a minor improvement to the nuance of your analysis.

[1] Xu et al., What Can Neural Networks Reason About? ICLR 2020.

**Questions:**

None, please see weaknesses above.

---

> ### Author Response · Authors · 2025-11-23
>
> We thank the reviewer for their thoughtful feedback. Below we offer our perspective on the mentioned weaknesses.
>
>
> > **W1:** “The results derived in this paper need significant contextualization and connection to practical phenomena to become more beneficial to the community.”
>
> We thank the reviewer for raising the importance of contextualizing $\zeta$-TMD with practical phenomena. Our choice of datasets and synthetic tasks is directly motivated by the theoretical role of $\zeta$-TMD: The synthetic cycle-counting task allows us to control the exact structure–label relationship and therefore isolates the expressivity-generalization trade-off implied by Theorem 5.1, while the TU benchmarks provide realistic graph-level prediction settings where structure-label correlation varies naturally across datasets. This experimental design is intentional: It allows us to empirically validate the theoretical mechanism behind the bounds.
> To reinforce this interpretability, we will expand Section 6 to more explicitly explain why these datasets were chosen and how they expose the theoretical principles. We will also integrate the $\zeta$-TMD perspective more clearly with the empirical findings, for example, by highlighting the rank correlation between $\zeta$-TMD and test error across the models (see also our response to **W5** of Reviewer cYcW) and by adding results on the ogb-molhiv dataset, where for each test graph the minimal distance to the train set is approximated using a reasonably-sized test graph-specific sample of the train set. The ogb-molhiv dataset serves as a partially OOD setting where structure shift is more pronounced.
>
> We believe that this framing will clarify how $\zeta$-TMD functions as a tool to explain performance trends that capacity-only bounds cannot capture, thereby strengthening the takeaway message of the paper.
>
> > **W2:** “At a high level, the insights drawn in this paper align with intuition [...]”
>
> We appreciate the reviewer’s comment on strengthening the practical guidance and agree that making $\zeta$-TMD selection more concrete would increase accessibility. We also acknowledge the value of theoretical work that formalizes and explains widely held intuitions: This is an important contribution in itself. However, indeed our goal is not only to confirm intuition, but to provide a principled framework that explains and predicts empirical behavior that practitioners routinely observe, grounding these intuitions in theory.
>
> The observation in [1], e.g, that increasing expressivity (primarily by adding layers in their work) improves train and test performance up to a point, after which performance saturates or even worsens, fits directly into our observations and results. In our framework, increasing expressivity initially improves the ability of a GNN to capture structure–label correlation, which possibly improves generalization. Beyond that level, additional expressivity no longer aligns with the underlying correlation structure and the bound worsens, which manifests as empirical saturation or drop-off. This matches findings such as Maskey et al. (2024) and Bouritsas et al. (2020), which show for datasets like ZINC that architectures capturing increasingly larger cycles improve performance only up to approximately 6-cycles, beyond which accuracy no longer increases. This is precisely because labels appear correlated with these structural motifs and not with ones of greater complexity.
>
> This illustrates how $\zeta$-TMD can guide model selection: One should increase expressivity only while it reduces the resulting distance of training and test graphs for a given task, rather than working under the assumption that “more expressivity is always better.”
>
> Maskey et al. (2024). *Weisfeiler–Leman Go Loopy: A New Hierarchy for Graph Representational Learning.* NeurIPS.
>
> Bouritsas et al. (2020). *Improving Graph Neural Network Expressivity via Subgraph Isomorphism Counting.* TPAMI.

---

> > ### Author Response · Authors · 2025-11-23
> >
> > > **W3:** “Speaking of algorithmic alignment, I feel that that nuance is missing in this work. This is not a fundamental limitation, as the work in itself is already substantial, but I do still think that a comparison with intuitive alignment arguments would be beneficial for grounding your intuitions and conclusions, and would add good color [...]”
> >
> > We appreciate the suggestion to relate our work more explicitly to algorithmic alignment. Conceptually, our framework is very much in the same spirit as Xu et al. (2020): In some sense they posit an underlying *label-generating algorithm* and show that networks whose computation graphs align with this algorithm enjoy better sample complexity; we posit an underlying *label-generating mechanism that is Lipschitz w.r.t. a task-aligned pseudometric* $\zeta$, and show that GNN architectures aligned with $\zeta$ admit tighter, correlation-aware generalization bounds.
> >
> > In algorithmic alignment, the “ideal” architecture is the one whose modules can simulate the true reasoning algorithm via simple local steps, rather than having to learn complex control flow (e.g. for-loops) inside a generic MLP. Our $\zeta$-TMD view is the distributional analogue of this: we assume labels are generated by a (possibly unknown, stochastic) process that is smooth in $\zeta$, and show that GNNs that preserve small $\zeta$-distances between graphs that share labels are “ideally aligned” in the sense of minimizing our correlation-aware bound. Overly expressive architectures that break this alignment (by being able to represent functions that are not or less $\zeta$-smooth) worsen the bound, just as overexpressive but poorly aligned architectures in (Xu et al., 2020) require more samples to simulate the same algorithm. This is indeed nicely illustrated by the example you provide on a “cycle-only model” and another model that is able to learn cycles among other patterns.
> > Thus, both frameworks point to the same qualitative message: *the right notion of alignment is not “maximum expressivity”, but matching the structure of the label-generating mechanism*. We will add a paragraph in the discussion explicitly drawing this parallel.

---

### Official Review · Reviewer_wMWG · 2025-10-31

**Soundness:** 3
**Presentation:** 3
**Contribution:** 2
**Rating:** 4
**Confidence:** 3

**Summary:**

The paper studies when higher expressivity in Graph Neural Networks improves or harms generalization performance. It introduces a family of pseudometrics called ζ Tree Mover Distances that quantify structural similarity between graphs at varying expressive levels, and uses these metrics to model structure–label correlation. The authors derive data dependent generalization bounds that decompose into a model capacity term and a structural similarity term, showing that increased expressivity can worsen generalization when it does not align with task relevant structure. Empirically, they demonstrate that moderately expressive GNNs outperform both vanilla MPNNs and highly expressive higher order GNNs on synthetic and real datasets when label signals correlate with specific structural features. Their results show that accuracy decreases as test graphs become structurally farther from the training graphs under the relevant pseudometric, matching the theoretical predictions.

**Strengths:**

- The paper tackles a central open problem in graph learning: understanding when increased GNN expressivity improves versus harms generalization. This is a core theoretical theme highlighted in recent GNN survey and workshop discussions, and the paper makes progress on a topic with clear community interest.
- The ζ-TMD construction formalizes the intuition that model expressivity should match task relevant structural signals. By connecting GNN expressivity to pseudometrics that capture structural similarity, the paper provides a structured way to reason about alignment between graph structure and supervision.
- The paper demonstrates cases where moderately expressive models outperform both simpler and more expressive architectures, consistent with the theory. The experiments, though modest in scale, validate the predicted trade-off between expressivity and structure–label alignment and show trends consistent with the generalization bounds.

**Weaknesses:**

- Although the paper proposes a ζ-TMD framework, the incremental contribution over closely related analyses of structure label alignment and robustness in recent literature (for example Ma et al., 2021; Li et al., 2024; Vasileiou et al., 2024) is not entirely explicit. The manuscript would benefit from a sharper articulation of what new theoretical insight is enabled beyond these frameworks, particularly given that alignment based perspectives have been explored previously.
- The approach assumes access to an appropriate ζ that correlates with label structure. However, identifying such a metric in practice is nontrivial, and the paper does not provide a principled procedure to select or estimate it. This limits practical applicability, especially when the relevant structural signal is unknown.
- Experiments are conducted primarily on synthetic tasks and small scale TU datasets. These benchmarks do not fully stress the claimed benefits, and are not representative of modern large scale or heterophilic graph learning settings. Validation on more challenging datasets such as OGB or heterophilic graphs would strengthen empirical support.
- The generalization bounds are informative but rely on constants and assumptions that are difficult to estimate or control, which limits actionable guidance for model selection in practice. The paper frames the results as qualitative, but clearer direction on how practitioners should use the theory would increase impact.
- The narrative focuses on expressivity alignment, but does not explicitly connect to orthogonal failure modes such as over-smoothing, over-squashing, and information bottlenecks in deep GNNs. Clarifying how the proposed theory relates to or differs from these phenomena would improve conceptual clarity.

**Questions:**

- In practice, how should one determine the correct pseudometric ζ for a given task? Does the proposed theory offer any algorithmic method or diagnostic criteria to identify such a metric when the relevant structural signal is not known a priori?
- Why are the empirical evaluations limited to synthetic tasks and relatively small TU datasets? Can the authors provide results on larger and more challenging graph benchmarks such as OGB or heterophilic datasets to demonstrate generality and real world relevance?
- How does this framework relate to other established causes of generalization failure in GNNs, such as over smoothing, over squashing, and information bottlenecks? Are these phenomena orthogonal to the expressivity alignment perspective, or can the proposed theory explain or incorporate them?

---

> ### Author Response · Authors · 2025-11-23
>
> We thank the reviewer for their thoughtful feedback. Below we offer our perspective on the mentioned weaknesses and address the specific questions.
>
> > **W1**: “Although the paper proposes a ζ-TMD framework, the incremental contribution over closely related analyses of structure label alignment and robustness in recent literature (for example Ma et al., 2021; Li et al., 2024; Vasileiou et al., 2024) is not entirely explicit. [...]”
>
> We clarify in our manuscript that Ma et al. (2021) also examine structure–label alignment, but only in a significantly more restricted setting. As stated in Lines 142-144 in the related work section: “The work most closely related to ours is Ma et al. (2021), which models label–feature alignment but does not support graph-level tasks or trainable GNNs.” In contrast, our framework explicitly models structure–label correlation via task-aligned pseudometrics and applies to expressive, end-to-end trainable GNNs, enabling analysis across varying architectural expressivity. This allows us to derive correlation-aware bounds that align with empirical trends more strongly than capacity-only bounds and that also remain valid for modern expressive architectures.
>
> We also discuss additional related works in that section (Section 1.2) and in the extensive Appendix A, clarifying how our contributions extend beyond robustness perspectives (e.g., Li et al., 2024; Vasileiou et al., 2024) by providing a generalization framework rather than solely stability results.
>
> > **W2 & Q1:** “The approach assumes access to an appropriate ζ that correlates with label structure. [...]”
>
> The assumption that there exists a $\zeta$ capturing structure-label correlation is not a limitation of our approach but rather the core premise that makes generalization analysis possible. As we motivate in the introduction, without some form of label–data correlation, generalization is fundamentally impossible, i.e., a model cannot generalize from training data if labels are arbitrary. Our framework makes this assumption explicit through a task-aligned pseudometric. Importantly, our way of modeling data–label correlation is very general. It can be viewed simply as assuming the existence of a representation space in which classes concentrate around centers, a standard and intuitive assumption in practice. This makes the $\zeta$-based correlation assumption broad and widely applicable rather than restrictive.
>
> Rather than constraining applicability, the framework explains existing practical behavior: It shows why increasing expressivity is beneficial only when the model aligns with the task-relevant structure, and harmful otherwise. Our experiments (Section 6) illustrate that $\zeta$-alignment strongly predicts generalization trends across multiple expressivity levels, supporting the practical relevance of the assumption.
>
> We will clarify this point in the revision.
> > **W3 & Q2:** “Experiments are conducted primarily on synthetic tasks and small scale TU datasets. [...]”
>
> We appreciate the suggestion regarding larger-scale benchmarks. We are currently running experiments on OGBG-MOLHIV,  where for each test graph the minimal distance to the train set is approximated using a reasonably-sized test graph-specific sample of the train set. We will include the results in the revised version, as this dataset provides a meaningful partially-OOD scenario that aligns well with our structural-shift perspective. Thank you for highlighting this direction.
>
> Regarding heterophilic graphs: heterophily refers to node-level label dissimilarity (i.e., neighbors having different labels) in node classification settings, and does not correspond to structure–label correlation measured via graph-level pseudometrics. Our work focuses on graph-level prediction, where distances between entire graphs determine generalization behavior, rather than neighbor-level label agreement. Therefore, heterophily is orthogonal to our theoretical setting; see Lim et al. (2021), *Large-Scale Learning on Non-Homophilous Graphs*, for a clear discussion of heterophily in the node-classification context.
>
> While we believe that our analysis can be generalized to node-level setting, this is currently out-of-scope.

---

> > ### Author Response · Authors · 2025-11-23
> >
> > > **W4:** “The generalization bounds are informative but rely on constants and assumptions that are difficult to estimate or control, [...]”
> >
> > We agree that the bounds contain constants that are challenging to estimate exactly. However, the actionable component of our theory is not the absolute numerical value of the bound, but the relationship between generalization performance and the structural distance to the training graphs. Both our theoretical results and empirical findings demonstrate that $\zeta$-alignment strongly correlates with test performance, providing a practical diagnostic that capacity-only bounds cannot capture.
> >
> > The resulting guidance for practitioners is intuitive and consistent with established modeling practice: Select architectures whose inductive bias aligns with the domain structure, and avoid overshooting expressivity, as excessively expressive models generalize poorly, particularly under distribution shift. Our experiments across 13 expressivity levels empirically confirm this principle. We also confirm this via calculating correlations, see our response to W5 of Reviewer cYcQ.
> >
> > > **W5 & Q3:** “The narrative focuses on expressivity alignment, but does not explicitly connect to orthogonal failure modes such as over-smoothing, over-squashing, and information bottlenecks in deep GNNs. [...]”
> >
> > Oversmoothing, over-squashing, and information bottlenecks are phenomena primarily analyzed in node-level tasks. In graph-level prediction, these effects are not known to be nearly as problematic; recent work shows that highly smoothing GNNs can perform strongly on graph-level benchmarks. For example, Tönshoff et al. (2023), *Where Did the Gap Go? Reassessing the Long-Range Graph Benchmark*, demonstrate that simple, strongly smoothing GNNs achieve competitive performance, challenging the conventional narrative around oversmoothing in graph-level settings. There also is an ongoing discussion in the community at the moment, as illustrated for example by Arnaiz-Rodríguez and Errica, *Oversmoothing, “Oversquashing”, Heterophily, Long-Range, and more: Demystifying Common Beliefs in Graph Machine Learning*, on the importance and correct categorization of the mentioned phenomena. While this discussion is ongoing, it is somewhat unappealing to mix these ideas with our separate study on the generalization ability of expressive GNNs.
> >
> > Thus, these mechanisms are orthogonal to the failure mode our framework targets. Our focus is explicitly on graph-level generalization, modeled via structure–label correlation and expressivity alignment, rather than node-level information compression dynamics.
> >
> > ---
> > We hope these clarifications address the concerns raised and demonstrate both the novelty and relevance of our contribution. We kindly ask the reviewer to reconsider the score in light of the additional evidence, and we would be grateful for any further questions or suggestions.

---

> > ### Comment · Reviewer_uzRa · 2025-11-26
> >
> > Thank authors for the clarification.
> >
> > W1:
> >
> > To me the differences between the mentioned works are still kind of vague. Could you elaborate more on the technical differences?  For example, what are their technical limitations and how this work address them?
> >
> > W2:
> >
> > Do you mean that  ζ is impossible to estimate in practice?  Does real-world datasets exhibit the correlation-pattern proposed in this paper?
> >
> > W3:
> >
> > Why not conduct experiments on real-world dataset?

---

> ### Author Response · Authors · 2025-11-27
>
> Dear Reviewer uzRa,
>
> Thank you very much for your fast response and active engagement in the discussion. We are more than happy to provide further explanation concerning your questions. We want to highlight that we furthermore provided responses to your review above. We hope that the above responses, together with our further clarifications below address your remaining concerns.
>
> > **W1:** Technical differences and limitations of prior work.
>
> Our goal in this work is to analyze end-to-end trainable GNNs across a range of expressivities, and to study generalization when the graph structure itself (not only aggregated node features) correlates with the labels. Addressing this setting requires reasoning directly in graph space, where distances reflect structural similarity between graphs rather than distances between fixed embeddings as in (Ma et al., 2021).
>
> To enable this, we introduce pseudometrics over graphs ($\zeta$-TMD), which provide the necessary generality to capture structural properties beyond only node attributes and allow our framework to model many families of expressive GNNs. This makes it possible to analyze structure-label correlation at the graph level and derive bounds that depend explicitly on architectural expressivity. This is one central question our work addresses.
>
> In contrast, prior analyses such as Ma et al. (2021) operate under a fixed, non-learnable encoder and reduce graph comparison to $l_2$​-distance in a Euclidean latent space (after aggregating node features). This means that graphs can only be compared via simple statistics such as sums or means of features, and no structural information beyond features is considered. As a result, those approaches rely on standard Euclidean PAC-Bayes theory, rather than requiring analysis over graph pseudometrics or expressive, trainable GNN architectures.
>
> Our setting is significantly more technically challenging, because we must perform a perturbation analysis of MPNNs in graph space,: i.e., understanding how predictions change under perturbations of the input graph with respect to $\zeta$-TMDs, and performing PAC-Bayes analysis under perturbations of all trainable GNN layers, rather than only perturbing a classifier head. The technical development of these results is detailed in Lemma C.6 and Section E.2.
>
> We will revise the paper to make these distinctions more explicit. We furthermore want to mention our three-page Appendix A in which we extensively discuss related work. The further clarification we worked out with you here will significantly strengthen this discussion.
>
> > **W2:** ”Do you mean that ζ is impossible to estimate in practice? Does real-world datasets exhibit the correlation-pattern proposed in this paper?”
>
> No, the underlying pseudo-metric is not impossible to estimate in practice. In fact, in our response to W4 of Reviewer cYcW we sketch an estimation scheme for $\zeta$-TMDs. In the response you react to here, we rather tried to make the point that identifying the *optimal* $\zeta$ is inherently difficult, much like identifying the optimal GNN architecture, simply because there are infinitely many possible pseudometrics. In practice, however, one does not need the optimal choice: the $\zeta$-TMDs we considered already provide meaningful structure–label correlation. Our experiments indicate that these choices yield high Spearman/Kendall correlations (see response to W5 of Reviewer cYcW) between $\zeta$-TMD distance and test error, suggesting that real-world datasets do exhibit the type of correlation pattern modeled in our theory.
>
> > **W3:** “Why not conduct experiments on real-world datasets?”
>
> We believe there may be a misunderstanding: Our experiments already include real-world datasets, as shown in Figures 1–6, 8, 9, and 10. Additionally, as noted earlier, we are currently completing experiments on ogbg-molhiv, and will include the results in the revised version. Providing empirical results on 7 real-world datasets is significant for a paper with a strong theoretical focus like ours.

---

### Official Review · Reviewer_uzRa · 2025-11-01

**Soundness:** 3
**Presentation:** 3
**Contribution:** 2
**Rating:** 4
**Confidence:** 3

**Summary:**

This paper studies when increased GNN expressivity helps or hurts generalization by introducing ζ-Tree Mover Distances (ζ-TMDs), task-aligned pseudometrics built via strong simulation of color refinement algorithms. It proves ζ-MPNNs are Lipschitz w.r.t. ζ-TMD, then derives PAC-Bayes bounds that decompose into a capacity term and a structural similarity term (distance between train/test under ζ-TMD). The analysis shows that expressivity improves generalization only when it enhances structure–label alignment; otherwise, it can worsen the bound. Empirically, synthetic and TU datasets support the theory and illustrate tighter (though still loose) bounds than standard PAC-Bayes.

**Strengths:**

Clear metric → stability → generalization pipeline with non-trivial derivations. he paper defines task-aligned pseudometrics ζ-TMDs (Def. 3.1; Prop. 3.2–3.3), proves Lipschitz stability of ζ-MPNNs with respect to ζ-TMD (Theorem 3.4; e.g., “∥h(G) − h(H)∥ ≤ L · ζ-TMDT+1(G, H)”), and derives PAC-Bayes generalization bounds that separate a capacity term from a structural-similarity term (Theorem 4.1 and its simplified Eq. (4)). These steps connect distance between graphs → Lipschitz robustness → generalization in a coherent chain and require several technical lemmas (Appendix E, e.g., Lemmas E.2–E.5). The framework extends beyond standard MPNNs via strong simulation (Appendix B), covering F-MPNNs (Cor. 3.5) and k-GNNs (Cor. C.9).

ζ-TMD extends the TMD framework to strongly simulatable color refinements, sharpening the bounds via task-aligned structure–label correlation. The ζ-TMD pseudometrics let the bound depend on structure–label alignment, not just model capacity: “generalization improves when the model maps structurally similar graphs (with respect to the ζ-TMD that correlates with the labels) to similar representations while keeping model capacity in check” (Sec. 4; Eq. (4)). This is a principled and flexible way to encode task-relevant structure.

When more expressivity helps vs. hurts is theoretically clarified and empirically supported. heorem 5.1 shows that moving from an under-expressive F′-MPNN to a task-aligned F-MPNN preserves the bound, whereas going beyond the task (to F˜-MPNN) can worsen the structural term since ξF˜ ≥ ξF (Lemma G.1), yielding a looser bound. This matches the experiments (Fig. 1 left; Table 1), where overly expressive models overfit. The paper even articulates the design principle: “the optimal GNN is precisely as expressive as required… no more, no less” (Sec. 5). Tighter-than-standard PAC-Bayes: the authors report their bounds are “significantly tighter,” often orders of magnitude smaller than standard PAC-Bayes bounds (e.g., 10^16 vs. 10^4; Sec. 6). To be precise, this is an orders-of-magnitude improvement (not necessarily “exponential” in a formal sense).

**Weaknesses:**

Choosing ζ is nontrivial and can be the bottleneck; once ζ is fixed, several proofs become mechanical.  The main benefit hinges on picking a ζ that actually aligns with labels. The authors acknowledge this: “the relevant pseudometric is often unknown and possibly expensive to compute” (Limitations), and Rζ can inflate size/degree/features (Eq. (4) discussion: “model complexity may increase if the graph transformation Rζ enlarges…”). Since many results follow from strong simulation plus known Lipschitz/OT arguments, the conceptual challenge is predominantly in selecting/designing ζ.

The “increasing expressivity hurts generalization” conclusion rests on upper bounds and monotone dependence on ξ, not a tight characterization.  Theorem 5.1 compares upper bounds; it shows that a more expressive model can have a larger ξ term (ξF˜ ≥ ξF), hence a looser bound, but this is not a lower-bound or necessity result. Comparing two upper bounds does not prove a performance drop must occur—only that it may, given ξ and capacity growth. A converse or data-dependent lower bound (or a matching rate) would strengthen the claim.

Bound tightness and trend-shape mismatch in the empirical validation; limited breadth of alternatives. Fig. 3 shows the bound tracks trends but is very loose in scale (theoretical curves on the order of 10^4 vs. empirical errors below 1—i.e., ~4 orders of magnitude gap). Moreover, as TMD-to-train grows, empirical error often saturates/plateaus, whereas the bound’s dependence on ξζ is monotone and cannot capture this flattening. The paper notes the bounds are “qualitative guidelines rather than precise estimates” (Limitations), but some calibration (e.g., data-dependent Lipschitz constants, local smoothness, or normalized structural terms) would help. The empirical scope, while decent (synthetic + six TU datasets), is still limited relative to alternatives: few large-scale graphs, no runtime/compute study for ζ-TMD, and limited comparison to other distance-based theories (e.g., graphon cut metrics, Gromov–Wasserstein) or strong baselines beyond the selected GNN family variants. Ablations on the choice of ζ (how sensitive are results to different CRAs?) would also strengthen the evidence. Practitioner guidance is limited because the bounds are quite loose (often several orders of magnitude above empirical error). Adding one or two strong, actionable experiments—e.g., a ζ-selection/augmentation protocol that measurably improves test accuracy by reducing ξζ, or a procedure to choose model expressivity based on estimated ζ-TMD alignment—would better demonstrate practical utility.

**Questions:**

Please refer to weakness.

---

> ### Author Response · Authors · 2025-11-23
>
> We thank the reviewer for their thoughtful feedback. Below we offer our perspective on the mentioned weaknesses and then address the specific questions.
>
> > **W1**: “Choosing ζ is nontrivial and can be the bottleneck; once ζ is fixed, several proofs become mechanical. [...]“
>
> We respectfully disagree with the suggestion that the main challenge “reduces to choosing $\zeta$” and that, once $\zeta$ is fixed, the rest becomes mechanical. Formalizing task-aligned pseudometrics is exactly the core contribution of this paper: We show that generalization depends not only on model capacity but also on structural alignment, captured by $\zeta$-TMD. This perspective enables us to formally explain when increased expressivity helps or hurts, which has been an open question in graph ML.
>
> Choosing $\zeta$ is not a weakness but is instead shown by us to be an inherent part of model design and similar in difficulty to choosing depth, k-WL order, or motif sets. Our framework provides a principled way to reason about this, instead of the current trial-and-error tendency to push expressivity as high as computationally feasible by default. We provide practical guidance (start simple, increase expressivity gradually, use validation-based diagnostics).
>
> So rather than a bottleneck, $\zeta$ is the conceptual bridge that connects expressivity, structure, and generalization: this is where the novelty of our work lies.
>
> > **W2:** “The “increasing expressivity hurts generalization” conclusion rests on upper bounds and monotone dependence on ξ, not a tight characterization. Theorem 5.1 compares upper bounds; it shows that a more expressive model can have a larger ξ term (ξF˜ ≥ ξF), hence a looser bound, but this is not a lower-bound or necessity result. Comparing two upper bounds does not prove a performance drop must occur—only that it may, given ξ and capacity growth. A converse or data-dependent lower bound (or a matching rate) would strengthen the claim.”
>
> We would like to clarify that we do not claim “increasing expressivity hurts generalization” in general. Our result is more nuanced: We identify scenarios where increased expressivity reduces structural alignment (captured via $\zeta$-TMD), which in turn loosens the generalization bound beyond the capacity-only terms. This provides a concrete explanation for why expressivity can sometimes harm performance: not because expressivity itself is bad, but because it can break task-relevant similarity when the added expressivity is not aligned.
>
> In contrast, we prove that structural similarity upper-bounds the generalization gap and we empirically validate this relationship, observing strong negative rank correlations (see **W5** in our response to Reviewer cYcW) between $\zeta$-TMD distance and accuracy across datasets. This supports our central claim: Expressivity is beneficial only when it improves alignment, not by default.
>
> We agree that lower bounds on the generalization gap would also be of interest. However, these are tougher to prove, less common in our literature and often require stronger assumptions than we make. They are therefore, beyond the scope of our already 48-page submission.

---

> > ### Author Response · Authors · 2025-11-23
> >
> > > **W3:** “Bound tightness and trend-shape mismatch in the empirical validation [...]”
> >
> > We agree that the absolute scale of the bound may be loose and explicitly note in the paper that it is intended as qualitative guidance rather than a calibrated estimator. However, the relevant comparison is not between $10^4$ and empirical error $<1$, but between our bound and prior PAC-Bayes graph bounds, which are on the order of $10^16$ (Liao et al., 2021). Our correlation-aware analysis tightens these bounds by 12 orders of magnitude while additionally modeling structure–label alignment and architectural expressivity. We will highlight this comparison more clearly in the revision.
> >
> > Regarding the plateau effect, this behavior is expected and arises from the empirical setup: the plots show cumulative accuracy/error as increasingly distant test graphs are added (as described in Line 2347). Once nearly all test graphs beyond a certain $\zeta$-TMD threshold are included, only very few additional graphs remain, and these tend to have similar structural discrepancy. As a result, the empirical curve naturally stabilizes because the dataset is effectively exhausted in that region.
> > This behavior does not contradict our theory. Our bound is a worst-case guarantee and makes no assumptions about the empirical distribution of test graphs across distance buckets. Therefore, it increases monotonically with structural discrepancy by design.
> >
> > Thank you for pointing this out, we will add this important detail to the Figure captions and main paper.
> >
> > With respect to alternative distance metrics, we note that comparisons to graphon cut distances or Gromov–Wasserstein are not conceptually aligned with our setting. We do not use graphons anywhere in our methods or experiments; our analysis is conducted on finite graphs, whereas graphon cut metrics are defined in the graphon limit and rely on dense-graph asymptotics. If you are instead referring to Graph Edit Distance, the finite-graph analogue of the cut distance, we note that its computation is NP-hard, making it highly impractical for our purposes. Consequently, such comparisons are not computationally realistic or theoretically meaningful for our framework.
> >
> > Our use of $\zeta$-TMD is deliberate: it is efficiently computable on finite graphs and directly tied to WL expressivity and CRAs, making it the appropriate metric for both our theory and empirical validation. We will add runtime measurements and an OGB experiment to broaden the empirical scope and strengthen practical utility.
> >
> > Our empirical evaluation is broader than suggested: it includes a controlled synthetic task, six TU datasets (BZR, MUTAGENICITY, NCI109, PROTEINS, AIDS, COX2), and fixed-encoder studies in Appendix I, not only the three datasets highlighted in Figure 3. This scope is consistent with prior theoretical work on expressivity and generalization, and it isolates architectural effects cleanly. We also evaluate 13 different expressivity levels, providing extensive $\zeta$-ablations (MPNN, F-MPNN variants, Sub-G, L-G, LF-G, etc.).
> >
> > We will add a runtime table for $\zeta$-TMD (per-dataset compute cost). We will report results on ogb-molhiv where for each test graph the minimal distance to the train set is approximated using a reasonably-sized test graph-specific sample of the train set. These experiments are currently running and particularly interesting because it provides a clearer OOD perspective compared to TU datasets. Thank you for the suggestion.
> >
> > Runtimes of TMD calculations for the exact pairwise distances over the whole dataset in a single process and in 128 parallel processes. All provided values are extrapolated from the average runtime observed per pair of graphs on a sample from each dataset.
> >
> > ### 2 Layer
> > | Dataset | Avg time / pair (s) | 1 process (h) | 128 processes (h) |
> > |--|--|--|--|
> > | MUTAG | 0.079 | 0.07 | 0.00 |
> > | Mutagenicity  | 0.160 | 75.35 | 0.59 |
> > | NCI109 | 0.205 | 87.44 | 0.68 |
> > | PROTEINS | 0.315 | 9.80 | 0.08 |
> > | BZR | 0.245 | 1.02 | 0.01 |
> > | COX2 | 0.330 | 1.81 | 0.01 |
> > | AIDS | 0.037| 3.65 | 0.03 |
> > | ogbg-molhiv | 0.113 | 4265.98 | 33.33 |
> >
> >
> > ### 4 Layer
> > | Dataset | Avg time / pair (s) | 1 process (h) | 128 processes (h) |
> > |--|--|--|--|
> > | MUTAG | 0.205 | 0.18 | 0.00 |
> > | Mutagenicity  | 0.564 | 265.22 | 2.07 |
> > | NCI109 | 0.491 | 209.31| 1.64 |
> > | PROTEINS | 0.657 | 20.47 | 0.16 |
> > | BZR | 0.617 | 2.56 | 0.02 |
> > | COX2 | 0.920 | 5.04 | 0.04 |
> > | AIDS | 0.150 | 14.98 | 0.12 |
> > | ogbg-molhiv | 0.425 | 15965.21 | 124.73 |
> >
> > ### 6 Layer
> > | Dataset | Avg time / pair (s) | 1 process (h) | 128 processes (h) |
> > |--|--|--|--|
> > | MUTAG | 0.319 | 0.28 | 0.00 |
> > | Mutagenicity | 0.804 | 378.32 | 2.96 |
> > | NCI109 | 0.684 | 291.52 | 2.28 |
> > | PROTEINS | 1.904 | 59.28 | 0.46 |
> > | BZR | 1.032 | 4.28 | 0.03 |
> > | COX2 | 1.560 | 8.55 | 0.07 |
> > | AIDS | 0.207 | 20.73 | 0.16 |
> > | ogbg-molhiv | 0.649 | 24384.49 | 190.50 |
> >
> > Liao et al. (2021). *A Pac-Bayesian Approach to Generalization Bounds for Graph Neural Networks.* ICLR.

---

### Official Review · Reviewer_cYcW · 2025-11-02

**Soundness:** 3
**Presentation:** 2
**Contribution:** 2
**Rating:** 6
**Confidence:** 3

**Summary:**

This paper explores the complex relationship between the expressivity of Graph Neural Networks (GNNs) and their generalisation. It suggests that generalisation depends on model capacity and the alignment between task labels and graph similarity. To capture this, the authors introduce a family of pseudometrics called ζ-Tree Mover Distances (ζ-TMDs), which extend the Tree Mover’s Distance by using “strongly simulatable” colour refinement algorithms to transform graphs before distance computation.

Theoretical results show that ζ-MPNNs (MPNNs consistent with a given CRA ζ) are Lipschitz with respect to ζ-TMD and yield PAC-Bayes based generalisation bounds. The test loss decomposes into a capacity term involving layer widths, spectral norms, and graph degree, and a structural similarity term that measures the distance between the training and test sets under ζ-TMD.

The paper emphasises that “more expressivity” can have both positive and negative effects. Increased expressivity improves performance if it reduces structural similarity term (i.e., better label-structure alignment), but it can degrade otherwise due to added capacity without alignment gains. The paper supports this through synthetic cycle-counting tasks and several TUDatasets, showing that moderately expressive, task-aligned models (e.g., F4-MPNN) generalise best, and empirical errors increase with the TMD distance from the training graphs, tracking the bound’s qualitative trend.

**Strengths:**

- The idea of task-aligned pseudometrics for graphs, formalized as ζ-TMDs, is novel and compelling. Extending TMD through strong simulation of CRAs unifies diverse expressive GNNs (k-GNNs, subgraph models, motif-augmented MPNNs) under a single stability/robustness lens.

- Proofs are extensive and careful, showing Lipschitz continuity of ζ-MPNNs and deriving PAC-Bayes style bounds where the generalization gap depends on $\xi_\zeta$. The synthetic and real-world experiments are well targeted to the claims (memorization under label noise, alignment-driven generalization, expressivity vs. overfitting).

- Definitions of CRAs, strong simulation, and ζ-TMD are clearly laid out. The decomposition of the bound into “capacity” and “structural similarity” terms makes the trade-offs intuitive. The paper situates itself well among WL-expressivity, graphon bounds, and robustness literature.

**Weaknesses:**

- Identifying the right ζ (task-aligned pseudometric) can be hard:
The framework presumes labels are “strongly correlated” with a chosen $\zeta-TMD$. In practice, discovering the right ζ is nontrivial and can be expensive (e.g., motif counts, product graphs).

- The paper assumes the CRA to be strongly simulatable, but did not mention under what condition is a CRA strongly simulatable, some examples would be good.

- Bound looseness and constants:
The PAC-Bayes bound involves many constants (widths, spectral norms, degrees) and uses assumptions (e.g., Assumption D.1) that can be stringent. The bound is qualitative and not tight.

- Computational overhead of ζ-TMD and strong simulation:
Many expressive CRAs require transformations (e.g., k-tuple product graphs) that scale poorly, and computing TMD itself is nontrivial. This, together with the looseness of the bound, greatly hinders the practicality of the results.

- Experimental scope:
 The authors empirically shows the bound demonstrates similar trend as empirical error in Fig 3, but this is not quantitative. If possible, I would like to see the correlation between the bound and empirical error on a wider range of datasets.

- Presentation polish:
Some minor issues (placeholder captions, typographical glitches, compressed figure axes) impact readability, e.g.
  * missing caption is Fig 10
  * P in L157 is not defined.
  * L165-167 "We say that a graph invariant ζ is more expressive than another graph invariant θ if ζ(G) = ζ(H) implies θ(G) = θ(H) for all G, H ∈ G." This doesn't feel right, if ζ and θ have the same expressivity, then ζ(G) = ζ(H) implies θ(G) = θ(H), according to this definition, ζ is more expressive than θ?

**Questions:**

1. How would you recommend practitioners choose ζ in real tasks?

2. What is the complexity of computing ζ-TMD at depth T for the CRAs considered? Can you provide scalable approximations (e.g., subsampling rooted trees, contrastive encoders trained to approximate TMD distances)?

3. Assumption and robustness:
How sensitive are your bounds and empirical trends to violations of $y \sim \zeta-TMD^{T+1}$ and Assumption D.1? Could you adopt alternative robustness frameworks to relax or complement D.1?

4. Does your Lipschitz analysis extend to attention-based graph transformers or alternative poolings (e.g., mean/max, Set2Set)? What constraints would be required?

5. Empirical alignment tests:
Could you add a diagnostic that measures the correlation between label distributions and ζ-TMD distances to training graphs, to operationalize “alignment” before model design?

6. Experiments: in Fig 2 and 3, the accuracy/empirical error increases/stabilises after certain TMD, can you think of any reasons?

---

> ### Author Response · Authors · 2025-11-23
>
> We thank the reviewer for their thoughtful feedback. Below we offer our perspective on the mentioned weaknesses and then address the specific questions.
>
> > **W1**: “Identifying the right ζ (task-aligned pseudometric) can be hard”
>
> Identifying the optimal task-aligned pseudometric is indeed as hard as identifying the optimal GNN: solving either would essentially solve graph ML model selection. Our contribution is therefore not a “silver bullet selector,” but a principled design lens: (i) avoid overshooting expressivity, and (ii) start from cheap, robust ζ (e.g., degree, short cycles) before escalating to more expensive CRAs. This turns the theoretical results into a practical, low-overhead procedure and counters the prevalent “more expressivity by default” trend, in which more expressivity is often implied to correspond to better performance.
>
> Beyond that, our results help explain a range of empirical observations. For instance, Boa et al. (2024) and Bouritsas et al. (2020) show that relatively low-order homomorphism / subgraph counts are often sufficient when they match the task’s structural signal, and Bouritsas et al. (2020) and Maskey et al. (2024) demonstrate on ZINC and molecular benchmarks that adding cycle counts up to length 6 improves performance, while going beyond that saturates or hurts performance. This is precisely the behavior our task-alignment perspective predicts when added expressivity no longer improves structure–label correlation but only increases complexity.
>
> Boa et al. (2024). *Homomorphism Counts as Structural Encodings for Graph Learning.* ICLR.
>
> Maskey et al. (2024). *Weisfeiler–Leman Go Loopy: A New Hierarchy for Graph Representational Learning.* NeurIPS.
>
> Bouritsas et al. (2020). *Improving Graph Neural Network Expressivity via Subgraph Isomorphism Counting.* TPAMI.
>
>
>
> > **W2**: “The paper assumes the CRA to be strongly simulatable, but did not mention under what condition is a CRA strongly simulatable, some examples would be good.”
>
> We kindly refer the reviewer to Section 3 (Lines 201-5 and 232-248) of the main paper, where we formally define the conditions under which a CRA is strongly simulatable (including a concrete illustrative example), and to Appendix B (“Simulatable Color-Refinement Algorithms”). Appendix B provides an intuitive explanation: “Intuitively, a GNN with $t > 0$ layers can be simulated if we can map its input domain to the set of graphs and achieve the same expressivity with a $t$-layer MPNN on this adapted set of graphs” alongside multiple concrete instances. In particular, Table 3 lists a range of simulatable CRAs, from $k$-WL to recent extensions in the literature. We will add an explicit pointer to Table 3 in Section 3 to make this clearer.
>
> > **W3**: “Bound looseness and constants”
>
> We acknowledge that our bound is qualitative and involves several constants. This degree of looseness is common in theoretically grounded generalization analyses, including Rademacher, PAC-Bayes, and NTK-based bounds, where achieving simultaneously tight constants and broad applicability remains an open challenge.
>
> The key contribution of our bound is not numerical tightness, but a structural decomposition of the generalization gap into (i) a capacity term and (ii) a task-alignment / structural-similarity term that models structure–label correlation. Purely capacity-based bounds (e.g., VC/norm-based or standard PAC-Bayes) cannot capture this phenomenon. Indeed, when structural alignment is ignored, existing bounds become vacuous by many orders of magnitude (e.g., our bound is on the order of 10^4 vs. 10^16 for related PAC-Bayes bounds; see Page 9 of our submission) and may even negatively correlate with empirical performance (Jiang et al., 2019). In contrast, our bound exhibits high rank correlation (Spearman / Kendall) with the true test error, demonstrating meaningful predictive power.
>
> Jiang et al. (2019). *Fantastic Generalization Measures and Where to Find Them.* ICLR.

---

> ### Author Response · Authors · 2025-11-23
>
> > **W4**: “Computational overhead of ζ-TMD and strong simulation”
>
> We would like to clarify that our approach does not require computing TMDs as a preprocessing step. Instead, TMDs are introduced as a theoretical tool to quantify structural similarities between training and test graphs to better understand generalization behavior. The practical implications we discuss in the manuscript therefore focus on modeling insights, such as beginning with simpler, less expressive baselines and incorporating domain knowledge or task-relevant data augmentations.
>
> If practitioners are interested in approximating these distances in practice, recent work by Chuang and Jegelka (2024) demonstrates that randomly initialized MPNNs can serve as effective proxies for TMD, exhibiting strong Pearson and Spearman correlations with exact distances. Analogously, for $\zeta$-TMD, one may approximate it via randomly initialized $\zeta$-MPNNs, providing a computationally tractable first proxy without requiring expensive exact computation.
>
> When exact or near-exact TMD values are still desired, several practical tricks can substantially reduce computational cost. First, instead of computing all train–test pairs, one can preselect a small candidate subset of training graphs for each test graph using cheap heuristics such as graph size, degree distributions, node-label histograms, or low-cost 1-WL feature histograms, and then evaluate TMD only within this subset. Second, the TMD computation itself can be sped up by truncating the tree depth, subsampling nodes (or neighborhoods), and replacing exact optimal transport with entropic-regularized (Sinkhorn) OT, which admits efficient batched GPU implementations. Finally, these computations can be parallelized across CPU cores and accelerated via vectorized GPU routines for the inner OT problems. Together, such approximations and engineering strategies make both TMD and $\zeta$-TMD more tractable for large graph datasets in practice.
> We will add these explanations to our manuscript.
>
> Chuang and Jegelka (2022).  *​​Tree Mover's Distance: Bridging Graph Metrics and Stability of Graph Neural Networks.* NeurIPS.
>
> > **W5**: “Experimental scope: The authors empirically show the bound demonstrates similar trend as empirical error in Fig 3, but this is not quantitative.”
>
> We compute rank correlations between $\zeta$-TMD and accuracy. On NC1, the association is perfect across depths (Spearman/Kendall = −1.00/−1.00). On BZR, we observe similarly strong correlations: depth 2 −0.90/−0.80, depth 4 −1.00/−1.00, depth 6 −0.96/−0.90. On Mutagenicity, we again see a consistent trend: 2-layer −0.96/−0.90, 4-layer −0.79/−0.71, 6-layer −0.71/−0.52. These results suggest that there is indeed a correlation between the $\zeta$-TMD and accuracy as our bound shows.
>
> Correlation Summary (MUTAG, NC1, BZR)
>
> | Dataset / Setting        | Spearman ($\rho$) | Kendall ($\tau_b$) |
> |--|--|--|
> | Mutagenicity (2 layers)  | -0.96 | -0.90 |
> | Mutagenicity (4 layers)  | -0.79 | -0.71 |
> | Mutagenicity (6 layers)  | -0.71 | -0.52 |
> | NC1 (2 layers) | -1.00  | -1.00 |
> | NC1 (4 layers) | -1.00 | -1.00 |
> | NC1 (6 layers) | -1.00 | -1.00 |
> | BZR (2 layers) | -0.90 | -0.80 |
> | BZR (4 layers) | -1.00 | -1.0 |
> | BZR (6 layers) | -0.96 | -0.90 |
>
>
> > **W6**: “Presentation polish: Some minor issues (placeholder captions, typographical glitches, compressed figure axes) impact readability, e.g. …"
>
> Thank you, we will fix the typos. The symbol $P$ denotes the palette, i.e., the co-domain (set of colors) of the CRA.
> Regarding the definition of expressivity: Indeed, under our definition, “more expressive” includes the possibility of being equally expressive. This is standard terminology in the literature; to explicitly exclude equality, one typically writes “strictly more expressive.”
>
> > **Q1**: “How would you recommend practitioners choose ζ in real tasks?”
>
> We recommend selecting $\zeta$ in consultation with domain experts and starting with lower expressivity levels. This is particularly important in OOD settings, which are common in practice, where higher expressivity increases the risk of overfitting. In fact, we show that overfitting can become more severe for highly expressive models when train–test structural discrepancies are large. Therefore, a gradual increase in $\zeta$, guided by validation performance and domain knowledge, is a practical strategy. For example, for molecule datasets, cycle counts, particularly those indicating aromatic rings, appear to be suitable choices for $\zeta$ in a variety of tasks. Please also refer to our response to **W1** for further discussion.

---

> > ### Author Response · Authors · 2025-11-23
> >
> > > **Q2**: Complexity of computing ζ-TMD and scalable approximations.
> >
> > The complexity of computing $\zeta$-TMD is equal to applying the transformation $R^{\zeta}$ induced by the CRA, followed by computing TMD on the transformed graphs. Thus, the overall cost depends on the complexity of the chosen CRA. The TMD component itself has complexity $\mathcal{O}(n^3log ⁡n)$.
> >
> > In practice, $\zeta$-TMD can be substantially accelerated through standard approximations, for example, subsampling rooted trees, reducing the number of OT iterations, or using randomly initialized WL-style feature maps. These approximations can reduce the cost to nearly linear time in graph size. We will include a discussion of these practical approximations in the revised version.
> >
> >
> > >  **Q3**: Assumption and robustness.
> >
> > Please note that the assumption  $y \sim \zeta-\text{TMD}$ is central to our framework: Modeling structure–label correlation is precisely what enables us to derive bounds that are meaningfully tighter than general i.i.d. PAC-Bayes bounds, and correlate with empirical performance of GNNs. Without such an assumption, one cannot theoretically distinguish aligned settings from arbitrary ones, and the analysis collapses to standard vacuous capacity-only bounds; see also our response to W3.
> >
> > That said, our empirical results suggest that the framework exhibits robustness even when the assumption is not perfectly satisfied: we observe high rank correlations between empirical performance and TMD-based measures across real datasets (as shown in our answer to W5), meaning that the trends remain stable rather than catastrophically failing in mildly misaligned regimes.
> > Importantly, one of the conclusions we draw from our work is that practitioners should select or design graph models that are sufficiently expressive to approximate a $\zeta$-TMD aligned with the label structure. In other words, it is of practical advantage to choose $\zeta$ so that the corresponding TMD captures relevant structural properties of the data, thereby making the assumption beneficial rather than restrictive.
> >
> > Regarding Assumption D.1, it is adapted from Ma et al. (2021) and serves as a mild regularity requirement ensuring that margin loss cannot deviate excessively when weights remain bounded; under i.i.d. data this property becomes trivial. Empirically, our high rank correlations show that the trends remain stable even if the alignment assumption is not perfectly satisfied. Exploring alternative robustness frameworks is an interesting future direction.
> >
> >
> > > **Q4**: Does your Lipschitz analysis extend to attention-based graph transformers or alternative poolings (e.g., mean/max, Set2Set)? What constraints would be required?
> >
> > Yes. Our Lipschitz analysis is formulated in a multiset framework, independent of sum-pooling assumptions. See Page 22, Lines 1145-7.: ”We reformulate and prove a more general version of Theorem 3.4, where we consider general message functions on multisets that are Lipschitz continuous on the multiset domain, […]“. Any aggregation or message function with a bounded Lipschitz constant, including mean pooling, max pooling, attention, or Set2Set, fits directly into our analysis. We will clarify this explicitly in the camera-ready version.
> >
> >
> > > **Q5**: Empirical alignment tests: Could you add a diagnostic that measures the correlation between label distributions and ζ-TMD distances to training graphs, to operationalize “alignment” before model design?
> >
> > In the standard practical scenario where a small labeled validation set is available, $\zeta$ can be chosen by computing the correlation (e.g., Spearman / Kendall) between validation accuracy and $\zeta$-TMD distance to the training set. This provides a lightweight diagnostic for selecting an appropriate $\zeta$ and avoiding overly expressive architectures. We will add this suggestion to the revised version.
> >
> > > **Q6**: Experiments: in Fig 2 and 3, the accuracy/empirical error increases/stabilises after certain TMD, can you think of any reasons?
> >
> > The observed plateau in Fig. 2 and 3 is expected and arises from the empirical setup: The plots show cumulative accuracy/error as increasingly distant test graphs are added (as described in Line 2347). Once nearly all test graphs beyond a certain $\zeta$-TMD threshold are included, only very few additional graphs remain, and these have a lower impact on the cumulative quantities. As a result, the empirical curve naturally stabilizes because the dataset is effectively exhausted in that region. Our bound is a worst-case guarantee and makes no assumptions about the empirical distribution of test graphs across distance buckets. Therefore, it increases monotonically with structural discrepancy by design.
> >
> > Thank you for pointing this out, we will add this important detail to the Figure captions and main paper.

---

### Author Response · Authors · 2025-12-03
**Final Summary Comment**

Dear Area Chair,

Thank you very much for handling our paper! In this comment, we summarize the main weaknesses raised by the reviewers and how we have addressed them.

**Access to $\eta$:** Multiple reviewers (cYcW, uzRa, wMWG) have raised the point that finding the right pseudo-metric $\eta$ can be hard. Indeed, selecting the perfect $\eta$ is equivalent to solving graph ML model selection. However, our work does not require us to find the perfect $\eta$. In contrast, our results suggest a principled way of selecting a strong $\eta$ (or finding a strong GNN): One should start with a low-expressivity model and slowly increase expressivity until increasing expressivity reduces performances. More fundamentally, our work serves as a theoretical justification for a common empirical observation in the associated literature: Increased expressivity is only of practical benefit if the added expressivity improves the structure-label correlation. We argue that our understanding and formalization of this observation is a valuable contribution to the literature even without the provision of a universal construction for $\eta$ that productively improves expressivity in all graph learning tasks (we doubt that such an $\eta$ exist and therefore prefer to leave the construction of suitable pseudo-metrics $\eta$ to different empirical subcommunities within which such a construction can sensibly be pursued).

**Bounds & Constants:**  Two reviewers (cYcW, uzRa) have raised concerns about the tightness of our bounds. This somewhat misses the point of our work: The key contribution of our bound is not numerical tightness, but a structural decomposition of the generalization gap into (i) a capacity term and (ii) a task-alignment / structural-similarity term that models structure–label correlation. Purely capacity-based bounds (e.g., VC/norm-based or standard PAC-Bayes) cannot capture this phenomenon. Indeed, when structural alignment is ignored, existing bounds become vacuous by many orders of magnitude (e.g., our bound is on the order of 10^4 vs. 10^16 for related PAC-Bayes bounds; see Page 9 of our submission) and may even negatively correlate with empirical performance (Jiang et al., 2019). In contrast, our bound exhibits high rank correlation (Spearman / Kendall) with the true test error, demonstrating meaningful predictive power.

**Experiments:** Multiple reviewers (cYcW, uzRa, wMWG) have asked for a wider experimental scope.  However, in several of these remarks, our additional real-world datasets in the main paper and in the appendix were not considered. If these results are taken into account, our empirical evaluation is already sufficiently broad: it includes a controlled synthetic task, six TU datasets (BZR, MUTAGENICITY, NCI109, PROTEINS, AIDS, COX2), and fixed-encoder studies in Appendix I. This scope is consistent with prior theoretical work on expressivity and generalization, and isolates architectural effects cleanly. We also evaluate 13 different expressivity levels, providing extensive $\eta$-ablations (MPNN, F-MPNN variants, Sub-G, L-G, LF-G, etc.). We furthermore agreed to provide approximate empirical results for the ogb-molhiv dataset.  We have also extended our quantitative results by performing a statistical analysis of accuracy and $\eta$-TMD on several datasets. These empirical measurements mirror our theoretical results and show that there is a strong correlation between accuracy and $\eta$-TMD.

We believe these changes to have convincingly addressed the reviewer’s comments. We thank them for their feedback and thank you for the time you invest in our work.

Kind regards,

The Authors of Submission5573

Jiang et al. (2019). *Fantastic Generalization Measures and Where to Find Them.* ICLR.

---

### Meta-Review · Area_Chair_CU1Z · 2026-01-03

**Summary:**

The paper addresses a fundamental question in Graph Representation Learning: the trade-off between model expressivity and generalization. It introduces a family of pseudometrics, $\zeta$-Tree Mover Distances ($\zeta$-TMDs), to quantify the structural similarity between training and test graphs relative to a specific task. The authors derive PAC-Bayes generalization bounds decomposing the error into model capacity and structural alignment terms. Theoretical and empirical results suggest that increased expressivity only improves performance if it aligns with the task's structural requirements (reducing the $\zeta$-TMD between train and test); otherwise, it may degrade generalization.

The reviewers generally appreciated the novelty of the theoretical framework and the intuition that "more expressivity is not always better." However, significant concerns were raised regarding:
1. Practicality: The difficulty of selecting the correct pseudometric $\zeta$ without prior domain knowledge (effectively the model selection problem).
2. Baselines & Context: The distinction from prior work (specifically Ma et al., 2021) and the relevance of orthogonal GNN issues (oversmoothing, heterophily).
3. Bounds: The looseness of the derived bounds and the reliance on comparing upper bounds to infer performance degradation.
4. Experiments: The reliance on synthetic and small-scale TUDatasets.

Given the theoretical insights and the plausible (albeit unilateral) defenses provided by the authors against generic critiques, I recommend accepting the paper.

**Reviewer Concerns:**

Due to the interruption of the rebuttal process, I have evaluated the author's responses as independent arguments.

Concerns Adequately Addressed:
- Differentiation from Prior Work: The authors convincingly distinguished their work from Ma et al. (2021) by clarifying that their framework supports end-to-end trainable GNNs and graph-space metrics, whereas Ma et al. rely on fixed encoders and Euclidean distances.
- Orthogonal GNN Issues: Critiques regarding oversmoothing and heterophily (raised by Reviewer wMWG) were effectively argued as orthogonal to the graph-level classification focus of this work.
- Computational Feasibility: Concerns about the cost of computing $\zeta$-TMD were addressed with new runtime tables and approximation strategies (e.g., subsampling).
- Bound Validity: While acknowledging the bounds are loose, the authors provided strong rank correlation data (Spearman/Kendall) to demonstrate that the metric meaningfully tracks empirical error.


Outstanding Concerns:
- Selection of $\zeta$: The practical challenge of selecting the correct pseudometric without prior domain knowledge remains a "model selection" bottleneck, as noted by multiple reviewers.
- Upper Bound Logic: The theoretical limitation noted by Reviewer uzRa remains: proving a looser upper bound for expressive models does not strictly prove that performance must degrade, only that it may.

**Reviewer Scores:**

- Reviewer cYcW: Remains 6. While the author's response on correlations was strong, the reviewer's initial concerns about the practicality of selecting $\zeta$ likely prevent a move to a higher score band.
- Reviewer uzRa: Remains 4. The authors addressed the novelty concern regarding Ma et al., but this reviewer expressed fundamental skepticism about the "upper bound" logic (arguing that a looser upper bound doesn't prove performance degradation). This theoretical disagreement likely keeps the score at 4.
- Reviewer wMWG: Remains 4. Although the authors refuted the relevance of heterophily/oversmoothing, the reviewer felt the paper lacked "actionable guidance" for practitioners. This practical gap likely keeps the score at 4.
- Reviewer iyDd: Remains 6. This reviewer was positive about the theoretical depth but felt the results needed significant contextualization. While the authors promised this in the revision, the score likely holds at 6.

---

### Decision · Program_Chairs · 2026-01-26

Accept (Poster)